# Revisiting Optimal Convergence Rate for Smooth and Non-convex Stochastic Decentralized Optimization

**Kun Yuan**[1,3]*, **Xinmeng Huang**[2]*, **Yiming Chen**[1,4]*, **Xiaohan Zhang**[2], **Yingya Zhang**[1], **Pan Pan**[1]

[1]DAMO Academy, Alibaba Group   [2]University of Pennsylvania
[3]Peking University   [4]MetaCarbon

{kun.yuan, yingya.zyy, panpan.pp}@alibaba-inc.com
xinmengh@sas.upenn.edu   yiming@metacarbon.vip   zxiaohan@seas.upenn.edu

## Abstract

Decentralized optimization is effective to save communication in large-scale machine learning. Although numerous algorithms have been proposed with theoretical guarantees and empirical successes, the performance limits in decentralized optimization, especially the influence of network topology and its associated weight matrix on the optimal convergence rate, have not been fully understood. While Lu and Sa [44] have recently provided an optimal rate for non-convex stochastic decentralized optimization with weight matrices defined over linear graphs, the optimal rate with *general* weight matrices remains unclear.

This paper revisits non-convex stochastic decentralized optimization and establishes an optimal convergence rate with general weight matrices. In addition, we also establish the optimal rate when non-convex loss functions further satisfy the Polyak-Lojasiewicz (PL) condition. Following existing lines of analysis in literature cannot achieve these results. Instead, we leverage the Ring-Lattice graph to admit general weight matrices while maintaining the optimal relation between the graph diameter and weight matrix connectivity. Lastly, we develop a new decentralized algorithm to nearly attain the above two optimal rates under additional mild conditions.

## 1   Introduction

**1.1 Motivation.** Decentralized optimization is an emerging paradigm for large-scale machine learning. By letting each node average with its neighbors, decentralized algorithms save great communication overhead compared to traditional approaches with a central server. While numerous effective decentralized algorithms (see, e.g., [51, 11, 62, 39, 5, 41, 78]) have been proposed in literature showing both theoretical guarantees and empirical successes, the performance limits in decentralized optimization have not been fully clarified. This paper provides further understandings in optimal convergence rate in decentralized optimization and develops algorithms to achieve these rates.

**Weight matrix and connectivity measure.** Assume $n$ computing nodes are connected by some network topology (graph). We associate the topology with a weight matrix $W = [w_{ij}]_{i,j=1}^n \in \mathbb{R}^{n \times n}$ in which $w_{ij} \in (0,1)$ if node $j$ is connected to node $i$ otherwise $w_{ij} = 0$. We introduce $\beta := \|W - \frac{1}{n}\mathbb{1}_n\mathbb{1}_n^T\| \in (0,1)$, where $\mathbb{1}_n = (1, \ldots, 1)^T \in \mathbb{R}^n$ has all entries being 1, as the *connectivity measure* to gauge how well the network topology is connected. Quantity $\beta \to 0$ (which implies $W \to \frac{1}{n}\mathbb{1}_n\mathbb{1}_n^T$) indicates a well-connected topology while $\beta \to 1$ (which implies $W \to I$) indicates a badly-connected topology. It is the weight matrix $W$ and its connectivity measure $\beta$ that bring major challenges to convergence analysis in decentralized optimization.

---

*Equal Contribution. Corresponding Author: Kun Yuan

36th Conference on Neural Information Processing Systems (NeurIPS 2022).

Table 1: Rate comparison between different algorithms in smooth and non-convex stochastic decentralized optimization. Parameter $n$ denotes the number of all computing nodes, $\beta \in [0,1)$ denotes the connectivity measure of the weight matrix, $\sigma^2$ measures the gradient noise, $b^2$ denotes data heterogeneity, and $T$ is the number of iterations. Other constants such as the initialization $f(x^{(0)}) - f^\star$ and smoothness constant $L$ are omitted for clarity. Detailed complexities with constant $L$ are listed in Table 6 in Appendix B. The definition of transient iteration complexity can be found in Remark 6 (the smaller the better). "LB" is lower bound while "UB" is upper bound. Notation $\tilde{O}(\cdot)$ hides all logarithm factors.

|    | **References** | **Gossip matrix** | **Convergence rate** | **Tran. iters.** |
|----|------------|--------------|----------------|--------------|
| LB | [44] | $\beta = \cos(\pi/n)$ | $\Omega\big(\frac{\sigma}{\sqrt{nT}} + \frac{1}{T(1-\beta)^{\frac{1}{2}}}\big)$ | $O\big(\frac{n}{(1-\beta)\sigma^2}\big)$ |
|    | Theorem 2 | $\beta \in [0, \cos(\pi/n)]$ | $\Omega\big(\frac{\sigma}{\sqrt{nT}} + \frac{1}{T(1-\beta)^{\frac{1}{2}}}\big)$ | $O\big(\frac{n}{(1-\beta)\sigma^2}\big)$ |
| UB | DSGD [30] | $\beta \in [0,1)$ | $O\big(\frac{\sigma}{\sqrt{nT}} + \frac{\sigma^{\frac{2}{3}}}{T^{\frac{2}{3}}(1-\beta)^{\frac{1}{3}}} + \frac{b^{\frac{2}{3}}}{T^{\frac{2}{3}}(1-\beta)^{\frac{2}{3}}}\big)$ | $O\big(\frac{n^3}{(1-\beta)^2\sigma^2}\big)^\dagger$ |
|    | D$^2$/ED [66] | $\beta \in [0,1)$ | $O\big(\frac{\sigma}{\sqrt{nT}} + \frac{1}{T(1-\beta)^3}\big)$ | $O\big(\frac{n}{(1-\beta)^6\sigma^2}\big)$ |
|    | DSGT [29] | $\beta \in [0,1)$ | $\tilde{O}\big(\frac{\sigma}{\sqrt{nT}} + \frac{\sigma^{\frac{2}{3}}}{T^{\frac{2}{3}}(1-\beta)^{\frac{1}{3}}}\big)$ | $\tilde{O}\big(\frac{n^3}{(1-\beta)^2\sigma^2}\big)$ |
|    | DeTAG [44] | $\beta \in [0,1)$ | $\tilde{O}\big(\frac{\sigma}{\sqrt{nT}} + \frac{1}{T(1-\beta)^{\frac{1}{2}}}\big)$ | $\tilde{O}\big(\frac{n}{(1-\beta)\sigma^2}\big)$ |
|    | MG-DSGD | $\beta \in [0,1)$ | $\tilde{O}\big(\frac{\sigma}{\sqrt{nT}} + \frac{1}{T(1-\beta)^{\frac{1}{2}}}\big)$ | $\tilde{O}\big(\frac{n}{(1-\beta)\sigma^2}\big)$ |

$^\dagger$ The complete complexity is $O\big(\frac{n^3}{(1-\beta)^2 \min\{\sigma^2,(1-\beta)^2 b^2\}}\big)$, which reduces to $O\big(\frac{n^3}{(1-\beta)^2\sigma^2}\big)$ for small $\sigma^2$.

**Decentralized optimization.** Decentralized approaches are built upon the partial averaging $x_i^+ = \sum_{j \in \mathcal{N}_i} w_{ij}x_j$ (or $x^+ = Wx$ in a more compact manner) where $\mathcal{N}_i$ is the set of neighbors of node $i$ (including node $i$ itself). The sparsity and connectivity of $W$ significantly influence the efficiency and effectiveness of the partial averaging. Decentralized gradient descent [51, 79], diffusion [11, 59] and dual averaging [19] are well-known decentralized methods. Other advanced algorithms, which can correct the bias caused by heterogeneous data distributions, include decentralized ADMM [45, 63], explicit bias-correction [62, 80, 38], gradient tracking [50, 17, 54, 74], and dual acceleration [60, 68].

In the stochastic regime, decentralize SGD [11, 59, 39, 30] and its momentum and adaptive variants [41, 78, 47] have attracted a lot of attentions. After being extended to directed topologies [5, 75], time-varying topologies [30, 48, 69], asynchronous settings [40], and data-heterogeneous scenarios [66, 73, 43, 2, 29], decentralize SGD has much more applicable scenarios with significantly improved performance. Decentralized algorithms are also closely related to federated learning methods [46, 64, 76, 28, 37] when they admit weight matrices alternating between $\frac{1}{n}\mathbb{1}_n\mathbb{1}_n^T$ and the identity matrix.

**Prior understandings in theoretical limits.** A series of pioneering works have shed lights on the theoretical limits in decentralized optimization. These works establish the optimal convergence rate for convex or non-stochastic decentralized optimization [60, 61, 65, 32], and propose decentralized algorithms that (nearly) match these optimal bounds [60, 61, 68, 65, 33, 32]. However, there are few studies on the theoretical limits in non-convex stochastic decentralized optimization, which is quite common in large-scale deep learning, even when the costs are smooth.

A recent novel work [44] establishes the optimal complexity for smooth and non-convex stochastic decentralized optimization. While this optimal bound is inspiring, it is valid for a restrictive family of weight matrix $W$ associated with the linear graph whose connectivity measure $\beta = \cos(\pi/n)^2$. However, the linear graph is just one of the commonly used topologies. A more difficult but fundamental question still remains open:

*What is the optimal convergence rate of smooth and non-convex stochastic decentralized optimization with a general weight matrix (that does not necessarily satisfy $\beta = \cos(\pi/n)$)?*

It is not known whether the lower bound in [44] still holds for other commonly-used weight matrices resulted from grids, toruses, hypercubes, etc. whose connectivity measure $\beta \neq \cos(\pi/n)$.

**Challenges.** Following existing lines of analysis, to our knowledge, cannot provide an assuring answer. It is known that the communication complexity in decentralized optimization is typically

---

$^2$See Corollary 1 in the latest arXiv version of [44] uploaded in Jan. 2022: arXiv:2006.08085v4.

Table 2: Rate comparison between different algorithms in smooth and non-convex stochastic decentralized optimization under the PL condition. Constants such as the initialization $f(x^{(0)}) - f^\star$ and smoothness constant $L$ are omitted for clarity. Detailed complexities with constants $L$ and $\mu$ are listed in Table 6 in Appendix B.

| | References | Gossip matrix | Convergence rate | Tran. iters. complexity |
|---|---|---|---|---|
| LB | Theorem 3 | $\beta \in [0, \cos(\pi/n)]$ | $\Omega\left(\frac{\sigma^2}{nT} + e^{-T(1-\beta)^{\frac{1}{2}}}\right)$ | $\tilde{O}\left(\frac{1}{(1-\beta)^{1/2}}\right)$ |
| UB | DSGD [30][†] | $\beta \in [0, 1)$ | $\tilde{O}\left(\frac{\sigma^2}{nT} + \frac{\sigma^2}{T^2(1-\beta)} + \frac{b^2}{T^2(1-\beta)^2}\right)$ | $\tilde{O}\left(\frac{n}{(1-\beta)\min\{1,(1-\beta)b^2\}}\right)$ |
| | DAGD[57][†‡] | $\beta \in [0, 1)$ | $\tilde{O}\left(\frac{\sigma^2}{nT} + e^{-T(1-\beta)^{\frac{1}{2}}}\right)$ | $\tilde{O}\left(\frac{1}{(1-\beta)^{1/2}}\right)$ |
| | D²/ED [2, 77] | $\beta \in [0, 1)$ | $\tilde{O}\left(\frac{\sigma^2}{nT} + \frac{\sigma^2}{T^2(1-\beta)} + e^{-T(1-\beta)}\right)$ | $\tilde{O}\left(\frac{n}{1-\beta}\right)$ |
| | DSGT [2] | $\beta \in [0, 1)$ | $\tilde{O}\left(\frac{\sigma^2}{nT} + \frac{\sigma^2}{T^2(1-\beta)} + e^{-T(1-\beta)}\right)$ | $\tilde{O}\left(\frac{n}{1-\beta}\right)$ |
| | DSGT [73] | $\beta \in [0, 1)$ | $\tilde{O}\left(\frac{\sigma^2}{nT} + \frac{\sigma^2}{T^2(1-\beta)^3} + e^{-T(1-\beta)}\right)$ | $\tilde{O}\left(\frac{n}{(1-\beta)^3}\right)$ |
| | MG-DSGD | $\beta \in [0, 1)$ | $\tilde{O}\left(\frac{\sigma^2}{nT} + e^{-T(1-\beta)^{\frac{1}{2}}}\right)$ | $\tilde{O}\left(\frac{1}{(1-\beta)^{1/2}}\right)$ |

[†] This rate is derived under the strongly-convex assumption.

[‡] This rate is achieved by utilizing increasing (non-constant) mini-batch sizes.

proportional to the diameter $D$ of the network topology [60]. Establishing the relation between $D$ and $\beta$ is key to clarifying the influence of the weight matrix on optimal convergence rate. Existing works [60, 61, 44] utilize the linear graph to establish the optimal relation as $D = \Theta(1/\sqrt{1-\beta})$. However, weight matrices associated with linear graph are quite *limited*. Given a fixed network size $n$ (which is a prerequisite when discussing complexities in distributed stochastic optimization [44, 24, 31, 21]), they have to satisfy $\beta = \cos(\pi/n)$. Weight matrices with other $\beta$ cannot be implied from the linear graph. To examine the lower bounds with a more general weight matrix $W$, [65] proposes a wider class of networks named linear-star graph, which, however, leads to *suboptimal* relation between $D$ and $\beta$. As a result, a network topology that admits general weight matrices $W$ while maintaining the optimal relation between $D$ and $\beta$ is critical to clarify the above fundamental question.

**1.2 Main results.** This paper successfully identifies the desired network topology which facilitates the theoretical limits over the general weight matrices. Our contributions are summarized as follows.

- We discover that ring-lattice graph maintains the optimal relation between $D$ and $\beta$ for a much broader class of weight matrices. Given any fixed $n$ and $\beta \in [0, \cos(\pi/n)]$, we can always construct a weight matrix $W \in \mathbb{R}^{n \times n}$ associated with some ring-lattice graph, whose connectivity measure is $\beta$, that satisfies $D = \Theta(1/\sqrt{1-\beta})$. Since $\cos(\pi/n)$ approaches to 1 when $n$ is large, the ring-lattice graph admits far broader weight matrices than linear graph used in [60, 61, 44].

- With the above results, we provide the optimal convergence rate for smooth and non-convex stochastic decentralized optimization that holds for any weight matrix with $\beta \in [0, \cos(\pi/n)]$. In contrast, the optimal rate in [44] only applies to weight matrix satisfying $\beta = \cos(\pi/n)$. We also provide the optimal convergence rate when the non-convex cost functions further satisfy the Polyak-Lojasiewicz (PL) condition [27] for any weight matrix with $\beta \in [0, \cos(\pi/n)]$.

- We prove that the above two optimal complexities can be nearly-attained (under mild assumptions and up to the condition number $L/\mu$ as well as other logarithmic factors) by simply integrating multiple gossip communication [42, 58] and gradient accumulation [60, 57, 44] to the vanilla decentralized SGD (D-SGD) algorithm. Compared to DeTAG [44], the extended D-SGD avoids the gradient tracking steps and saves half of the communication overheads. Compared to DAGD [57], the extended D-SGD achieves the near-optimality using constant mini-batch sizes and has theoretical guarantees in the non-convex (as well as the PL) scenario.

All established results in this paper as well as those of existing state-of-the-art decentralized algorithms are listed in Tables 1 and 2. The smoothness constant $L$ and PL constant $\mu$ are omitted in these tables for clarity. The detailed complexities of these algorithms with constant $L$ and $\mu$ are listed in Tables 6 and 7 in Appendix B.

**1.3 Other related works.** Some other related works are discussed as follows.

**Lower bounds and optimal algorithms.** The lower bounds and optimal algorithms for stochastic and convex optimization have been intensively studied in [35, 36, 55, 1, 18, 15]. References [9, 10] established the lower bounds to find stationary solutions for non-convex optimization, and [21, 3] proposed algorithms with near-optimal convergence rate for non-convex finite-sum minimization. In deterministic decentralized optimization, the optimal convergence rate was established for strongly convex optimization [60], non-smooth convex optimization [61], and non-convex optimization [65]. Furthermore, the lower bounds and optimal algorithms over time-varying topologies are studied in [33, 57, 58, 26]. In stochastic decentralized optimization, works [77] and [30] established the lower bounds for strongly-convex decentralized SGD. [44] proves the optimal complexities for non-convex problems when weight matrices satisfy $\beta = \cos(\pi/n)$.

**Ring-lattice Graph.** The ring-lattice graph was used in [70, 22, 12] as a basis model to generate the small world graph. However, its relation between graph diameter and connectivity measure is not clarified to our knowledge. A related work [72] examines the robustness of the ring-lattice graph, but does not discuss its connectivity measure. Furthermore, ring-lattice has never been utilized to establish the influence of the weight matrix on the optimal rate in decentralized optimization before.

## 2 Problem setup

**Problem setup.** Consider the following problem with a network of $n$ computing nodes:

$$\min_{x \in \mathbb{R}^d} f(x) = \frac{1}{n} \sum_{i=1}^{n} f_i(x) \quad \text{where} \quad f_i(x) := \mathbb{E}_{\xi_i \sim D_i} F(x; \xi_i). \tag{1}$$

Function $f_i(x)$ is local to node $i$, and random variable $\xi_i$ denotes the local data that follows distribution $D_i$. Each local data distribution $D_i$ can be different across all nodes.

**Assumptions.** The optimal convergence rate is established under the following assumptions.

- **Function class.** We assume each local loss function is $L$-smooth, i.e.,

$$\|\nabla f_i(x) - \nabla f_i(y)\| \le L\|x - y\|, \quad \forall x, y \in \mathbb{R}^n, \tag{2}$$

  for some constant $L > 0$. We let the class $\mathcal{F}_L$ denote the set of all functions satisfying (2). We may also assume the global loss $f(x) = \frac{1}{n} \sum_{i=1}^{n} f_i(x)$ satisfies the $\mu$-PL condition:

$$2\mu(f(x) - f^\star) \le \|\nabla f(x)\|^2, \quad \forall x \in \mathbb{R}^n, \tag{3}$$

  where $f^\star$ is the optimal function value in problem (1) and $\mu > 0$ is some constant. We let the function class $\mathcal{F}_{L,\mu}$ denote the set of all functions satisfying both (2) and (3).

- **Oracle class.** We assume each node $i$ interacts with $f_i(x)$ via the stochastic gradient oracle $\tilde{g}_i(x, \xi)$ where $\xi$ is a random variable. In addition, we assume oracle $\tilde{g}_i(x, \xi)$ satisfies

$$\mathbb{E}_\xi[\tilde{g}_i(x, \xi)] = \nabla f_i(x) \quad \text{and} \quad \mathbb{E}_\xi\|\tilde{g}_i(x, \xi) - \nabla f_i(x)\|^2 \le \sigma^2. \tag{4}$$

  We let $\mathcal{O}_{\sigma^2}$ denote the class of all such stochastic gradient oracles.

- **Weight matrix class.** We let $\mathcal{W}_{n,\beta}$ denote the class of all gossip matrices $W \in \mathbb{R}^{n \times n}$ satisfying

$$W \mathbb{1}_n = \mathbb{1}_n, \quad \mathbb{1}_n^T W = \mathbb{1}_n^T, \quad \|W - \frac{1}{n}\mathbb{1}_n\mathbb{1}_n^T\| = \beta \in [0, 1), \tag{5}$$

  where $\mathbb{1}_n = (1, \dots, 1)^T \in \mathbb{R}^n$ with all entries being 1. If $W$ is positive semi-definite, then connectivity measure $\beta$ is its second largest eigenvalue.

- **Algorithm class.** We consider an algorithm $A$ in which each node $i$ accesses an unknown local function $f_i$ via the stochastic gradient oracle $\tilde{g}_i(x; \xi_i) \in \mathcal{O}_{\sigma^2}$. Each node $i$ running algorithm $A$ will maintain a local model copy $x_i^{(k)}$ at iteration $k$. We assume $A$ to follow the partial averaging policy, i.e., each node communicates via protocol

$$z_i^{(k)} = \sum_{j \in \mathcal{N}_i} w_{ij} y_j^{(k)}, \quad \forall i \in [n]$$

  for some $W = [w_{ij}]_{i,j=1}^n \in \mathcal{W}_{n,\beta}$ where $[n] := \{1, \cdots, n\}$, and $y$ and $z$ are the input and output variables of the communication protocol. In addition, we assume $A$ to follow the zero-respecting

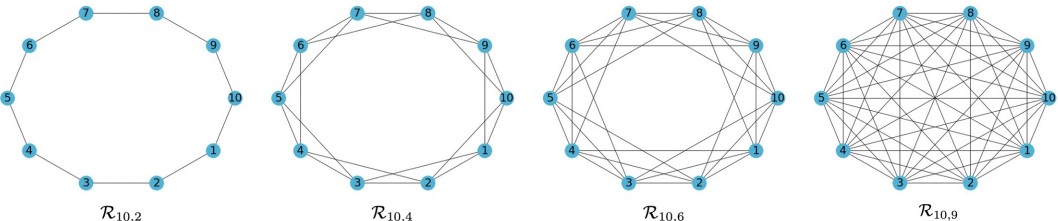

Figure 1: An illustration of the ring-lattice graph with size 10 and degrees 2, 4, 6, 9. It is observed that $\mathcal{R}_{10,2}$ is a ring graph while $\mathcal{R}_{10,9}$ is a complete graph.

policy [9, 10, 25]. Informally speaking, the zero-respecting policy requires that the number of non-zero entries of local model copy $x_i^{(k)}$ can only be increased by sampling its own stochastic gradient oracle or interacting with the neighboring nodes. The zero-respecting policy is widely followed by all methods in Tables 1 and 2 and their momentum and adaptive variants. We let $\mathcal{A}_W$ be the set of all algorithms following the partial averaging and zero-respecting policies.

**Lower bound metrics.** With the above classes, we now introduce the measures for convergence rate's lower bound. Given loss functions $\{f_i\}_{i=1}^n \subseteq \mathcal{F}_L$ (or $\{f_i\}_{i=1}^n \subseteq \mathcal{F}_{L,\mu}$), stochastic gradient oracles $\{\tilde{g}_i\}_{i=1}^n \subseteq \mathcal{O}_{\sigma^2}$, a weight matrix $W \in \mathcal{W}_{n,\beta}$, an algorithm $A \in \mathcal{A}_W$, the number of gradient queries (or communications rounds) $T$, we let $\hat{x}_{A,\{f_i\}_{i=1}^n,\{\tilde{g}_i\}_{i=1}^n,W,T}$ (which will be denoted as $\hat{x}$ when there is no ambiguity) be the output of algorithm $A$ with $\hat{x} \in \mathrm{Span}\left(\{\{x_j^{(t)}\}_{j=1}^n\}_{t=1}^T\right)$. The lower bound for smooth and non-convex stochastic decentralized optimization is defined as

$$\inf_{A \in \mathcal{A}_W} \sup_{W \in \mathcal{W}_{n,\beta}} \sup_{\{\tilde{g}_i\}_{i=1}^n \in \mathcal{O}_{\sigma^2}} \sup_{\{f_i\}_{i=1}^n \subseteq \mathcal{F}_L} \mathbb{E}\|\nabla f(\hat{x}_{A,\{f_i\}_{i=1}^n,\{\tilde{g}_i\}_{i=1}^n,W,T})\|^2. \tag{6}$$

The lower bound under the PL condition is defined as

$$\inf_{A \in \mathcal{A}_W} \sup_{W \in \mathcal{W}_{n,\beta}} \sup_{\{\tilde{g}_i\}_{i=1}^n \in \mathcal{O}_{\sigma^2}} \sup_{\{f_i\}_{i=1}^n \subseteq \mathcal{F}_{L,\mu}} \mathbb{E}[f(\hat{x}_{A,\{f_i\}_{i=1}^n,\{\tilde{g}_i\}_{i=1}^n,W,T}) - f^\star]. \tag{7}$$

**Fixed network size $n$.** This paper considers *stochastic* decentralized optimization where *the network size $n$ is a fixed constant*. A fixed $n$ is a prerequisite in distributed stochastic optimization which enables distributed algorithms to achieve the linear speedup in convergence rate $O(\sigma/\sqrt{nT})$, see the algorithms listed in Tables 1 and 2. In decentralized deterministic optimization, however, size $n$ does not appear in the convergence rate. Thus, it does not need to be specified and can be varied freely when establishing the lower bounds [60, 61].

## 3    Ring-Lattice graph

The main challenge in analyzing theoretical limits in decentralized optimization lies in clarifying the influence of network topology and its associated weight matrix. It is intuitive that the graph diameter $D$ affects the convergence rate. In the worst case, one node needs to take at least $D$ communication steps to transmit its own gradient information to the other. If the relation between $D$ and $\beta$ is established, the influence of the weight matrix on lower bounds will become clear. However, existing results either establishes suboptimal relation between $D$ and $\beta$ [65] or admits restrictive weight matrices [60, 61, 44] given fixed $n$. Now we show that the ring-lattice graph maintains the optimal relation between $D$ and $\beta$ for a broad class of weight matrices. All proof details are in Appendix C.

**Definition 1** (RING-LATTICE GRAPH [71])**.** *Given any positive integers $n, k$ such that $k$ is even and $2 \leq k < n - 1$, the $k$-regular ring-lattice graph over nodes $\{1, \cdots, n\}$, denoted by $\mathcal{R}_{n,k}$, is an undirected graph in which each node $i \in [n]$ is connected with the set of nodes $\{(i + \ell) \bmod n : 1 \leq |\ell| \leq k/2\}$ where $(i + \ell) \bmod n$ is defined as*

$$(i + \ell) \bmod n = \begin{cases} i + \ell & \text{if } 1 \leq i + \ell \leq n; \\ i + \ell + n & \text{if } i + \ell < 1; \\ i + \ell - n & \text{if } i + \ell > n. \end{cases}$$

*Apparently, each node in $\mathcal{R}_{n,k}$ has degree $k$.*

Ring-lattice is a $k$-regular graph. When $k = 2$, ring-lattice will reduce to the ring graph. While Definition 1 excludes the case in which $k = n-1$, the complete graph can be exceptionally viewed as the ring lattice with degree $k = n - 1$. Therefore, for general $2 \le k \le n - 1$, $\mathcal{R}_{n,k}$ can be regarded as an intermediate state between the ring and complete graphs, see the illustration in Fig. 1. The following lemma clarifies the diameter of $\mathcal{R}_{n,k}$.

**Lemma 1.** *The diameter of the ring-lattice graph $\mathcal{R}_{n,k}$ is $D = \Theta(n/k)$.*

It is derived from the above lemma that the diameters of $\mathcal{R}_{n,2}$ and $\mathcal{R}_{n,n}$ are $\Theta(n)$ and $\Theta(1)$, respectively, which are consistent with the ring and complete graphs.

The following fundamental theorem establishes the relation between $D$ and connectivity measure $\beta$.

**Theorem 1.** *Given a fixed $n$ and any $\beta \in [0, \cos(\pi/n)]$, we can always construct a ring-lattice graph $\mathcal{R}_{n,k}$ so that*

  *(1) it has an associated weight matrix $W \in \mathcal{W}_{n,\beta}$, i.e., $W \in \mathbb{R}^{n \times n}$ and $\|W - \frac{1}{n}\mathbb{1}_n\mathbb{1}_n^T\| = \beta$,*

  *(2) its diameter $D$ satisfies $D = \Theta(1/\sqrt{1-\beta})$.*

**Remark 1.** *The works [60, 61, 44] utilize the linear graph to establish the same relation $D = \Theta(1/\sqrt{1-\beta})$. However, given fixed network size $n$, the linear graph admits limited weight matrices with $\beta = \cos(\pi/n)$.*

**Remark 2.** *Sun and Hong [65] propose a slightly wider class of networks named linear-star graph, which, however, leads to a suboptimal relation $D = \Omega(1/[n(1-\beta)])$ [65, Proof of Corollary 3.1].*

**Remark 3.** *It is worth noting that for deterministic decentralized optimization where $n$ may vary, one can tune $n$ to make $\beta = \cos(\pi/n)$ lie in the full interval $(0,1)$ as shown in [60, 61]. In other words, [60, 61] proved that $D = \Theta(1/\sqrt{1-\beta})$ for any $\beta \in (0,1)$ if $n$ is a varying parameter. However, this result is not valid in stochastic scenarios where $n$ is required to be fixed.*

## 4 Lower bounds with general weight matrices

This section will establish the lower bounds for smooth and non-convex decentralized stochastic optimization. All proof details are in Appendix D. When the cost function does not satisfy the PL condition (3), we will derive that the convergence rate is lower bounded by $\Omega(\frac{\sigma}{\sqrt{nT}} + \frac{D}{T})$. This result together with Theorem 1 leads to:

**Theorem 2.** *For any $L > 0$, $n \ge 2$, $\beta \in [0, \cos(\pi/n)]$, and $\sigma > 0$, there exists a set of loss functions $\{f_i\}_{i=1}^n \subseteq \mathcal{F}_L$, a set of stochastic gradient oracles $\{\tilde{g}_i\}_{i=1}^n \subseteq \mathcal{O}_{\sigma^2}$, and a weight matrix $W \in \mathcal{W}_{n,\beta}$, such that it holds for any $A \in \mathcal{A}_W$ starting form $x^{(0)}$ that*

$$\mathbb{E}\|\nabla f(\hat{x}_{A, \{f_i\}_{i=1}^n, \{\tilde{g}_i\}_{i=1}^n, W, T})\|^2 = \Omega\left(\frac{\sigma\sqrt{L}}{\sqrt{nT}} + \frac{L}{T\sqrt{1-\beta}}\right). \tag{8}$$

**Remark 4** (**Comparison with existing bounds**)**.** *Comparing to the lower bound established in [44], Theorem 2 admits a much broader class of weight matrices with $\beta \in [0, \cos(\pi/n)]$. While the interval $[0, \cos(\pi/n)] \subset [0,1)$, it approaches to $[0,1)$ as $n$ goes large. Such interval is broad enough to cover most weight matrices (generated through the Laplacian rule $W = I - L/d_{\max}$) resulted from common topologies such as line, ring, grid, torus, hypercube, exponential graph, complete graph, Erdos-Renyi graph, geometric random graph, etc. whose $\beta$ lies in the interval $[0, \cos(\pi/n)]$ when $n$ is sufficiently large, see details in Appendix D.*

**Remark 5** (**Linear speedup**)**.** *The first term $1/\sqrt{nT}$ dominates (8) when $T$ is sufficiently large, which will require $T = O(1/(n\epsilon^2))$ (it is inversely proportional to $n$) iterations to reach accuracy $\epsilon$). Therefore, if the term involving $nT$ is dominating the rate for some $T$, we say the algorithm is in its linear-speedup stage. Linear speedup is a fundamental property that distributed algorithms enjoy.*

**Remark 6** (**Transient iteration complexity**)**.** *Due to the topology-incurred overhead in convergence rate, a decentralized algorithm has to experience a transient stage to achieve linear-speedup. Transient iterations are referred to those iterations before an algorithm reaches linear-speedup stage. To reach linear speedup in the lower bound (8), $T$ has to satisfy $\frac{1}{T\sqrt{1-\beta}} \le \frac{\sigma}{\sqrt{nT}}$ and hence $T = \mathcal{O}(\frac{n}{(1-\beta)\sigma^2})$, which is the transient iteration complexity shown in the first line in Table 1. The smaller the transient iterations are, the faster the algorithm can achieve linear speedup. Transient iteration complexity is an important metric to evaluate how sensitive the algorithm is to $\beta$.*

---
**Algorithm 1:** Decentralized SGD with multiple gossip steps (MG-DSGD)
---
**Require:** Initialize $x_i^{(0)} = 0$, the weight matrix $W$, and the rounds of gossip steps $R$.

**for** $k = 0, 1, 2, ..., K$, every node $i$ **do**

    Sample $\{\xi_i^{(k,r)}\}_{r=1}^R$ independently and let $g_i^{(k)} = \frac{1}{R}\sum_{r=1}^R \nabla F(x_i^{(k)}; \xi_i^{(k,r)})$;   $\triangleright$ grad. accumulation

    Update $\phi_i^{(k+1)} = x_i^{(k)} - \gamma g_i^{(k)}$;            $\triangleright$ local stochastic gradient descent

    Update $x_i^{(k+1)} = \textbf{FastGossipAverage}(\{\phi_i^{(k+1)}\}_{i=1}^n, W, R)$;     $\triangleright$ multiple gossips

---

---
**Algorithm 2:** $x_i = \textbf{FastGossipAverage}(\{\phi_i\}_{i=1}^n, W, R)$
---
**Require:** $\{\phi_i\}_{i=1}^n$, $W$, $R$, and $\beta$; let $z_i^{(0)} = z_i^{(-1)} = \phi_i$ and $\eta = \frac{1-\sqrt{1-\beta^2}}{1+\sqrt{1+\beta^2}}$.

**for** $r = 0, 1, 2, ..., R-1$, every node $i$ **do**

    Update $z_i^{(r+1)} = (1+\eta)\sum_{j \in \mathcal{N}_i} w_{ij} z_i^{(r)} - \eta z_i^{(r-1)}$;        $\triangleright$ fast gossip averaging

**Output:** $x_i = z_i^{(R)}$;

---

When the cost function satisfies PL condition (3), we prove the convergence rate is lower bounded by $\Omega(\frac{\sigma}{\sqrt{nT}} + \exp(-CT/D))$ where $C$ is some constant. This result together with Theorem 1 leads to:

**Theorem 3.** *For any $L > 0$, $n \geq 2$, $\beta \in [0, \cos(\pi/n)]$, and $\sigma > 0$, there exists a set of loss functions $\{f_i\}_{i=1}^n \subseteq \mathcal{F}_{L,\mu}$, a set of stochastic gradient oracles $\{\tilde{g}_i\}_{i=1}^n \subseteq \mathcal{O}_{\sigma^2}$, and a weight matrix $W \in \mathcal{W}_{n,\beta}$, such that it holds for any $A \in \mathcal{A}_W$ starting from $x^{(0)}$ that*

$$\mathbb{E}[f(\hat{x}_{A, \{f_i\}_{i=1}^n, \{\tilde{g}_i\}_{i=1}^n, W, T})] - f^\star = \Omega\left(\frac{\sigma^2}{\mu nT} + \frac{\mu}{L}\exp(-T\sqrt{\mu/L}\sqrt{1-\beta})\right). \tag{9}$$

*Rate (9) results in an $\tilde{\mathcal{O}}((1-\beta)^{-\frac{1}{2}})$ transient complexity.*

Expression (9) is the first lower bound for smooth and non-convex stochastic decentralized optimization under the PL condition to our knowledge. Please note that the above lower bound also applies to strongly-convex stochastic decentralized optimization, see the construction details in Appendix D.

## 5 DSGD with multiple gossip communications

This section will present a decentralized algorithm that achieve the lower bounds established in Sec. 4 up to constants $L$ and $\mu$. The new algorithm is a direct extension of the vanilla decentralized SGD (DSGD) [11, 30]. Inspired by the algorithm development in [57, 44], we add two additional components to DSGD: gradient accumulation and multiple-gossip communication. The main recursions are listed in Algorithm 1 which utilizes the fast gossip average step [42] in Algorithm 2. We call the new algorithm as MG-DSGD where "MG" indicates "multiple gossips". All proofs are in Appendix E.

The DeTAG algorithm proposed in [44] applies the gradient accumulation and multiple-gossip communication techniques to gradient tracking, which achieves the optimal rate in non-convex (without PL condition) stochastic decentralized optimization. This paper finds in MG-DSGD that the multiple gossip technique can significantly reduce the influence of data heterogeneity, and the gradient tracking step can be saved to achieve the convergence optimality. Similar observations also appeared in [57] albeit in the strongly-convex scenario. Removing gradient tracking from algorithm updates brings two benefits. First, it saves half of the communication overhead and memory storage per iteration. Second, it enables numerous effective acceleration techniques designed exclusively for decentralized SGD such as Quasi-global momentum [41], DecentLaM [78], adaptive momentum [47], etc. which cannot be easily extended to gradient tracking.

Let $T$ be the toal number of gradient queries (or decentralized communications) at each node. Since each node takes $R$ gradient queries and $R$ communications at round $k$, it holds that $T = KR$ when MG-DSGD finishes after $K$ rounds. The following theorems clarify the convergence rate of MG-DSGD where $T = KR$. We omit the initialization constants below. Our analysis techniques relate to the classic SGD analysis but with inaccurate oracles in the nonconvex case [20].

**Theorem 4.** *Given $L > 0$, $n \geq 2$, $\beta \in [0, 1)$, $\sigma > 0$, and let $A$ denote Algorithm 1. Assuming that $\frac{1}{n} \sum_{i=1}^{n} \|\nabla f_i(x) - \nabla f(x)\|^2 \leq b^2$ for any $x \in \mathbb{R}^n$ and the number of gossip rounds $R$ is set as in Appendix E, the convergence of $A$ can be bounded for any $\{f_i\}_{i=1}^{n} \subseteq \mathcal{F}_L$ and any $W \in \mathcal{W}_{n,\beta}$ that*

$$\frac{1}{K} \sum_{k=1}^{K} \mathbb{E} \|\nabla f(\bar{x}^{(k)})\|^2 = \tilde{O} \left( \frac{\sigma \sqrt{L}}{\sqrt{nT}} + \frac{L}{T\sqrt{1-\beta}} \right) \quad \text{where} \quad \bar{x}^{(k)} = \frac{1}{n} \sum_{i=1}^{n} x_i^{(k)}. \tag{10}$$

**Remark 7.** *The rate (10) matches with the lower bound (8) up to logarithm factors. Furthermore, we remark that the gradient similarity assumption $\frac{1}{n} \sum_{i=1}^{n} \|\nabla f_i(x) - \nabla f(x)\|^2 \leq b^2$ is not required to obtain the lower bound in Theorem 2. Due to these two restrictions, MG-DSGD is a nearly-optimal (not fully-optimal) algorithm. However, while the analysis in MG-DSGD needs to assume bounded gradient similarity $b^2$, it only affects the convergence rate as a logarithm term hidden in the $\tilde{\mathcal{O}}(\cdot)$ notation. If one wants to completely removes the gradient similarity assumption from Theorem 4, he/she can refer to the techniques in gradient tracking [44, 73, 43].*

The rate (10) matches with the lower bound (8) up to logarithm factors. The comparison between MG-DSGD with other state-of-the-art algorithms is listed in Table 1. MG-DSGD matches DeTAG in convergence rate and saves half of the communications, see experiments in Fig. 2. While the analysis in MG-DSGD needs to assume bounded data heterogeneity $b^2$, it only affects the convergence rate as a logarithm term hidden in the $\tilde{\mathcal{O}}(\cdot)$ notation.

**Theorem 5.** *Under the same assumptions as in Theorem 4 and setting $R$ as in Appendix E, the convergence of $A$ can be bounded for any loss functions $\{f_i\}_{i=1}^{n} \subseteq \mathcal{F}_{L,\mu}$, and any $W \in \mathcal{W}_{n,\beta}$ by*

$$\frac{1}{K} \sum_{k=1}^{K} \mathbb{E}[f(\bar{x}^{(k)}) - f^\star] = \tilde{O} \left( \frac{L\sigma^2}{\mu^2 nT} + \exp(-\mu T \sqrt{1-\beta}/L) \right). \tag{11}$$

**Remark 8.** *Comparing (11) and lower bound (9), we find its dependence on $\sigma^2$, $n$, $T$, and $1 - \beta$ is optimal while that on $L$ and $\mu$ is not. One reason is that our algorithm does not involve any Nesterov acceleration [52] mechanism. The established upper bound in a recent work [57] is closer to the lower bound (9) by utilizing Nesterov acceleration, see Table 7 in Appendix B. However, it requires an increasingly large mini-batch of data per iteration, and only works for the strongly-convex scenario. It is not known whether the acceleration mechanisms in [57] can be extended to the non-convex and PL scenario. The bound (11) is state-of-the-art under the general PL condition to our knowledge.*

## 6 Experiments

This section will validate our theoretical results by empirically comparing different decentralized algorithms DSGD [39], D$^2$ [67], DSGT [82], DeTAG [44] and MG-DSGD in deep learning.

**Implementation setup.** We implement all decentralized algorithms with PyTorch [53] 1.6.0 using NCCL 2.8.3 (CUDA 10.1) as the communication backend, in which the partial averaging primitive is implemented by NCCL send/receive primitives. For parallel SGD, we used PyTorch's native Distributed Data Parallel (DDP) module. All the models and training scripts in this section run on servers with 8 NVIDIA V100 GPUs with each GPU treated as one node. Similar to existing works [44], we use Ring graph throughout all experiments.

**Tasks and training details.** A series of experiments are carried out with CIFAR-10 [34] and ImageNet [16] to compare the aforementioned methods. For CIFAR-10 dataset, it consists of 50,000 training images and 10,000 validation images in 10 classes. For ImageNet dataset, it consists of 1,281,167 training images and 50,000 validation images in 1000 classes. We utilize ResNet variants [23] on CIFAR-10 (ResNet-20 with 0.27M parameters and ResNet-18 with 11.17M parameters) and ResNet-50 on ImageNet. For training protocol, we train total 300 epochs and the learning rate is warmed up in the first 5 epochs and is decayed by a factor of 10 at 150 and 250-th epoch. For learning rate, we tuned a strong baseline in the PSGD setting (5e-3 for single node) and used the same setting in all decentralized methods. The batch size is set to 128 on each node. All CIFAR-10 experiments are repeated three times with different seeds. For fair comparison, we maintain the total number of gradient queries the same for algorithms in experiments, i.e., $T = KR$.

**Performance with homogeneous data.** Table 3 compares the proposed MG-DSGD with centralized parallel SGD and other baselines with homogeneous data distributions. For DeTAG and MG-DSGD, the communication round $R$ is set as 2. We also performed an ablation study on the effect of

Table 3: Accuracy comparison on CIFAR-10 with homogeneous data (8 nodes).

| METHODS | WITH MOMENTUM | | WITHOUT MOMENTUM | |
| | RESNET18 | RESNET20 | RESNET18 | RESNET20 |
|---|---|---|---|---|
| PSGD | 93.99 ± 0.52 | 91.62 ± 0.13 | 93.22 ± 0.19 | 89.06 ± 0.08 |
| DSGD | 94.54 ± 0.05 | 91.46 ± 0.06 | 93.67 ± 0.04 | 89.14 ± 0.08 |
| DSGT | 94.34 ± 0.04 | 91.64 ± 0.01 | 93.57 ± 0.02 | 88.85 ± 0.11 |
| $D^2$ | 92.51 ± 0.07 | 89.83 ± 0.08 | 92.18 ± 0.08 | 88.67 ± 0.11 |
| DeTAG | 94.40 ± 0.13 | **91.77 ± 0.25** | 93.17 ± 0.18 | 88.97 ± 0.11 |
| MG-DSGD | **94.57 ± 0.05** | **91.77 ± 0.09** | **93.75 ± 0.12** | **89.18 ± 0.08** |

Table 4: Accuracy comparison on CIFAR-10 with heterogeneous data (ResNet-20).

| METHODS | DSGD | DSGT | $D^2$ | DeTAG | MG-DSGD |
|---|---|---|---|---|---|
| $\alpha = 0.1$ | 58.14 ± 1.76 | 55.16 ± 0.42 | 53.85 ± 0.40 | 58.77 ± 1.30 | **59.42 ± 3.36** |
| $\alpha = 1$ | 84.74 ± 0.97 | 84.74 ± 0.32 | 84.1 ± 0.64 | 84.82 ± 0.19 | **84.91 ± 0.74** |
| $\alpha = 10$ | 90.19 ± 0.47 | 89.83 ± 0.12 | 89.80 ± 0.25 | 90.27 ± 0.08 | **90.43 ± 0.04** |

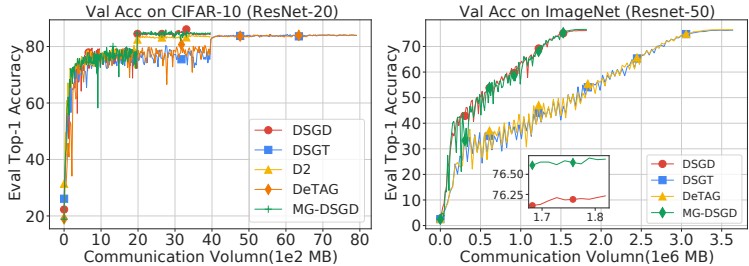

Figure 2: Accuracy in terms of the communication volumes. Left: CIFAR-10 with heterogeneous data ($\alpha = 1$); Right: ImageNet with homogeneous data.

momentum acceleration. Except for the ResNet-20 scenario with momentum in which MG-DSGD and DeTAG achieve the same performance, MG-DSGD consistently outperforms baseline methods in other scenarios. Moreover, MG-DSGD is more communication-efficient than DSGT and DeTAG.

**Performance with heteogeneous data.** Following previous works [81, 41], we simulate data heterogeneity among nodes with a Dirichlet distribution-based partitioning. A quantity $\alpha$ controls the degree of the data heterogeneity. The training data for a particular class tends to concentrate in a single node as $\alpha \to 0$, i.e. becoming more heterogeneous, while the homogeneous data distribution is achieved as $\alpha \to \infty$. We tested $\alpha = 0.1/1/10$ for all compared methods as corresponding to a setting with high/mild/low heterogeneity. As shown in Table 4, DeTAG and MG-DSGD outperform the other baselines by visible margins, and MG-DSGD is with slightly better accuracies. The left plot in Fig. 2 illustrates that MG-DSGD achieve a competitive accuracy to DeTAG using much less communication budgets on CIFAR-10 with $\alpha = 1$.

**Large-batch training in ImageNet.** To train ResNet-50 on ImageNet, utilization of the large-batch is necessary to speedup the training process. Since MG-DSGD is based on DSGD, we can extend the large-batch acceleration techniques introduced in [78, 41] to MG-DSGD to achieve better performance. These techniques are developed exclusively for DSGD and cannot be integrated to DeTAG easily. Following [78], we train total 120 epochs. The learning rate is warmed up in the first 20 epochs and is decayed in a cosine annealing scheduler.

Table 5: Performance of different methods (ImageNet).

| Methods | Acc. |
|---|---|
| DSGD | 76.23 ± 0.06 |
| DSGT | 76.37 ± 0.07 |
| DeTAG | 76.57 ± 0.02 |
| MG-DSGD | **76.64 ± 0.03** |

The results are listed in Table 5, showing that MG-DSGD has better accuracy than other approaches. Fig. 2(right) illustrates how different algorithms perform in terms of communication budgets used.

**More results.** In Appendix F, we depict the accuracy curves in terms of epochs (rather than communication volumes). The ablation on the influence of inner loop $R$ is also listed there.

## 7 Conclusion

This paper revisits the theoretical limits in smooth and non-convex stochastic decentralized optimization. Existing lower bound proved in [44] is only valid for special weight matrices with connectivity

measure $\beta = \cos(\pi/n)$. It is not known what the lower bound is when using weight matrices with other $\beta$. To address this issue, this paper establishes the lower bound for any weight matrix with $\beta \in [0, \cos(\pi/n)]$. We also clarify the lower bound when loss functions satisfy the PL condition. A new algorithm is proposed to match these lower bounds up to logarithm factors.

## Acknowledgements

The authors are grateful to all anonymous reviewers in NeurIPS 2022 for the very helpful discussions during the rebuttal process as well as useful references on accelerated decentralized gradient descent.

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
