# A Preliminary

**Notation.** We first introduce necessary notations as follows.

- $\mathbf{x}^{(k)} = [(x_1^{(k)})^T; (x_2^{(k)})^T; \cdots; (x_n^{(k)})^T] \in \mathbb{R}^{n \times d}$
- $\nabla F(\mathbf{x}^{(k)}; \xi^{(k)}) = [\nabla F_1(x_1^{(k)}; \xi_1^{(k)})^T; \cdots; \nabla F_n(x_n^{(k)}; \xi_n^{(k)})^T] \in \mathbb{R}^{n \times d}$
- $\nabla F(\mathbf{x}^{(k)}; \xi^{(k,r)}) = [\nabla F_1(x_1^{(k)}; \xi_1^{(k,r)})^T; \cdots; \nabla F_n(x_n^{(k)}; \xi_n^{(k,r)})^T] \in \mathbb{R}^{n \times d}$
- $\nabla f(\mathbf{x}^{(k)}) = [\nabla f_1(x_1^{(k)})^T; \nabla f_2(x_2^{(k)})^T; \cdots; \nabla f_n(x_n^{(k)})^T] \in \mathbb{R}^{n \times d}$
- $\bar{\mathbf{x}}^{(k)} = [(\bar{x}^{(k)})^T; (\bar{x}^{(k)})^T; \cdots; (\bar{x}^{(k)})^T] \in \mathbb{R}^{n \times d}$ where $\bar{x}^{(k)} = \frac{1}{n} \sum_{i=1}^{n} x_i^{(k)}$
- $W = [w_{ij}] \in \mathbb{R}^{n \times n}$ is the weight matrix.
- $\mathbb{1}_n = \text{col}\{1, 1, \cdots, 1\} \in \mathbb{R}^n$.
- Given two matrices $\mathbf{x}, \mathbf{y} \in \mathbb{R}^{n \times d}$, we define inner product $\langle \mathbf{x}, \mathbf{y} \rangle = \text{tr}(\mathbf{x}^T \mathbf{y})$ and the Frobenius norm $\|\mathbf{x}\|_F^2 = \langle \mathbf{x}, \mathbf{x} \rangle$.
- Given $W \in \mathbb{R}^{n \times n}$, we let $\|W\|_2 = \sigma_{\max}(W)$ where $\sigma_{\max}(\cdot)$ denote the maximum sigular value.

**DSGD in matrix notation.** The recursion of DSGD can be written in matrix notation:
$$\mathbf{x}^{(k+1)} = W\left(\mathbf{x}^{(k)} - \gamma \nabla F(\mathbf{x}^{(k)}; \xi^{(k)})\right) \tag{12}$$

**MG-DSGD in matrix notation.** The main recursion of MG-DSGD can be written in matrix notation:
$$\mathbf{g}^{(k)} = \frac{1}{R} \sum_{r=1}^{R} \nabla F(\mathbf{x}^{(k)}; \xi^{(k,r)}) \tag{13}$$
$$\mathbf{x}^{(k+1)} = \bar{M}(\mathbf{x}^{(k)} - \gamma \mathbf{g}^{(k)}) \tag{14}$$
where the weight matrix $\bar{M} = M^{(R)}$ and $M^{(R)}$ is achieved via the fast gossip averaging loop:
$$M^{(-1)} = M^{(0)} = I \tag{15}$$
$$M^{(r+1)} = (1+\eta)WM^{(r)} - \eta M^{(r-1)}, \quad \forall r = 0, 1, \cdots, R-1. \tag{16}$$

**Smoothness.** Since each $f_i(x) \in \mathcal{F}_L$ is $L$-smooth, it holds that $f(x) = \frac{1}{n} \sum_{i=1}^{n} f_i(x)$ is also $L$-smooth. As a result, the following inequality holds for any $\boldsymbol{x}, \boldsymbol{y} \in \mathbb{R}^d$:
$$f_i(\boldsymbol{x}) - f_i(\boldsymbol{y}) - \frac{L}{2}\|\boldsymbol{x} - \boldsymbol{y}\|^2 \leq \langle \nabla f_i(\boldsymbol{y}), \boldsymbol{x} - \boldsymbol{y} \rangle \tag{17}$$

**Network weighting matrix.** Since the weight matrix $W \in \mathcal{W}_{n,\beta}$, it holds that
$$\|W - \frac{1}{n}\mathbb{1}_n \mathbb{1}_n^T\|_2 = \beta. \tag{18}$$
Furthermore, one can establish the following upper result for the multi-gossip matrix $\bar{M}$.

**Proposition 1** (Proposition 3 of [42]). *Let $M^{(r)}$ defined by iterations (15) and (16), then it holds that for any $r = 0, \ldots, R$,*
$$M^{(r)}\mathbb{1}_n = [M^{(r)}]^T \mathbb{1}_n = \mathbb{1}_n, \quad \text{and} \quad \|M^{(r)} - \frac{1}{n}\mathbb{1}_n\mathbb{1}_n^T\|_2 \leq \sqrt{2}\left(1 - \sqrt{1-\beta}\right)^r. \tag{19}$$

Therefore, when $r$ grows, $M^{(r)}$ exponentially converges to $\frac{1}{n}\mathbb{1}_n\mathbb{1}_n^T$.

**Submultiplicativity of the Frobenius norm.** Given matrices $W \in \mathbb{R}^{n \times n}$ and $\mathbf{y} \in \mathbb{R}^{n \times d}$, it holds that
$$\|W\mathbf{y}\|_F \leq \|W\|_2 \|\mathbf{y}\|_F. \tag{20}$$
To verify it, by letting $y_j$ be the $j$-th column of $\mathbf{y}$, we have $\|W\mathbf{y}\|_F^2 = \sum_{j=1}^{d} \|Wy_j\|_2^2 \leq \sum_{j=1}^{d} \|W\|_2^2 \|y_j\|_2^2 = \|W\|_2^2 \|\mathbf{y}\|_F^2$.

## B  More detailed comparison

Table 6: Rate comparison in smooth and non-convex stochastic decentralized optimization. Parameter $n$ denotes the number of all computing nodes, $\beta \in [0, 1)$ denotes the connectivity measure of the weight matrix, $\sigma^2$ measures the gradient noise, $b^2$ denotes data heterogeneity, $L$ denotes smoothness constant, and $T$ is the number of iterations. The definition of transient iteration complexity can be found in Remark 6 (the smaller the better). "LB" is lower bound while "UB" is upper bound. Notation $\tilde{O}(\cdot)$ hides all logarithm factors.

| | References | Gossip matrix | Convergence rate | Tran. iters. |
|---|---|---|---|---|
| LB | [44] | $\beta = \cos(\pi/n)$ | $\Omega\big(\frac{\sigma\sqrt{L}}{\sqrt{nT}} + \frac{L}{T(1-\beta)^{\frac{1}{2}}}\big)$ | $O\big(\frac{nL}{(1-\beta)\sigma^2}\big)$ |
| | Theorem 2 | $\boldsymbol{\beta \in [0, \cos(\pi/n)]}$ | $\Omega\big(\frac{\sigma\sqrt{L}}{\sqrt{nT}} + \frac{L}{T(1-\beta)^{\frac{1}{2}}}\big)$ | $O\big(\frac{nL}{(1-\beta)\sigma^2}\big)$ |
| UB | DSGD [30] | $\beta \in [0, 1)$ | $O\big(\frac{\sigma\sqrt{L}}{\sqrt{nT}} + \frac{\sigma^{\frac{2}{3}}L^{\frac{2}{3}}}{T^{\frac{2}{3}}(1-\beta)^{\frac{1}{3}}} + \frac{b^{\frac{2}{3}}L^{\frac{2}{3}}}{T^{\frac{2}{3}}(1-\beta)^{\frac{2}{3}}}\big)$ | $O\big(\frac{n^3L}{(1-\beta)^2\sigma^2}\big)^{\dagger}$ |
| | D$^2$/ED [66] | $\beta \in [0, 1)$ | $O\big(\frac{\sigma\sqrt{L}}{\sqrt{nT}} + \frac{L}{T(1-\beta)^3}\big)$ | $O\big(\frac{nL}{(1-\beta)^6\sigma^2}\big)$ |
| | DSGT [29] | $\beta \in [0, 1)$ | $\tilde{O}\big(\frac{\sigma\sqrt{L}}{\sqrt{nT}} + \frac{\sigma^{\frac{2}{3}}L^{\frac{2}{3}}}{T^{\frac{2}{3}}(1-\beta)^{\frac{1}{3}}}\big)$ | $\tilde{O}\big(\frac{n^3L}{(1-\beta)^2\sigma^2}\big)$ |
| | DeTAG [44] | $\beta \in [0, 1)$ | $\tilde{O}\big(\frac{\sigma\sqrt{L}}{\sqrt{nT}} + \frac{L}{T(1-\beta)^{\frac{1}{2}}}\big)$ | $\tilde{O}\big(\frac{nL}{(1-\beta)\sigma^2}\big)$ |
| | MG-DSGD | $\beta \in [0, 1)$ | $\tilde{O}\big(\frac{\sigma\sqrt{L}}{\sqrt{nT}} + \frac{L}{T(1-\beta)^{\frac{1}{2}}}\big)$ | $\tilde{O}\big(\frac{nL}{(1-\beta)\sigma^2}\big)$ |

$^{\dagger}$ The complete complexity is $O\big(\frac{n^3L}{(1-\beta)^2 \min\{\sigma^2,(1-\beta)^2b^2\}}\big)$, which reduces to $O\big(\frac{n^3L}{(1-\beta)^2\sigma^2}\big)$ for small $\sigma^2$.

Table 7: Rate comparison between different algorithms in smooth and non-convex stochastic decentralized optimization under the PL condition. Quantity $\mu$ is the PL constant.

| | References | Gossip matrix | Convergence rate | Tran. iters. |
|---|---|---|---|---|
| LB | Theorem 3 | $\beta \in [0, \cos(\pi/n)]$ | $\Omega\big(\frac{\sigma^2}{\mu nT} + \frac{\mu}{L}e^{-T\sqrt{\mu(1-\beta)/L}}\big)$ | $\tilde{O}\big(\frac{(L/\mu)^{1/2}}{(1-\beta)^{1/2}}\big)$ |
| UB | DSGD [30]$^{\dagger}$ | $\beta \in [0, 1)$ | $\tilde{O}\big(\frac{\sigma^2}{\mu nT} + \frac{L\sigma^2}{\mu^2T^2(1-\beta)} + \frac{Lb^2}{\mu^2T^2(1-\beta)^2}\big)$ | $\tilde{O}\big(\frac{nL/\mu}{(1-\beta)\min\{1,(1-\beta)b^2\}}\big)$ |
| | DAGD[57]$^{\dagger\ddagger}$ | $\beta \in [0, 1)$ | $\tilde{O}\big(\frac{\sigma^2}{\mu nT} + e^{-T\sqrt{\mu(1-\beta)/L}}\big)$ | $\tilde{O}\big(\frac{(L/\mu)^{1/2}}{(1-\beta)^{1/2}}\big)$ |
| | D$^2$/ED [77]$^{\dagger}$ | $\beta \in [0, 1)$ | $\tilde{O}\big(\frac{\sigma^2}{\mu nT} + \frac{L\sigma^2}{\mu^2T^2(1-\beta)} + e^{-\mu T(1-\beta)/L}\big)$ | $\tilde{O}\big(\frac{nL/\mu}{1-\beta}\big)$ |
| | DSGT [2] | $\beta \in [0, 1)$ | $\tilde{O}\big(\frac{L\sigma^2}{\mu^2 nT} + \frac{L^2\sigma^2}{\mu^3T^2(1-\beta)} + e^{-\mu T(1-\beta)/L}\big)$ | $\tilde{O}\big(\frac{nL/\mu}{1-\beta}\big)$ |
| | DSGT [73] | $\beta \in [0, 1)$ | $\tilde{O}\big(\frac{L\sigma^2}{\mu^2 nT} + \frac{L^2\sigma^2}{\mu^3T^2(1-\beta)^3} + e^{-\mu T(1-\beta)/L}\big)$ | $\tilde{O}\big(\frac{nL/\mu}{(1-\beta)^3}\big)$ |
| | MG-DSGD | $\beta \in [0, 1)$ | $\tilde{O}\big(\frac{L\sigma^2}{\mu^2 nT} + e^{-T\mu(1-\beta)^{\frac{1}{2}}/L}\big)$ | $\tilde{O}\big(\frac{L/\mu}{(1-\beta)^{1/2}}\big)$ |

$^{\dagger}$ These rates are derived under the strongly-convex assumption, not the general PL condition.

$^{\ddagger}$ This rate is achieved by utilizing increasing (non-constant) mini-batch sizes.

## C  Ring-Lattice graph

For any $2 \leq k < n - 1$ and $k$ is even, the ring-lattice graph $\mathcal{R}_{n,k}$ has several preferable properties. The following lemma estimates the order of the diameter of $\mathcal{R}_{n,k}$.

**Lemma 1 (Formal version)** *For any two nodes $1 \leq i < j \leq n$, the distance between $i$ and $j$ is $\lceil\frac{2\min\{j-i,i+n-j\}}{k}\rceil$. In particularly, the diameter of $\mathcal{R}_{n,k}$ is $D_{n,k} = \lceil\frac{2\lfloor n/2\rfloor}{k}\rceil = \Theta(\frac{n}{k})$.*

*Proof.* By symmetry of $\mathcal{R}_{n,k}$, it suffices to consider the case that $j - i \leq i + n - j$. On the one side, the path $\{i, k/2 + i, k + i, \ldots, (\lceil 2(j-i)/k\rceil - 1)k/2 + i, j\}$ connects the pair $(i, j)$ with length $\lceil 2(j-i)/k\rceil$, which leads to $\text{dist}(i, j) \leq \lceil 2(j-i)/k\rceil$. On the other side, let $\{i_0 \triangleq$

$i, i_1, \ldots, i_{m-1}, i_m \triangleq j\}$ be one of the shortest paths connecting $(i, j)$. Without loss of generality, we assume $i < i_1 < \cdots < i_{m-1} < j$. Since $|i_r - i_{r-1}| \leq k/2$ for any $r = 1, \ldots, m$ by the definition of $\mathcal{R}_{n,k}$, it holds that

$$\text{dist}(i, j) = m \geq \frac{|i_m - i_0|}{k/2} = \frac{2(j - i)}{k}.$$

Since $\text{dist}(i, j)$ is a integer, we reach $\text{dist}(i, j) \geq \lceil \frac{2(j-i)}{k} \rceil$. The diameter is readily obtained by maximizing the expression of $\text{dist}(i, j)$ with respect $i$ and $j$. $\qquad \square$

The following lemma clarifies the Laplacian matrix associated with $\mathcal{R}_{n,k}$ and its eigenvalues.

**Lemma 2.** *The Laplacian matrix of $\mathcal{R}_{n,k}$ is given by $L_{n,k} = kI - A_{n,k}$ with adjacency matrix $A_{n,k} = \sum_{\ell=1}^{k/2}(J^\ell + (J^T)^\ell)$ where*

$$J = \begin{bmatrix} 0 & 1 & 0 & \cdots & 0 \\ 0 & 0 & 1 & \cdots & 0 \\ \vdots & \ddots & \ddots & \ddots & \vdots \\ 0 & 0 & \ddots & 0 & 1 \\ 1 & 0 & \cdots & 0 & 0 \end{bmatrix} \in \mathbb{R}^{n \times n} \quad and \quad J^T = J^{-1} \in \mathbb{R}^{n \times n}.$$

*Moreover, the eigenvalues of the Laplacian matrix $L_{n,k}$ are given by*

$$\mu_j \triangleq k + 1 - \sum_{\ell=-k/2}^{k/2} \cos(\frac{2\pi(j-1)\ell}{n}) \in [0, 2k], \quad \forall j \in \{1 \ldots, n\} \tag{21}$$

*where $\mu_j$ is the $j$-th eigenvalue of $L_{n,k}$. In addition, it holds that*

$$(1 - \pi^2/12)\frac{\pi^2 k(k+1)(k+2)}{6n^2} \leq \min_{2 \leq j \leq n} \mu_j \leq \frac{\pi^2 k(k+1)(k+2)}{6n^2}. \tag{22}$$

*Proof.* (*Laplacian matrix.*) Since every node in $\mathcal{R}_{n,k}$ is of degree $k$, the degree matrix of $\mathcal{R}_{n,k}$ is $kI$. The adjacency matrix $A_{n,k}$ can be easily achieved by following the construction of $\mathcal{R}_{n,k}$.

(*Eigenvalues.*) Let $\omega := e^{2\pi i/n}$ (where $i$ is the imaginary number), then $J$ can be decomposed as $J = U\text{diag}(\omega^0 = 1, \omega, \ldots, \omega^{n-1})U^H$ with a unitary matrix $U = \frac{1}{\sqrt{n}}[\omega^{(p-1)(q-1)}]_{p,q=1}^n$. Since $J^\ell = U\text{diag}(1, \omega^\ell, \ldots, \omega^{\ell(n-1)})U^H$ for all $-k/2 \leq \ell \leq k/2$, and $J^T = J^H = J^{-1} = U\text{diag}(1, \bar{\omega} \ldots, \bar{\omega}^{n-1})U^H$ where $\bar{\omega} = e^{-2\pi i/n} = \omega^{-1}$, we have that

$$I + A_{n,k} = \sum_{\ell=-k/2}^{k/2} J^\ell = U\left(\sum_{\ell=-k/2}^{k/2} \text{diag}(1, \omega^\ell, \ldots, \omega^{\ell(n-1)})\right)U^H$$

$$= U\left(\sum_{\ell=-k/2}^{k/2} \text{diag}(1, \cos(\frac{2\pi\ell}{n}), \ldots, \cos(\frac{2\pi\ell(n-1)}{n}))\right)U^H$$

$$= U\text{diag}\left(k + 1, \sum_{\ell=-k/2}^{k/2} \cos(\frac{2\pi\ell}{n}), \ldots, \sum_{\ell=-k/2}^{k/2} \cos(\frac{2\pi\ell(n-1)}{n})\right)U^H.$$

Therefore, the eigenvalues of $L_{n,k}$ are real numbers $\{\mu_j \triangleq k + 1 - \sum_{\ell=-k/2}^{k/2} \cos(\frac{2\pi(j-1)\ell}{n}) : j = 1, \ldots, n\}$. It is easy to verify that $\mu_j \geq 0$ and

$$\mu_j = k - \sum_{1 \leq |\ell| \leq k/2, \, k \neq 0} \cos(\frac{2\pi(j-1)\ell}{n}) \leq 2k \quad \forall j = 1, \ldots, n.$$

Next we establish the bounds for $\min_{2 \leq j \leq n} \mu_j$. For the upper bound, by the inequality $\cos(\theta) \geq 1 - \theta^2/2$ for any $\theta \in \mathbb{R}$, we have that

$$\min_{2 \leq j \leq n} \mu_j \leq \mu_2 \leq (k+1) - \sum_{\ell=-k/2}^{k/2} (1 - \frac{2\pi^2 \ell^2}{n^2}) = 2\pi^2 \sum_{\ell=-k/2}^{k/2} \frac{\ell^2}{n^2} = \frac{\pi^2 k(k+1)(k+2)}{6n^2}. \tag{23}$$

For the lower bound, by applying Lemma 3 (see below) with $h_k(\theta)$ defined as $\sum_{\ell=-k/2}^{k/2}\cos(2\ell\theta)$ and $\alpha = \frac{\pi}{n}$, we reach

$$\min_{2\leq j\leq n}\mu_j \geq \inf_{\theta\in[\alpha,\pi-\alpha]}(k+1-h_k(\theta))$$

$$\geq k+1-\max\left\{\underbrace{(k+1)\left(1-\frac{k(k+2)\pi^2/n^2}{1+(k+1)^2\pi^2/n^2}\right)^{\frac{1}{2}}}_{\text{I}},\underbrace{h_k(\pi/n)}_{\text{II}}\right\} \qquad (24)$$

For term I, using the inequality $(1-z)^{\frac{1}{2}}\leq 1-\frac{1}{2}z$ for any $z\in[0,1]$, and $k+1\leq n$, we have

$$\text{I} = (k+1)\left(1-\frac{k(k+2)\pi^2/n^2}{1+(k+1)^2\pi^2/n^2}\right)^{\frac{1}{2}}$$

$$\leq (k+1)\left(1-\frac{k(k+2)\pi^2/n^2}{2(1+(k+1)^2\pi^2/n^2)}\right) \leq (k+1)\left(1-\frac{\pi^2 k(k+2)}{2(1+\pi^2)}\right). \qquad (25)$$

For term II, using the inequality $\cos(\theta)\leq 1-\theta^2/2+\theta^4/24 \leq 1-\frac{12-\pi^2}{24}\theta^2$ for all $\theta\in[0,\pi]$, we have

$$\text{II} = h_k(\pi/n) = \sum_{\ell=-k/2}^{k/2}\cos(\frac{2\pi\ell}{n}) \leq \sum_{\ell=-k/2}^{k/2}\left(1-\frac{(12-\pi^2)\pi^2}{6}\frac{\ell^2}{n^2}\right)$$

$$= k+1-\frac{(12-\pi^2)\pi^2}{6}\sum_{\ell=-k/2}^{k/2}\frac{\ell^2}{n^2} = k+1-(1-\pi^2/12)\frac{\pi^2 k(k+1)(k+2)}{6n^2}. \qquad (26)$$

Plugging (25) and (26) into (24), we reach (22). $\qquad\square$

We also establish an auxiliary lemma to facilitate the results in Lemma 2.

**Lemma 3.** *For any $k\geq 1$ and $k$ is even, let $h_k(\theta) = \sum_{\ell=-k/2}^{k/2}\cos(2\ell\theta)$ for any $\theta\in[\alpha,\pi-\alpha]$ with some $\alpha\in(0,\frac{\pi}{k+1}]$. It holds that*

$$\max_{\theta\in[\alpha,\pi-\alpha]}h_k(\theta) \leq \max\left\{(k+1)\sqrt{\frac{1+\alpha^2}{1+(k+1)^2\alpha^2}}, h_k(\alpha)\right\} \qquad (27)$$

*Proof.* Since $|h_k(\pi/2)| = 1$ while $(k+1)\sqrt{\frac{1+\alpha^2}{1+(k+1)^2\alpha^2}} > 1$, we know (27) holds when $\theta = \pi/2$. Since $h_k(\pi-\theta) = h_k(\theta)$ by the definition of $h_k(\theta)$, it suffices to consider $\max_{\theta\in(\alpha,\pi/2)}|h_k(\theta)|$.

For any $\theta\in(\alpha,\pi/2)$, it holds that

$$h_k(\theta) = \sum_{\ell=-k/2}^{k/2}\cos(2\ell\theta) = \frac{1}{2\sin(\theta)}\sum_{\ell=-k/2}^{k/2}2\cos(2\ell\theta)\sin(\theta)$$

$$= \frac{1}{2\sin(\theta)}\sum_{\ell=-k/2}^{k/2}(\sin((2\ell+1)\theta)-\sin((2\ell-1)\theta))$$

$$= \frac{\sin((k+1)\theta)}{\sin(\theta)}.$$

Therefore, we have that

$$h_k'(\theta) = \frac{(k+1)\cos((k+1)\theta)\sin(\theta)-\sin((k+1)\theta)\cos(\theta)}{\sin^2(\theta)}$$

$$= \frac{\cos((k+1)\theta)\cos(\theta)}{\sin^2(\theta)}((k+1)\tan(\theta)-\tan((k+1)\theta)). \quad \text{(if } \cos((k+1)\theta)\cos(\theta)\neq 0)$$

The special case that $\cos((k+1)\theta)\cos(\theta) = 0$ and $h_k'(\theta) = 0$ leads to that $\cos(\theta) = 0$, which is impossible for $z \in (\alpha, \pi/2)$. Therefore, all the extrema of $h_k$ on $(\alpha, \pi/2)$ must satisfy

$$\tan((k+1)\theta) = (k+1)\tan(\theta). \tag{28}$$

By combining the cases of endpoints, i.e., $\theta = \alpha$ and $\theta = \pi/2$, with (28) for the interior extrema, we have that

$$\max_{\theta \in [\alpha, \pi/2]} h_k(\theta) = \max\{\sup_{\substack{\theta \in (\alpha, \pi/2) \\ (28)\text{ holds}}} h_k(\theta), h_k(\alpha), h_k(\pi/2)\}$$

$$\leq \max\{\sup_{\substack{\theta \in (\alpha, \pi/2) \\ (28)\text{ holds}}} \sqrt{h_k(\theta)^2}, h_k(\alpha), 1\}$$

$$= \max\{\sup_{\substack{\theta \in (\alpha, \pi/2) \\ (28)\text{ holds}}} \left(\frac{\frac{\tan((k+1)\theta)^2}{1+\tan((k+1)\theta)^2}}{\frac{\tan(\theta)^2}{1+\tan(\theta)^2}}\right)^{\frac{1}{2}}, h_k(\alpha)^2, 1\}$$

$$\leq \max\{\sup_{\substack{\theta \in (\alpha, \pi/2) \\ (28)\text{ holds}}} \left(\frac{\frac{(k+1)^2\tan(\theta)^2}{1+(k+1)^2\tan(\theta)^2}}{\frac{\tan(\theta)^2}{1+\tan(\theta)^2}}\right)^{\frac{1}{2}}, h_k(\alpha), 1\}$$

$$\leq \max\{(k+1)\sup_{\theta \in (\alpha, \pi/2)} \left(\frac{1+\tan(\theta)^2}{1+(k+1)^2\tan(\theta)^2}\right)^{\frac{1}{2}}, h_k(\alpha)\}$$

$$\leq \max\{(k+1)\left(\frac{1+\alpha^2}{1+(k+1)^2\alpha^2}\right)^{\frac{1}{2}}, h_k(\alpha)\},$$

where the last inequality is because the function $\frac{1+z}{1+(k+1)^2 z}$ is decreasing with respect to $z \in [0, +\infty)$ and $\tan(\theta) \geq \tan(\alpha) \geq \alpha$. $\qquad\square$

**Theorem 1** *Given a fixed $n$ and any $\beta \in [0, \cos(\pi/n)]$, there exists a ring-lattice graph with an associated weight matrix $W$ such that*

*(i)* $W \in \mathcal{W}_{n,\beta}$ *, i.e.,* $W \in \mathbb{R}^{n \times n}$ *and* $\|W - \frac{1}{n}\mathbb{1}_n\mathbb{1}_n^T\| = \beta$;

*(ii) its diameter $D$ satisfies* $D = \Theta(1/\sqrt{1-\beta})$.

*Proof.* We prove this theorem in two cases:

*(Case 1: $0 \leq \beta \leq \min\{\cos(\pi/9), \cos(\pi/n)\}$.)* In this case, we consider the complete graph whose diameter $D = 1$, and let $W = \frac{1-\beta}{n}\mathbb{1}_n\mathbb{1}_n^T + \beta I$, then it holds that $\|W - \frac{1}{n}\mathbb{1}_n\mathbb{1}_n^T\|_2 = \beta$. Furthermore, since $1 \geq 1 - \beta \geq 1 - \cos(\pi/9) = \Omega(1)$, we naturally have $1 = D = \Omega((1-\beta)^{-\frac{1}{2}})$. Note that the complete graph can also be regarded as a special ring-lattice graph $\mathcal{R}_{n,n-1}$.

*(Case 2: $\cos(\pi/9) < \beta \leq \cos(\pi/n)$.)* Note that this case requires $n \geq 10$. Letting

$$k = 2\left\lceil n\sqrt{\frac{3(1-\beta)}{\pi^2(1-\pi^2/12)}}\right\rceil \geq \max\left\{2n\sqrt{\frac{3(1-\beta)}{\pi^2(1-\pi^2/12)}}, 2\right\}, \tag{29}$$

we consider the ring-lattice graph $\mathcal{R}_{n,k}$ with the weight matrix

$$W = I - \frac{1-\beta}{\min_{2\leq j\leq n}\mu_j}L_{n,k} \tag{30}$$

where $k$ is defined as in (29) and $\{\mu_j : 2 \leq j \leq n\}$ are given in Lemma 2. For such ring-lattice graph and its associated weight matrix defined in (30), we prove $D = \Omega((1-\beta)^{-\frac{1}{2}})$ in following steps:

- We first check whether $\mathcal{R}_{n,k}$ is well-defined, i.e., $2 \leq k < n - 1$ and $k$ is even. It is easy to observe that $k$ is even by (29). Furthermore, since $\cos(\pi/9) < \beta$ and $n \geq 10$, we have

$$k < 2 \left( n\sqrt{\frac{3(1-\beta)}{\pi^2(1-\pi^2/12)}} + 1 \right) < 2 \left( n\sqrt{\frac{3(1-\cos(\pi/9))}{\pi^2(1-\pi^2/12)}} + 1 \right) < 0.7n + 2 \leq n - 1. \tag{31}$$

- We next check whether $W_{n,k} \in \mathcal{W}_{n,\beta}$, i.e., $\|W - \frac{1}{n}\mathbb{1}_n\mathbb{1}_n^T\|_2 = \beta$. By Lemma 2, we have

$$\frac{1-\beta}{\min_{2 \leq j \leq n}\mu_j} \leq \frac{6(1-\beta)n^2}{(1-\pi^2/12)\pi^2 k(k+1)(k+2)} < \frac{1}{2k}\frac{12(1-\beta)n^2}{(1-\pi^2/12)\pi^2 k^2} \leq \frac{1}{2k}.$$

Since all eigenvalues of $L_{n,k}$ lie in $[0, 2k]$ (see (21)), it holds that $W$ is positive semi-definite (see definition (30)). Moreover,

$$\|W - \frac{1}{n}\mathbb{1}_n\mathbb{1}_n^T\|_2 = \max_{2 \leq \ell \leq n} \left\{ 1 - \frac{1-\beta}{\min\{\mu_j : 2 \leq j \leq n\}}\mu_\ell \right\} = \beta \quad i.e., \quad W \in \mathcal{W}_{n,\beta}.$$

- Finally, we check whether $\Theta(\frac{n}{k}) = D = \Theta((1-\beta)^{-\frac{1}{2}})$. By using the inequality $\cos(\theta) \leq 1 - \theta^2/2 + \theta^4/24$ with $\theta = \pi/n$, we have

$$\beta \leq \cos(\pi/n) \leq 1 - \pi^2/2n^2 + \pi^4/24n^4 \leq 1 - \pi^2/3n^2.$$

Therefore, we have that $n\sqrt{\frac{3(1-\beta)}{\pi^2(1-\pi^2/12)}} \geq n\sqrt{\frac{\pi^2}{n^2(1-\pi^2/12)}} \geq 2$ and hence

$$k < 2 \left( n\sqrt{\frac{3(1-\beta)}{\pi^2(1-\pi^2/12)}} + 1 \right) \leq 3n\sqrt{\frac{3(1-\beta)}{\pi^2(1-\pi^2/12)}}. \tag{32}$$

Combining (29) and (32), we achieve

$$\frac{n}{k} = \Theta((1-\beta)^{-\frac{1}{2}}).$$

This fact together with Lemma 1 leads to $D = \Theta(1/\sqrt{1-\beta})$.

$\square$

## D   Lower bounds

### D.1   Non-convex Case: Proof of Theorem 2

In this subsection, we establish the lower bound of the smooth and non-convex decentralized stochastic optimization. Our analysis builds upon [44] but utilizes the proposed ring-lattice graphs for the construction of worst-case instances, which significantly broadens the scope of weight matrices that the lower bound can apply to, i.e., from $\beta = \cos(\pi/n)$ in [44] to any $\beta \in [0, \cos(\pi/n)]$. Specifically, we will prove for any $\beta \in [0, \cos(\pi/n)]$ and $T = \Omega(1/\sqrt{1-\beta})$,

$$\inf_{A \in \mathcal{A}_W} \sup_{W \in \mathcal{W}_{n,\beta}} \sup_{\{\tilde{g}_i\}_{i=1}^n \subseteq \mathcal{O}_\sigma^2} \sup_{\{f_i\}_{i=1}^n \subseteq \mathcal{F}_L} \mathbb{E}\|\nabla f(\hat{x}_{A, \{f_i\}_{i=1}^n, \{\tilde{g}_i\}_{i=1}^n, W, T})\|^2$$

$$= \Omega \left( \frac{\sqrt{\Delta L}\sigma}{\sqrt{nT}} + \frac{\Delta L}{T\sqrt{1-\beta}} \right). \tag{33}$$

where $\Delta := \mathbb{E}[f(x^{(0)})] - \min_x f(x)$.

We establish the two terms in (33) separately as in [44] by constructing two hard-to-optimize instances. We denote the $j$-th coordinate of a vector $x \in \mathbb{R}^d$ by $[x]_j$ for $j = 1, \ldots, d$, and let $\text{prog}(x)$ be

$$\text{prog}(x) := \begin{cases} 0 & \text{if } x = 0; \\ \max_{1 \leq j \leq d}\{j : [x]_j \neq 0\} & \text{otherwise}. \end{cases}$$

Similarly, for a set of multiple points $\mathcal{X} = \{x_1, x_2, \dots\}$, we define $\text{prog}(\mathcal{X}) := \max_{x \in \mathcal{X}} \text{prog}(x)$. As described in [9, 4], a function $f$ is called zero-chain if it satisfies

$$\text{prog}(\nabla f(x)) \le \text{prog}(x) + 1, \quad \forall\, x \in \mathbb{R}^d,$$

which implies that, starting from $x = 0$, a single gradient evaluation can only earn at most one more non-zero coordinate for the model parameters. We next introduce a key zero-chain function to facilitate the analysis.

**Lemma 4** (Lemma 2 of [4]). *Let function*

$$\ell(x) := -\Psi(1)\Phi([x]_1) + \sum_{j=1}^{d-1} \Big( \Psi(-[x]_j)\Phi(-[x]_{j+1}) - \Psi([x]_j)\Phi([x]_{j+1}) \Big)$$

*where for $\forall\, z \in \mathbb{R}$,*

$$\Psi(z) = \begin{cases} 0 & z \le 1/2; \\ \exp\left(1 - \frac{1}{(2z-1)^2}\right) & z > 1/2, \end{cases} \quad \Phi(z) = \sqrt{e} \int_{-\infty}^{z} e^{-\frac{1}{2}t^2}\, \mathrm{d}t.$$

*Then $\ell$ satisfy several properties as below:*

1. *$\ell(x) - \inf_x \ell(x) \le \Delta_0 d$, $\forall\, x \in \mathbb{R}^d$ with $\Delta_0 = 12$.*

2. *$\ell$ is $L_0$-smooth with $L_0 = 152$.*

3. *$\|\nabla \ell(x)\|_\infty \le G_0$, $\forall\, x \in \mathbb{R}^d$ with $G_0 = 23$.*

4. *$\|\nabla \ell(x)\|_\infty \ge 1$ for any $x \in \mathbb{R}^d$ with $[x]_d = 0$.*

We next establish the two terms in (33) by constructing two hard-to-optimize instances.

**Instance 1.** The proof for the first term $\Omega((\frac{\Delta L \sigma^2}{nT})^{\frac{1}{2}})$ essentially follows [44]. We provide the proof for the sake of being self-contained.

(Step 1.) Let all $f_i$ be $L\lambda^2 \ell(x/\lambda)/L_0$ where $\ell$ is defined in Lemma 4 and $\lambda$ is to be specified, and thus $f = L\lambda^2 \ell(x/\lambda)/L_0$. Since $\nabla^2 f_i = L\nabla^2 \ell/L_0$ and $h$ is $L_0$-smooth (Lemma 4), we know $f_i$ is $L$-smooth for any $\lambda > 0$. By Lemma 4, we have

$$f(0) - \inf_x f(x) = \frac{L\lambda^2}{L_0}(\ell(0) - \inf_x \ell(x)) \le \frac{L\lambda^2 \Delta_0 d}{L_0}.$$

Therefore, to ensure $f_i \in \mathcal{F}_L$ for all $1 \le i \le n$ and $f(0) - \inf_x f(x) \le \Delta$, it suffices to let

$$\frac{L\lambda^2 \Delta_0 d}{L_0} \le \Delta, \quad \text{i.e.,} \quad d\lambda^2 \le \frac{L_0 \Delta}{L \Delta_0}. \tag{34}$$

(Step 2.) We construct the stochastic gradient oracle $\tilde{g}_i \; \forall\, i = 1, \dots, n$ as the follows:

$$[\tilde{g}_i(x)]_j = [\nabla f_i(x)]_j \left(1 + \mathbb{1}\{j > \text{prog}(x)\}\left(\frac{Z}{p} - 1\right)\right), \forall\, x \in \mathbb{R}^d, \; j = 1, \dots, d$$

with $Z \sim \text{Bernoulli}(p)$, and $p \in (0, 1)$ to be specified. The oracle $\tilde{g}_i(x)$ has probability $p$ to zero $[\nabla f_i(x)]_{\text{prog}(x)+1}$. It is easy to see $\tilde{g}_i$ is unbiased, i.e., $\mathbb{E}[\tilde{g}_i(x)] = \nabla f_i(x)$ for all $x \in \mathbb{R}^d$. Moreover, since $f_i$s are zero-chain, we have $\text{prog}(\tilde{g}_i(x)) \le \text{prog}(\nabla f_i(x)) \le \text{prog}(x) + 1$ and hence

$$\mathbb{E}[\|\tilde{g}_i(x) - \nabla f_i(x)\|^2] = |[\nabla f_i(x)]_{\text{prog}(x)+1}|^2 \mathbb{E}\left[\left(\frac{Z}{p} - 1\right)^2\right] = |[\nabla f_i(x)]_{\text{prog}(x)+1}|^2 \frac{1-p}{p}$$

$$\le \|\nabla f_i(x)\|_\infty^2 \frac{1-p}{p} \le \frac{L^2\lambda^2(1-p)}{L_0^2 p} \|\nabla \ell(x)\|_\infty^2$$

$$\overset{\text{Lemma 4}}{\le} \frac{L^2\lambda^2(1-p)G_0^2}{L_0^2 p}.$$

Therefore, to ensure $\tilde{g}_i \in \mathcal{O}_{\sigma^2}$ for all $1 \le i \le n$, it suffices to let

$$p = \min\{\frac{L^2\lambda^2 G_0^2}{L_0^2\sigma^2}, 1\}. \tag{35}$$

We let $W = (1-\beta)\frac{1}{n}\mathbb{1}_n\mathbb{1}_n^T + \beta I$, then obviously $W \in \mathcal{W}_{n,\beta}$.

(Step 3.) Next we show the error $\mathbb{E}[\|\nabla f(x)\|^2]$ is lower bounded by $\Omega((\frac{\Delta L\sigma^2}{nT})^{\frac{1}{2}})$, with any algorithm $A \in \mathcal{A}_W$. Let $x_i^{(t)}$, $\forall t = 0, \ldots$ and $1 \le i \le n$, be the $t$-th query point of node $i$. Let $\text{prog}^{(t)} = \max_{1\le i\le n, 0\le s<t} \text{prog}(x_i^{(s)})$. By Lemma 2 of [44], we have

$$\mathbb{P}(\text{prog}^{(T)} \ge d) \le e^{(e-1)npT-d}. \tag{36}$$

On the other hand, when $\text{prog}^{(T)} < d$, by the fourth point in Lemma 4, it holds that

$$\min_{\hat{x}\in\text{span}\{\{x_i^{(t)}\}_{1\le i\le n, 0\le t<T}\}} \|\nabla f(\hat{x})\| \ge \min_{[\hat{x}]_d=0} \|\nabla f(\hat{x})\| = \frac{L\lambda}{L_0}\min_{[\hat{x}]_d=0}\|\nabla\ell(\hat{x})\| \ge \frac{L\lambda}{L_0}. \tag{37}$$

Therefore, by combining (36) and (37), we have

$$\mathbb{E}[\|\nabla f(\hat{x})\|^2] \ge (1 - e^{(e-1)npT-d})\frac{L^2\lambda^2}{L_0^2}. \tag{38}$$

Let

$$\lambda = \frac{L_0}{L}\left(\frac{\Delta L\sigma^2}{3nTL_0\Delta_0 G_0^2}\right)^{\frac{1}{4}} \quad \text{and} \quad d = \left\lfloor\left(\frac{3L\Delta nTG_0^2}{\sigma^2 L_0\Delta_0}\right)^{\frac{1}{2}}\right\rfloor. \tag{39}$$

Then (34) naturally holds and $p = \min\{\frac{G_0^2}{\sigma^2}\left(\frac{\Delta L\sigma^2}{3nTL_0\Delta_0 G_0^2}\right)^{\frac{1}{2}}, 1\}$ by plugging (39) into (35). Without loss of generality, we assume $T$ is sufficiently large such that $d \ge 2$. Then, using the definition of $p$, we have that

$$(e-1)npT - d \le (e-1)nT\frac{G_0^2}{\sigma^2}\left(\frac{\Delta L\sigma^2}{3nTL_0\Delta_0 G_0^2}\right)^{\frac{1}{2}} - d$$

$$= \frac{e-1}{3}\left(\frac{3L\Delta nTG_0^2}{\sigma^2 L_0\Delta_0}\right)^{\frac{1}{2}} - d < \frac{e-1}{3}(d+1) - d \le 1 - e < 0$$

which, combined with (38), further implies

$$\mathbb{E}[\|\nabla f(\hat{x})\|^2] = \Omega\left(\frac{L^2\lambda^2}{L_0^2}\right) = \Omega\left(\left(\frac{\Delta L\sigma^2}{3nTL_0\Delta_0 G_0^2}\right)^{\frac{1}{2}}\right) = \Omega\left(\left(\frac{\Delta L\sigma^2}{nT}\right)^{\frac{1}{2}}\right).$$

**Instance 2.** The proof for the second term $\Omega(\frac{\Delta L}{T\sqrt{1-\beta}})$ utilizes weight matrices defined on the ring-lattice graphs described in Theorem 1.

(Step 1.) Let functions

$$\ell_1(x) := -\frac{n}{\lceil n/3\rceil}\Psi(1)\Phi([x]_1) + \frac{n}{\lceil n/3\rceil}\sum_{j\text{ even, }0<j<d}\left(\Psi(-[x]_j)\Phi(-[x]_{j+1}) - \Psi([x]_j)\Phi([x]_{j+1})\right)$$

and

$$\ell_2(x) := \frac{n}{\lceil n/3\rceil}\sum_{j\text{ odd, }0<j<d}\left(\Psi(-[x]_j)\Phi(-[x]_{j+1}) - \Psi([x]_j)\Phi([x]_{j+1})\right).$$

Compared to the $\ell$ function in instance 1, the $\ell_1$ and $\ell_2$ defined here are $3L_0$-smooth. Furthermore, let

$$f_i = \begin{cases} L\lambda^2\ell_1(x/\lambda)/(3L_0) & \text{if } i \in E_1 \triangleq \{j : 1 \le j \le \lceil\frac{n}{3}\rceil\}, \\ L\lambda^2\ell_2(x/\lambda)/(3L_0) & \text{if } i \in E_2 \triangleq \{j : \lfloor\frac{n}{2}\rfloor + 1 \le j \le \lfloor\frac{n}{2}\rfloor + \lceil\frac{n}{3}\rceil\}, \\ 0 & \text{else.} \end{cases}$$

where $\lambda > 0$ is to be specified. To ensure $f_i \in \mathcal{F}_L$ for all $1 \le i \le n$ and $f(0) - \inf_x f(x) \le \Delta$, it suffices to let

$$\frac{L\lambda^2 \Delta_0 d}{3L_0} \le \Delta, \quad \text{i.e.,} \quad d\lambda^2 \le \frac{3L_0 \Delta}{L\Delta_0}. \tag{40}$$

With the functions defined above, we have $f(x) = \frac{1}{n}\sum_{i=1}^n f_i(x) = L\lambda^2 \ell(x/\lambda)/(3L_0)$ and

$$\text{prog}(\nabla f_i(x)) \begin{cases} = \text{prog}(x) + 1 & \text{if } \{\text{prog}(x) \text{ is even and } i \in E_1\} \cup \{\text{prog}(x) \text{ is odd and } i \in E_2\} \\ \le \text{prog}(x) & \text{otherwise.} \end{cases}$$

Therefore, to make progress (i.e., to increase $\text{prog}(x)$), for any gossip algorithm $A \in \mathcal{A}_W$, one must take the gossip communication protocol to transmit information between $E_1$ to $E_2$ alternatively. Namely, it takes at least $\text{dist}(E_1, E_2)$ rounds of gossip communications for any possible gossip algorithm $A$ to increase $\text{prog}(\hat{x})$ by 1. Therefore, we have

$$\text{prog}^{(T)} = \max_{1 \le i \le n,\, 0 \le t < T} \text{prog}(x_i^{(t)}) \le \left\lfloor \frac{T}{\text{dist}(E_1, E_2)} \right\rfloor + 1, \quad \forall T \ge 0. \tag{41}$$

(Step 2.) We consider a gradient oracle that return lossless full-batch gradients, i.e., $\tilde{g}_i = \nabla f_i(x)$, $\forall x \in \mathbb{R}^d$, $1 \le i \le n$. For the construction of weight matrix, we consider the ring-lattice graph with diameter $D$ in Theorem 1 and its associated weight matrix $W$ such that $W \in \mathcal{W}_{n,\beta}$. Then by Lemma 1 and Theorem 1, we have $\text{dist}(E_1, E_2) = \text{dist}(\lceil n/3 \rceil, \lfloor n/2 \rfloor + 1) = \Theta(D) = \Theta(1/\sqrt{1-\beta})$. Suppose $\text{dist}(E_1, E_2) \ge 1/(C\sqrt{1-\beta})$ with some absolute constant $C$, then by (41), we have

$$\text{prog}^{(T)} = \max_{1 \le i \le n,\, 0 \le s \le T} \text{prog}(x_i^{(s)}) \le \left\lfloor C\sqrt{1-\beta}T \right\rfloor + 1, \quad \forall T \ge 0. \tag{42}$$

(Step 3.) We finally show the error $\mathbb{E}[\|\nabla f(x)\|^2]$ is lower bounded by $\Omega\left(\frac{\Delta L}{\sqrt{1-\beta}T}\right)$, with any algorithm $A \in \mathcal{A}_W$. For any $T \ge 1/(C\sqrt{1-\beta}) = \Omega(1/\sqrt{1-\beta})$, consider

$$d = \left\lfloor C\sqrt{1-\beta}T \right\rfloor + 2 < 4C\sqrt{1-\beta}T$$

and

$$\lambda = \left( \frac{3L_0 \Delta}{4L\Delta_0 C\sqrt{1-\beta}T} \right)^{\frac{1}{2}}. \tag{43}$$

Then (40) naturally holds. Since $\text{prog}^{(T)} < d$ by (42), following (37) and using (43), we have

$$\mathbb{E}[\|\nabla f(\hat{x})\|^2] \ge \min_{[\hat{x}]_d = 0} \|\nabla f(\hat{x})\|^2 \ge \frac{L^2 \lambda^2}{9L_0^2} = \Omega\left( \frac{\Delta L}{\sqrt{1-\beta}T} \right).$$

### D.2  Non-convex Case with PL Condition: Proof of Theorem 3

In this subsection, we provide the proof for smooth and non-convex decentralized stochastic optimization under the PL condition: for any $\beta \in [0, \cos(\pi/n)]$ and $T = \Omega(1/\sqrt{1-\beta})$,

$$\inf_{A \in \mathcal{A}_W} \sup_{W \in \mathcal{W}_{n,\beta}} \sup_{\{\tilde{g}_i\}_{i=1}^n \subseteq \mathcal{O}_\sigma^2} \sup_{\{f_i\}_{i=1}^n \subseteq \mathcal{F}_{L,\mu}} \mathbb{E}[f(\hat{x}_{A,\{f_i\}_{i=1}^n,\{\tilde{g}_i\}_{i=1}^n,W,T}) - f^\star]$$

$$= \Omega\left( \frac{\sigma^2}{\mu nT} + \frac{\mu \Delta}{L} \exp(-\sqrt{\mu/L}\sqrt{1-\beta}T) \right). \tag{44}$$

where $\Delta := \mathbb{E}[f(x^{(0)})] - \min_x f(x)$. We still prove the two terms in (44) separately.

**Instance 1.** Our proof for first term $\Omega(\frac{\sigma^2}{\mu nT})$ is inspired by [56], who study the lower bounds in the stochastic but single-node regime.

We consider all functions $f_i = f$ are homogeneous and $W = \frac{1-\beta}{n}\mathbb{1}_n\mathbb{1}_n^T + \beta I \in \mathcal{W}_{n\beta}$. We then choose two functions $f^1, f^{-1} \in \mathcal{F}_{L,\mu}$ which are close enough to each other so that $f^1$ and $f^{-1}$ are hard to distinguish within $T$ gradient queries on each node. The indistinguishability between $f^1$ and $f^{-1}$ follows the standard Le Cam's method in hypothesis testing. Next, we carefully show that

the indistinguishability between $f^1$ and $f^{-1}$ can be properly translated into the lower bound of the algorithmic performance.

Recall that $[x]_j$ denotes the $j$-th coordinate of vector $x \in \mathbb{R}^d$. Let

$$f^v = \frac{1}{2}\left(\mu([x]_1 - v\lambda)^2 + L\sum_{j=2}^{d}[x]_j^2\right),$$

where $v \in \{\pm 1\}$, $d$ can be any integer greater than 2, and $\lambda > 0$ is a quantity to be determined. Clearly, $f^v$ is $L$-smooth and $\mu$-strongly convex, and thus satisfies the $\mu$-PL condition. The optimum $x^{v,\star}$ of $f^v$ is $v\lambda e_1$ with function value $\min_{x \in \mathbb{R}^d} f^v(x) = 0$, where $e_1$ is the first canonical vector.

We construct the gradient oracle with $\mathcal{N}(0, \sigma^2)$ noise. Specifically, on query at point $x$ on node $i$, the oracle returns $\tilde{g}_i(x) = \nabla f_i(x) + s_i$ with independent noise $s_i \sim \mathcal{N}(0, \sigma^2)$. Obviously $\tilde{g}_i \in \mathcal{O}_{\sigma^2}$ for all $1 \le i \le n$.

Let $\mathbf{S}^{(T)} := \{(\mathbf{x}^{(t)} \triangleq (x_1^{(t)}, \dots, x_n^{(t)}), \tilde{\mathbf{g}}(\mathbf{x}^{(t)}) \triangleq (\tilde{g}_1(x_1^{(t)}), \dots, \tilde{g}_n(x_n^{(t)})))\}_{t=0}^{T-1}$ be set of variables corresponding to the sequence of $T$ queries on all nodes. Let $P^{v,T} := P(\mathbf{S}^{(T)} \mid f^v)$ be the joint distribution of $\mathbf{S}^{(T)}$ if the underlying function was $f^v$, i.e., $f_1 = \dots = f_n = f^v$.

**Lemma 5.** *Let $V \sim \mathrm{Unif}(\{\pm 1\})$, then for any optimization procedure $\hat{x}$ based on the observed gradients, it holds that*

$$\mathbb{E}_V[f^V(\hat{x})] \ge \mu\lambda^2/2 \inf_{\hat{V}} \mathbb{P}(\hat{V} \ne V) \tag{45}$$

*where the randomness is over the random index $V$ and the observed data $\mathbf{S}^{(T)}$ and the infimum is taken over all testing procedures $\hat{V}$ based on the observed queries.*

*Proof.* Since $P^{v,T}$ is probability of the observed queries conditioned on $\{V = v\}$, by Markov's inequality, we have

$$\mathbb{E}_V[f^V(\hat{x})] \ge \mu\lambda^2/2\, \mathbb{E}_V[P^{V,T}(f^V(\hat{x}) \ge \mu\lambda^2/2)]. \tag{46}$$

Now, we define the test $\hat{V}_{\mathrm{op}}$ based on the optimization procedure $\hat{x}$ as follows:

$$\hat{V}_{\mathrm{op}} = \begin{cases} v & \text{if } f^v(\hat{x}) < \mu\lambda^2/2 \\ \text{randomly pick one from } \{\pm 1\} & \text{otherwise.} \end{cases},$$

By the definition of $f^v$, at most one of $\{f^1(\hat{x}), f^{-1}(\hat{x})\}$ is strictly below $\mu\lambda^2/2$, so $\hat{V}_{\mathrm{op}}$ is well-defined. By the definition of $\hat{V}_{\mathrm{op}}$, $\{\hat{V}_{\mathrm{op}} \ne v\}$ implies $\{f^v(\hat{x}) \ge \mu\lambda^2/2\}$. Thus we have

$$P^{v,T}(f^v(\hat{x}) \ge \mu\lambda^2/2) \ge P^{V,T}(\hat{V}_{\mathrm{op}} \ne v) \ge \inf_{\hat{V}} \mathbb{P}(\hat{V} \ne v) \quad \text{for any } v \in \{\pm 1\} \tag{47}$$

Plugging (47) into (46), we reach the conclusion. $\qquad\square$

Based on Lemma 5, we know that the worst-case optimization performance in the above scenario can be lower bounded by the indistinguishabiligty between to hypothesises $f^1$ and $f^{-1}$. Recall the following standard result of Le Cam [8]

$$\inf_{\hat{V}}\left(P^{1,T}(\hat{V} \ne 1) + P^{-1,T}(\hat{V} \ne -1)\right) = 1 - \|P^{1,T} - P^{-1,T}\|_{TV} \tag{48}$$

where $\|P^{1,T} - P^{-1,T}\|_{TV}$ is the total variation distance between the two distributions. We can precisely measure the indistinguishabiligty between $f^1$ and $f^{-1}$ by $\|P^{1,T} - P^{-1,T}\|_{TV}$. Since the total variation distance is hard to compute, we turn to compute the KL-divergence, which is a relaxation of the total variation distance due to Pinsker's inequality [14]:

$$\|P^{1,T} - P^{-1,T}\|_{TV}^2 \le \frac{1}{2}\mathrm{D}_{\mathrm{KL}}(P^{1,T}\|P^{-1,T}). \tag{49}$$

**Lemma 6.** *The KL-divergence between $P^{1,T}$ and $P^{-1,T}$ is upper bounded by:*

$$\mathrm{D}_{\mathrm{KL}}(P^{1,T}\|P^{-1,T}) \le 2nT\mu^2\lambda^2/\sigma^2. \tag{50}$$

*Proof.* Since the observed data obey Markov's property, i.e., $\tilde{g}_i(x_i^{(t)}) \perp \mathbf{S}^{(t)} \mid x_i^{(t)}$ for any $1 \leq i \leq n$ and $0 \leq t < T$, we have

$$\mathrm{D}_{\mathrm{KL}}(P^{1,T}||P^{-1,T}) = \mathbb{E}_{P^{1,T}}\left[\log\left(P^{1,T}/P^{-1,T}\right)\right]$$

$$=\mathbb{E}_{P^{1,T}}\left[\log\left(\prod_{t=0}^{T-1}\mathbb{P}(\tilde{\mathbf{g}}(\mathbf{x}^{(t)}) \mid \mathbf{x}^{(t)}, f^1) / \prod_{t=0}^{T-1}\mathbb{P}(\tilde{\mathbf{g}}(\mathbf{x}^{(t)}) \mid \mathbf{x}^{(t)}, f^{-1})\right)\right]$$

$$=\sum_{t=0}^{T-1}\mathbb{E}_{\mathbf{S}^{(t)}|f^1}\left[\mathbb{E}_{\tilde{\mathbf{g}}(\mathbf{x}^{(t)})\sim\mathcal{N}(\nabla f^1(\boldsymbol{x}),\sigma^2)}\left[\log\left(\mathbb{P}(\mathbf{g}(\mathbf{x}^{(t)}) \mid \mathbf{x}^{(t)}, f^1)/\mathbb{P}(\mathbf{g}(\mathbf{x}^{(t)}) \mid \mathbf{x}^{(t)}, f^{-1})\right)\right]\right]$$

$$=\sum_{t=0}^{T-1}\sum_{i=1}^{n}\mathbb{E}_{\mathbf{S}^{(t)}|f^1}\left[\mathrm{D}_{\mathrm{KL}}\left(\mathbb{P}(\tilde{g}(x_i^{(t)}) \mid x_i^{(t)}, f^1)||\mathbb{P}(g(x_i^{(t)}) \mid x_i^{(t)}, f^{-1})\right)\right].$$

Then we apply the uniform upper bound over $x$ as follows:

$$\mathrm{D}_{\mathrm{KL}}(P^{1,T}||P^{-1,T}) \leq nT\sup_{x,i}\mathrm{D}_{\mathrm{KL}}\left(\mathbb{P}(\tilde{g}_i(x) \mid x, f^1)||\mathbb{P}(\tilde{g}_i(x) \mid x, f^{-1})\right). \tag{51}$$

Note the KL-divergence between two Gaussians can be computed explicitly from their means and variances as follows: for any $x \in \mathbb{R}^d$

$$\mathrm{D}_{\mathrm{KL}}\left(\mathbb{P}(\tilde{g}_i(x) \mid x, f^1)||\mathbb{P}(\tilde{g}_i(x) \mid x, f^{-1})\right)$$

$$=\mathrm{D}_{\mathrm{KL}}\left(\mathcal{N}(\nabla f^1(x),\sigma^2))||\mathcal{N}(\nabla f^{-1}(x),\sigma^2))\right) = \frac{1}{2\sigma^2}\|\nabla f^1(x) - \nabla f^{-1}(x)\|^2 = \frac{2\mu^2\lambda^2}{\sigma^2}. \tag{52}$$

Plugging (52) into (51), we reach the result (50). $\qquad\square$

Therefore, by Lemma 5 and Lemma 6, we can obtain the lower bound based (44). Specifically, for $f^1$, $f^{-1} \in \mathcal{F}_{L,\mu}$ constructed above, assume that the algorithm $A$ receives stochastic gradients with $\mathcal{N}(0,\sigma^2)$ noise, then we have

$$\inf_{A\in\mathcal{A}_W}\sup_{W\in\mathcal{W}_{n,\beta}}\sup_{\{\tilde{g}_i\}_{i=1}^n\subseteq\mathcal{O}_\sigma^2}\sup_{\{f_i\}_{i=1}^n\subseteq\mathcal{F}_{L,\mu}}\mathbb{E}[f(\hat{x}_{A,\{f_i\}_{i=1}^n,\{\tilde{g}_i\}_{i=1}^n,W,T}) - f^\star]$$

$$\geq \inf_{A\in\mathcal{A}_W}\max_{v\in\{\pm1\}}\mathbb{E}[f^v(\hat{x}_{A,\{f^v\}_{i=1}^n,\mathcal{N}(\nabla f^v(x),\sigma^2),\frac{1-\beta}{n}\mathbb{1}_n\mathbb{1}_n^T+\beta I,T})]$$

$$\geq \inf_{A\in\mathcal{A}_W}\mathbb{E}_{V\sim\mathrm{Unif}(\{\pm1\})}[\mathbb{E}[f^V(\hat{x}_{A,\{f^V\}_{i=1}^n,\mathcal{N}(\nabla f^V(x),\sigma^2),\frac{1-\beta}{n}\mathbb{1}_n\mathbb{1}_n^T+\beta I,T})]]. \tag{53}$$

Plugging Lemma 5 and Lemma 6 into (53) and using Pinsker's inequality, we immediately have

$$\inf_{A\in\mathcal{A}_W}\sup_{W\in\mathcal{W}_{n,\beta}}\sup_{\{\tilde{g}_i\}_{i=1}^n\subseteq\mathcal{O}_\sigma^2}\sup_{\{f_i\}_{i=1}^n\subseteq\mathcal{F}_{L,\mu}}\mathbb{E}[f(\hat{x}_{A,\{f_i\}_{i=1}^n,\{\tilde{g}_i\}_{i=1}^n,W,T}) - f^\star]$$

$$\geq\frac{\mu\lambda^2}{2}\left(1 - \sqrt{\mathrm{D}_{\mathrm{KL}}(P^{1,T}||P^{-1,T})/2}\right)$$

$$=\frac{\mu\lambda^2}{2}\left(1 - \frac{\mu\lambda}{\sigma}\sqrt{nT}\right). \tag{54}$$

Choosing $\lambda = \frac{2\sigma}{3\mu\sqrt{nT}}$ in (54), we reach the lower bound $\Omega(\frac{\sigma^2}{\mu nT})$.

**Instance 2.** Our proof for the term $\Omega(\mu\Delta/L\exp(-\sqrt{\mu/L}\sqrt{1-\beta}T))$ builds on the similar idea to [60]: splitting the function used by Nesterov to prove the lower bound for strongly convex and smooth optimization [52]. However, the number of nodes $n$ is fixed in our analysis while [60] needs $n$ to be varying when establishing the lower bound. Besides, our construction allows the objective functions to be suitable to an arbitrary initialization scope $\Delta = f(x^{(0)}) - \min_x f(x)$ which, however, is not fully addressed and discussed in [60]. These differences make our analysis novel and stronger.

We construct deterministic scenarios where no gradient noise is employed in the oracle. We assume the variable $x \in \ell_2 \triangleq \{([x]_1, [x]_2, \ldots,) : \sum_{r=1}^\infty [x]_r^2 < \infty\}$ to be infinitely dimensional and square-summable for simplicity. It is easy to adapt the argument for finitely dimensional variables as long

as the dimension is proportionally larger than $T$. Without loss of generality, we assume all the algorithms start from $x^{(0)} = 0 \in \ell_2$. Let $M$ be

$$M = \begin{bmatrix} 2 & -1 & & & \\ -1 & 2 & -1 & & \\ & -1 & 2 & -1 & \\ & & \ddots & \ddots & \ddots \end{bmatrix} \in \mathbb{R}^{\infty \times \infty},$$

then it is easy to see $0 \preceq M \preceq 4I$. Let $E_1 \triangleq \{j : 1 \le j \le \lceil \frac{n}{3} \rceil\}$ and $E_2 \triangleq \{j : \lfloor \frac{n}{2} \rfloor + 1 \le j \le \lfloor \frac{n}{2} \rfloor + \lceil \frac{n}{3} \rceil\}$ as in Appendix D.1, and let

$$f_i(x) = \begin{cases} \frac{\mu}{2}\|x\|^2 + \frac{L-\mu}{12}\frac{n}{\lceil n/3\rceil}\left([x]_1^2 + \sum_{r\ge 1}([x]_{2r} - [x]_{2r+1})^2 - 2\lambda[x]_1\right) & \text{if } i \in E_1, \\ \frac{\mu}{2}\|x\|^2 + \frac{L-\mu}{12}\frac{n}{\lceil n/3\rceil}\sum_{r\ge 1}([x]_{2r-1} - [x]_{2r})^2 & \text{if } i \in E_2, \\ \frac{\mu}{2}\|x\|^2 & \text{otherwise.} \end{cases} \quad (55)$$

where $\lambda \in \mathbb{R}$ is to be specified. It is easy to see that $[x]_1^2 + \sum_{r\ge 1}([x]_{2r} - [x]_{2r+1})^2 - 2\lambda[x]_1$ and $\sum_{r\ge 1}([x]_{2r-1} - [x]_{2r})^2$ are convex and 4-smooth. We thus have all $f_i$ are $L$-smooth and $\mu$-strongly convex, which implies $f_i \in \mathcal{F}_{L,\mu}$ for all $1 \le i \le n$. We further have $f(x) = \frac{1}{n}\sum_{i=1}^n f_i(x) = \frac{\mu}{2}\|x\|^2 + \frac{L-\mu}{12}\left(x^T M x - 2\lambda[x]_1\right)$.

For the functions defined above, we establish that

**Lemma 7.** *Denote $\kappa := L/\mu > 1$, then it holds that for any $x$ and $r \ge 1$ satisfying $\text{prog}(x) \le r$,*

$$f(x) - \min_x f(x) \ge \frac{\mu}{2L}\left(1 - \frac{6}{\sqrt{3 + 6\kappa} + 3}\right)^{2r}(f(x^{(0)}) - \min_x f(x)).$$

*Proof.* The minimum $x^\star$ of function $f$ satisfies $\left(\frac{L-\mu}{6}M + \mu\right)x - \lambda\frac{L-\mu}{6}e_1 = 0$, which is equivalent to

$$\left(2 + \frac{6}{\kappa - 1}\right)[x]_1 - [x]_2 = \lambda,$$

$$-[x]_{j-1} + \left(2 + \frac{6}{\kappa - 1}\right)[x]_j - [x]_{j+1} = 0, \quad \forall j \ge 2. \quad (56)$$

Let $q$ be the smallest root of the equation $q^2 - \left(2 + \frac{6}{\kappa-1}\right)q + 1 = 0$, then the variable

$$x^\star = \left([x^\star]_j = \lambda q^j\right)_{j \ge 1}$$

satisfies (56). By the strong convexity of $f$, $x^\star$ is the unique solution. Therefore, when $\text{prog}(x) \le r$, it holds that

$$\|x - x^\star\|^2 \ge \sum_{j=r+1}^{\infty} \lambda^2 q^{2j} = \lambda^2 \frac{q^{2(r+1)}}{1 - q^2} = q^{2r}\|x^{(0)} - x^\star\|^2.$$

Finally, noting that

$$q = 1 + \frac{3 - \sqrt{3 + 6\kappa}}{\kappa - 1} = 1 - \frac{6}{\sqrt{3 + 6\kappa} + 3},$$

and using the strong convexity of $f$, we reach the conclusion. $\qquad\square$

Moreover, it is easy to see that the optimal value of $f(x)$ is

$$\min_x f(x) = f(x^\star) = -\frac{L-\mu}{12}\lambda[x^\star]_1 = -\frac{L-\mu}{12}\lambda^2 q.$$

Therefore, for any given $\Delta > 0$, we can chose $\lambda = \sqrt{\frac{12\Delta}{(L-\mu)q}}$ such that $f(x^{(0)}) - \min_x f(x) = \Delta$.

Similar to the argument in Appendix D.1, we have

$$\text{prog}(\nabla f_i(x)) \begin{cases} = \text{prog}(x) + 1 & \text{if } \{\text{prog}(x) \text{ is even and } i \in E_1\} \cup \{\text{prog}(x) \text{ is odd and } i \in E_2\} \\ \leq \text{prog}(x) & \text{otherwise.} \end{cases}$$

We further have

$$\text{prog}^{(T)} = \max_{1 \leq i \leq n,\, 0 \leq s \leq T} \text{prog}(x_i^{(t)}) \leq \left\lfloor \frac{T}{\text{dist}(E_1, E_2)} \right\rfloor + 1, \quad \forall T \geq 0. \tag{57}$$

Therefore, with $T$ rounds of gossip communications budget in total, starting from $x_i^{(0)} = 0$ for any $i = 1, \ldots, n$, any gossip algorithm $A$ can only achieves at most $\text{prog}(\hat{x}) = 1 + \lfloor \frac{T}{\text{dist}(E_1, E_2)} \rfloor$, which, combined with Lemma 7, leads to lower bound that

$$f(\hat{x}) - \min_x f(x) \geq \frac{\mu \Delta}{2L} \left( 1 - \frac{6}{\sqrt{3 + 6\kappa} + 3} \right)^{2 + 2\lfloor \frac{T}{\text{dist}(E_1, E_2)} \rfloor}$$

$$= \Omega \left( \frac{\mu}{L} \exp \left( -\frac{\sqrt{6} \lfloor \frac{T}{\text{dist}(E_1, E_2)} \rfloor}{\sqrt{\kappa}} \right) \Delta \right). \tag{58}$$

where $\Delta$ measures the initialization $\mathbb{E}[f(x^{(0)})] - \min_x f(x)$. The rest follows using the ring-lattice associated weight matrix $W \in \mathcal{W}_{n,\beta}$ such that

$$\text{dist}(E_1, E_2) = \Omega((1 - \beta)^{-\frac{1}{2}}).$$

### D.3 Connectivity measures in common weight matrices

Table D.3 lists the order of the connectivity measure $\beta$ for weight matrices, if generated through the Laplacian rule $W = I - L/d_{\max}$ in which $L$ is the Laplacian matrix and $d_{\max}$ is the maximum degree, associated with commonly-used topologies. Since $\cos(\frac{\pi}{n}) \geq 1 - \frac{\pi^2}{2n^2}$, we have $[0, 1 - \frac{\pi^2}{2n^2}] \subseteq [0, \cos(\pi/n)]$. Since $\beta$ listed in Table D.3 lies in $[0, 1 - \frac{\pi^2}{2n^2}]$ for sufficiently large $n$, we conclude that our lower bound established in Theorems 2 and 3 applies to weight matrices associated with topologies listed in Table D.3.

Table 8: Order of connectivity measure $\beta$. "E.-R. Rand": Erdos-Renyi random graph $G(n, p)$ with probability $p = (1 + a) \ln(n)/n$ for some $a > 0$; "Geo. Rand": geometric random graph $G(n, r)$ with radius $r^2 = (1 + a) \ln(n)/n$ for some $a > 0$.

| Topology | Order of $\beta$ |
| --- | --- |
| Grid [49] | $1 - \Theta([n \ln(n)]^{-1})$ |
| Torus [49] | $1 - \Theta(n^{-1})$ |
| Hypercube [49] | $1 - \Theta([\ln(n)]^{-1})$ |
| Exponential [75] | $1 - \Theta([\ln(n)]^{-1})$ |
| Complete | $0$ |
| E.-R. Rand. [6] | $1 - \Theta([\ln(n)]^{-1})$ |
| Geo. Rand. [7] | $1 - \Theta(\ln(n)n^{-1})$ |

## E    Convergence in MG-DSGD

### E.1    Smooth and non-convex setting

This section examines the convergence rate of MG-DSGD under the smooth and non-convex setting. Since MG-DSGD is a direct variant based on the vanilla DSGD, we first establish the convergence of DSGD to facilitate the convergence of MG-DSGD.

#### E.1.1    Convergence rate of DSGD

The convergence rate of DSGD has been established in literatures such as [30, 13]. We adjust the analysis therein to achieve a slightly different result, which lays the foundation for the later convergence analysis in MG-DSGD.

The following lemma established in [30, Lemma 8] (also in [13, Lemma 6]) shows how $\mathbb{E}[f(\bar{x}^{(k)})]$ evolves with iterations.

**Lemma 8** (DESCENT LEMMA [30]). *If $\{f_i(x)\}_{i=1}^n \subseteq \mathcal{F}_L$, $\{\tilde{g}_i\} \subseteq O_{\sigma^2}$, and learning rate $\gamma < \frac{1}{4L}$, it holds for $k = 0, 1, 2, \cdots$ that*

$$\mathbb{E}[f(\bar{x}^{(k+1)})] \leq \mathbb{E}[f(\bar{x}^{(k)})] - \frac{\gamma}{4}\mathbb{E}\|\nabla f(\bar{x}^{(k)})\|^2 + \frac{3\gamma L^2}{4n}\mathbb{E}\|\mathbf{x}^{(k)} - \bar{\mathbf{x}}^{(k)}\|_F^2 + \frac{\gamma^2\sigma^2 L}{2n}, \qquad (59)$$

*where $\bar{\mathbf{x}}^{(k)} = [(\bar{x}^{(k)})^T; \cdots ; (\bar{x}^{(k)})^T] \in \mathbb{R}^{n \times d}$, and $\bar{x}^{(k)} = \frac{1}{n}\sum_{i=1}^n x_i^{(k)}$.*

The following lemma established in [13, Lemma 8] shows how the ergodic consensus term evolves with iterations.

**Lemma 9** (CONSENSUS LEMMA [13]). *If $\{f_i(x)\}_{i=1}^n \subseteq \mathcal{F}_L$, $\{\tilde{g}_i\} \subseteq O_{\sigma^2}$, $W \in \mathcal{W}_{n,\beta}$, $\frac{1}{n}\sum_{i=1}^n \|\nabla f_i(x) - \nabla f(x)\|^2 \leq b^2$ for any $x \in \mathbb{R}^d$, and learning rate $\gamma \leq \frac{1-\beta}{4\beta L}$, it holds that*

$$\frac{1}{K+1}\sum_{k=0}^K \mathbb{E}\|\mathbf{x}^{(k)} - \bar{\mathbf{x}}^{(k)}\|_F^2 \leq \frac{12n\beta^2\gamma^2}{(1-\beta)^2(K+1)}\sum_{k=0}^K \mathbb{E}\|\nabla f(\bar{x}^{(k)})\|^2 + \frac{2n\gamma^2\beta^2}{1-\beta}\left(\frac{3b^2}{1-\beta} + \sigma^2\right). \tag{60}$$

*Proof.* The Lemma 8 in [13] covers both DSGD and DSGD with periodic global average. To recover the result for DSGD, we let $C_\beta = 1/(1-\beta)$, $D_\beta = 1/(1-\beta)$ in [13, Lemma 8] to achieve (60). □

The following lemma establishes the convergence rate of DSGD under the smooth and non-convex setting.

**Lemma 10** (DSGD CONVERGENCE). *If $\{f_i(x)\}_{i=1}^n \subseteq \mathcal{F}_L$, $\{\tilde{g}_i\} \subseteq O_{\sigma^2}$, $W \in \mathcal{W}_{n,\beta}$, $\frac{1}{n}\sum_{i=1}^n \|\nabla f_i(x) - \nabla f(x)\|^2 \leq b^2$ for any $x \in \mathbb{R}^d$, and learning rate is set as*

$$\gamma = \min\left\{\frac{\sqrt{2n\Delta}}{\sigma\sqrt{L(K+1)}}, \frac{1-\beta}{6\beta L}, \frac{1}{4L}\right\}, \tag{61}$$

*where $\Delta = \mathbb{E}[f(x^{(0)})] - f^\star$, it holds that*

$$\frac{1}{K+1}\sum_{k=0}^K \mathbb{E}\|\nabla f(\bar{x}^{(k)})\|^2 = \mathcal{O}\left(\frac{\sigma\sqrt{\Delta L}}{\sqrt{nK}} + \frac{\beta^2 Ln\Delta}{(1-\beta)K} + \frac{\beta^2 Lnb^2\Delta}{(1-\beta)^2 K\sigma^2} + \frac{\beta L\Delta}{(1-\beta)K} + \frac{L\Delta}{K}\right) \tag{62}$$

**Remark 9.** *This result is slightly different from Theorem 2 in [30] where $\beta$ does not appear in the numerator of any terms in the upper bound therein. However, as we will show in the analysis of MG-DSGD, the $\beta$ appearing in the numerator of the third term in (62) can significantly reduce the influence of data heterogeneity $b^2$ if $\beta \to 0$.*

*Proof.* When $\gamma \leq 1/(4L)$, we know from Lemma 8 that

$$\mathbb{E}\|\nabla f(\bar{x}^{(k)})\|^2 \leq \frac{4}{\gamma}\mathbb{E}[f(\bar{x}^{(k)})] - \frac{4}{\gamma}\mathbb{E}[f(\bar{x}^{(k+1)})] + \frac{3L^2}{n}\mathbb{E}\|\mathbf{x}^{(k)} - \bar{\mathbf{x}}^{(k)}\|_F^2 + \frac{2\gamma\sigma^2 L}{n}, \qquad (63)$$

Taking running average over the above inequality, we obtain

$$\frac{1}{K+1}\sum_{k=0}^K \mathbb{E}\|\nabla f(\bar{x}^{(k)})\|^2 \leq \frac{4}{\gamma(K+1)}\left(\mathbb{E}[f(\bar{x}^{(0)})] - \mathbb{E}[f(\bar{x}^{(K+1)})]\right) + \frac{2\gamma\sigma^2 L}{n},$$

$$+ \frac{3L^2}{n(K+1)}\sum_{k=0}^K \mathbb{E}\|\mathbf{x}^{(k)} - \bar{\mathbf{x}}^{(k)}\|_F^2 \tag{64}$$

With Lemma (60) and the fact that $\mathbb{E}[f(\bar{x}^{(0)})] - \mathbb{E}[f(\bar{x}^{(K+1)})] \leq \mathbb{E}[f(\bar{x}^{(0)})] - f^\star$, (64) becomes

$$\left(1 - \frac{36\beta^2 L^2\gamma^2}{(1-\beta)^2}\right)\frac{1}{K+1}\sum_{k=0}^K \mathbb{E}\|\nabla f(\bar{x}^{(k)})\|^2$$

$$\leq \frac{4\left(\mathbb{E}[f(\bar{x}^{(0)})] - f^\star\right)}{\gamma(K+1)} + \frac{2\gamma\sigma^2 L}{n} + \frac{6\gamma^2\beta^2 L^2\sigma^2}{1-\beta} + \frac{18\gamma^2\beta^2 L^2 b^2}{(1-\beta)^2} \tag{65}$$

If learning rate $\gamma \le \frac{1-\beta}{12\beta L}$, it holds that $1 - \frac{36\beta^2 L^2 \gamma^2}{1-\beta} \ge 1/2$ and hence

$$\frac{1}{K+1}\sum_{k=0}^{K}\mathbb{E}\|\nabla f(\bar{x}^{(k)})\|^2 \le \frac{8\Delta}{\gamma(K+1)} + \frac{4\gamma\sigma^2 L}{n} + \frac{12\gamma^2\beta^2 L^2\sigma^2}{1-\beta} + \frac{36\gamma^2\beta^2 L^2 b^2}{(1-\beta)^2} \qquad (66)$$

where we introduce constant $\Delta = \mathbb{E}[f(\bar{x}^{(0)})] - f^\star$. Next we set

$$\gamma_1 = \frac{\sqrt{2\Delta n}}{\sigma\sqrt{L(K+1)}}, \quad \gamma_2 = \frac{1-\beta}{6\beta L}, \quad \gamma_3 = \frac{1}{4L}, \quad \gamma = \min\{\gamma_1, \gamma_2, \gamma_3\}. \qquad (67)$$

Apparently, it holds that

$$\frac{8\Delta}{\gamma(K+1)} \le \frac{8\Delta}{\gamma_1(K+1)} + \frac{8\Delta}{\gamma_2(K+1)} + \frac{8\Delta}{\gamma_3(K+1)}$$

$$= \mathcal{O}\left(\frac{\sigma\sqrt{\Delta L}}{\sqrt{nK}} + \frac{\beta\Delta L}{(1-\beta)K} + \frac{\Delta L}{K}\right) \qquad (68)$$

and

$$\frac{4\gamma\sigma^2 L}{n} + \frac{12\gamma^2\beta^2 L^2\sigma^2}{1-\beta} + \frac{36\gamma^2\beta^2 L^2 b^2}{(1-\beta)^2} \le \frac{4\gamma_1\sigma^2 L}{n} + \frac{12\gamma_1^2\beta^2 L^2\sigma^2}{1-\beta} + \frac{36\gamma_1^2\beta^2 L^2 b^2}{(1-\beta)^2}$$

$$= \mathcal{O}\left(\frac{\sigma\sqrt{\Delta L}}{\sqrt{nK}} + \frac{\beta^2 L\Delta n}{(1-\beta)K} + \frac{\beta^2 L\Delta n b^2}{(1-\beta)^2 K\sigma^2}\right) \qquad (69)$$

Combining (66), (68) and (69), we have (62).

$\square$

### E.1.2 Convergence rate of MG-DSGD

Recall the recursions of DSGD and MG-DSGD as follows.

$$\text{(DSGD)} \quad \mathbf{x}^{(k+1)} = W\big(\mathbf{x}^{(k)} - \gamma\nabla F(\mathbf{x}^{(k)};\xi^{(k)})\big) \qquad (70)$$

$$\text{(MG-DSGD)} \quad \mathbf{x}^{(k+1)} = \bar{M}\big(\mathbf{x}^{(k)} - \gamma\mathbf{g}^{(k)})\big), \quad \text{where} \quad \mathbf{g}^{(k)} = \frac{1}{R}\sum_{r=1}^{R}\nabla F(\mathbf{x}^{(k)};\xi^{(k,r)}) \qquad (71)$$

where $\bar{M}$ is doubly stochastic (see Prop. 1). MG-DSGD has two differences from the vanilla DSGD. First, the weight matrix $W$ is replaced with $\bar{M}$. Second, the stochastic gradient $\mathbf{g}^{(k)}$ is achieved via gradient accumulation. This implies that the convergence analysis of MG-DSGD can follow that of vanilla DSGD. We only need to pay attentions to the influence of $\bar{M}$ obtained by fast gossip averaging and the $\mathbf{g}^{(k)}$ achieved by gradient accumulation. To proceed, we notice that

$$\mathbb{E}[\|g_i^{(k)} - \nabla f_i(x)\|^2] = \frac{1}{R}\mathbb{E}[\|\nabla F(x_i^{(k)};\xi_i^{(k,r)}) - \nabla f_i(x)\|^2] \le \frac{\sigma^2}{R}$$

because $\{\xi_i^{(k,r)}\}_{r=1}^R$ are sampled independently. We introduce $\tilde{\sigma}^2 \triangleq \sigma^2/R$ for notation simplicity. In addition, we know from Prop. 1 that

$$\|\bar{M} - \frac{1}{n}\mathbb{1}_n\mathbb{1}_n^T\|_2 \le \tilde{\beta}, \quad \text{where} \quad \tilde{\beta} := \sqrt{2}\Big(1 - \sqrt{1-\beta}\Big)^R.$$

If we let

$$R = \left\lceil \frac{\max\{\ln(2), \frac{1}{2}\ln(n\max\{1, \frac{b^2}{\sigma^2\sqrt{1-\beta}}\})\}}{\sqrt{1-\beta}} \right\rceil = \tilde{O}\left(\frac{1}{\sqrt{1-\beta}}\right), \qquad (72)$$

then it holds that

$$\tilde{\beta} \triangleq \sqrt{2}(1 - \sqrt{1-\beta})^R \le \sqrt{2}e^{-\sqrt{1-\beta}R} \le \min\left\{\frac{1}{\sqrt{2}}, \sqrt{2}\left(n\max\left\{1, \frac{b^2}{\sigma^2\sqrt{1-\beta}}\right\}\right)^{-\frac{1}{2}}\right\}. \qquad (73)$$

We thus immediately have $1 - \tilde{\beta} = \Omega(1)$ and

$$\tilde{\beta}^2 n \max\{1, b^2 R/\sigma^2\} = \tilde{O}\left(\tilde{\beta}^2 n \max\left\{1, \frac{b^2}{\sigma^2\sqrt{1-\beta}}\right\}\right) = \tilde{O}(1). \tag{74}$$

**Theorem 4 (Formal version).** *Given $L > 0$, $n \geq 2$, $\beta \in [0, 1)$, $\sigma > 0$, and let $A$ denote Algorithm 1. Assuming that $\frac{1}{n}\sum_{i=1}^n \|\nabla f_i(x) - \nabla f(x)\|^2 \leq b^2$ for any $x \in \mathbb{R}^n$ and the number of gossip rounds $R$ is set as in (72), the convergence of $A$ can be bounded for any $\{f_i\}_{i=1}^n \subseteq \mathcal{F}_L$ and any $W \in \mathcal{W}_{n,\beta}$ that*

$$\frac{1}{K}\sum_{k=1}^K \mathbb{E}\|\nabla f(\bar{x}^{(k)})\|^2 = \tilde{O}\left(\frac{\sigma\sqrt{\Delta L}}{\sqrt{nT}} + \frac{L\Delta}{\sqrt{1-\beta}T}\right) \quad where \quad \bar{x}^{(k)} = \frac{1}{n}\sum_{i=1}^n x_i^{(k)} \tag{75}$$

*and $\Delta = \mathbb{E}[f(\bar{x}^{(0)})] - f^\star$.*

**Remark 10.** *Comparing with the established lower bound (33), we find the upper bound (75) matches it tightly in all constants.*

*Proof.* The convergence analysis of DSGD applies to MG-DSGD by replacing $\sigma^2$ with $\tilde{\sigma}^2$, and $\beta$ with $\tilde{\beta}$. As a result, we know from Lemma 10 that if the learning rate is set as

$$\gamma = \min\{\frac{\sqrt{\Delta n}}{\tilde{\sigma}\sqrt{L(K+1)}}, \frac{1-\tilde{\beta}}{6\tilde{\beta}L}, \frac{1}{4L}\} = \tilde{O}\left(\min\{\frac{\sqrt{2\Delta n}}{\sigma\sqrt{(1-\beta)LT}}, \frac{1}{L}\}\right) \tag{76}$$

where the last equality holds since $T = KR$ (where $T$ is the number of total gradient queries or communication rounds), it holds that

$$\frac{1}{K+1}\sum_{k=0}^K \mathbb{E}\|\nabla f(\bar{x}^{(k)})\|^2 = O\left(\frac{\tilde{\sigma}\sqrt{\Delta L}}{\sqrt{nK}} + \frac{\tilde{\beta}^2 L\Delta n}{(1-\tilde{\beta})K} + \frac{\tilde{\beta}^2 L\Delta n b^2}{(1-\tilde{\beta})^2 K\tilde{\sigma}^2} + \frac{\tilde{\beta}L\Delta}{(1-\tilde{\beta})K} + \frac{L\Delta}{K}\right)$$

$$\overset{\tilde{\beta}\leq\frac{1}{\sqrt{2}};\ T=KR}{=} O\left(\frac{\sigma\sqrt{\Delta L}}{\sqrt{nT}} + \frac{\tilde{\beta}^2 RL\Delta n}{T} + \frac{\tilde{\beta}^2 R^2 L\Delta n b^2}{T\sigma^2} + \frac{RL\Delta}{T}\right)$$

$$\overset{(72)}{=} \tilde{O}\left(\frac{\sigma\sqrt{\Delta L}}{\sqrt{nT}} + \frac{\tilde{\beta}^2 RL\Delta n}{T} + \frac{\tilde{\beta}^2 RL\Delta n \frac{b^2}{\sigma^2\sqrt{1-\beta}}}{T} + \frac{RL\Delta}{T}\right)$$

$$\overset{(74)}{=} \tilde{O}\left(\frac{\sigma\sqrt{\Delta L}}{\sqrt{nT}} + \frac{RL\Delta}{T}\right) \overset{(72)}{=} \tilde{O}\left(\frac{\sigma\sqrt{\Delta L}}{\sqrt{nT}} + \frac{L\Delta}{\sqrt{1-\beta}T}\right).$$

$\square$

## E.2 Smooth and non-convex setting under PL condition

Similar to Appendix E.1, we first establish the convergence of DSGD under the PL condition, and then derive the convergence of MG-DSGD.

### E.2.1 Convergence rate of DSGD

Substituting (3) to inequality (59), we achieve the following descent lemma.

**Lemma 11** (DESCENT LEMMA). *If $\{f_i(x)\}_{i=1}^n \subseteq \mathcal{F}_{L,\mu}$, $\{\tilde{g}_i\} \subseteq O_{\sigma^2}$ and learning rate $\gamma \leq \frac{1}{4L}$, it holds for $k = 0, 1, 2, \cdots$ that*

$$\mathbb{E}[f(\bar{x}^{(k+1)})] - f^\star \leq (1 - \frac{\mu\gamma}{2})(\mathbb{E}[f(\bar{x}^{(k)})] - f^\star) + \frac{3\gamma L^2}{4n}\mathbb{E}\|\mathbf{x}^{(k)} - \bar{\mathbf{x}}^{(k)}\|_F^2 + \frac{\gamma^2\sigma^2 L}{2n}. \tag{77}$$

The following lemma establishes the consensus lemma for smooth and non-convex loss functions satisfying the PL condition.

**Lemma 12** (CONSENSUS LEMMA). *If* $\{f_i(x)\}_{i=1}^n \subseteq \mathcal{F}_{L,\mu}$, $\{\tilde{g}_i\} \subseteq O_{\sigma^2}$, $W \in \mathcal{W}_{n,\beta}$, $\frac{1}{n}\sum_{i=1}^n \|\nabla f_i(x) - \nabla f(x)\|^2 \le b^2$ *for any* $x \in \mathbb{R}^d$, *and learning rate* $\gamma \le \frac{1-\beta}{3\beta L}$, *it holds for* $k = 0, 1, 2, \cdots$ *that*

$$\mathbb{E}\|\mathbf{x}^{(k+1)} - \bar{\mathbf{x}}^{(k+1)}\|^2 \le \left(\frac{1+\beta}{2}\right)\mathbb{E}\|\mathbf{x}^{(k)} - \bar{\mathbf{x}}^{(k)}\|^2 + \frac{6n\beta^2\gamma^2 L}{1-\beta}\left(\mathbb{E}f(\bar{x}^{(k)}) - f^\star\right)$$
$$+ n\gamma^2\beta^2\sigma^2 + \frac{3n\beta^2\gamma^2 b^2}{1-\beta}. \tag{78}$$

*Proof.* When $\{f_i\}_{i=1}^L \subseteq \mathcal{F}_L$, it is know from [13, Eq. (78)] that

$$\mathbb{E}\|\mathbf{x}^{(k+1)} - \bar{\mathbf{x}}^{(k+1)}\|^2 \le \left(\beta + \frac{3\beta^2\gamma^2 L^2}{1-\beta}\right)\mathbb{E}\|\mathbf{x}^{(k)} - \bar{\mathbf{x}}^{(k)}\|^2 + \frac{3n\beta^2\gamma^2\mathbb{E}\|\nabla f(\bar{x}^{(k)})\|^2}{1-\beta}$$
$$+ n\gamma^2\beta^2\sigma^2 + \frac{3n\beta^2\gamma^2 b^2}{1-\beta}. \tag{79}$$

If $\gamma \le (1-\beta)/(3\beta L)$, it holds that

$$\beta + \frac{3\beta^2\gamma^2 L^2}{1-\beta} \le \frac{1+\beta}{2}. \tag{80}$$

Moreover, since $\{f_i\}_{i=1}^L \subseteq \mathcal{F}_L$ and hence $f(x) \in \mathcal{F}_L$, we have

$$\|\nabla f(\bar{x}^{(t)})\|^2 \le 2L\left(f(\bar{x}^{(t)}) - f^\star\right). \tag{81}$$

Substituting (80) and (81) to (79), we achieve (78). $\qquad\square$

The following lemma establishes the convergence rate of DSGD under the PL condition. Using different proof techniques from those for the strongly-convex scenario in [30], we can show how fast the last-iterate variable, i.e., $f(\bar{x}^{(K)}) - f^\star$, converges. In contrast, authors in [30] show the ergodic convergence, i.e., $\frac{1}{H_K}\sum_{i=1}^K h_k f(\bar{x}^{(k)}) - f^\star$ where $h_k > 0$ is some positive weight and $H_K = \sum_{k=1}^K h_k$. Moreover, the $\beta$ appearing in the numerator of the third term in (83) can significantly reduce the influence of data heterogeneity $b^2$ when $\beta \to 0$.

**Lemma 13** (DSGD CONVERGENCE). *If* $\{f_i(x)\}_{i=1}^n \subseteq \mathcal{F}_{L,\mu}$, $\{\tilde{g}_i\} \subseteq O_{\sigma^2}$, $W \in \mathcal{W}_{n,\beta}$, $\frac{1}{n}\sum_{i=1}^n \|\nabla f_i(x) - \nabla f(x)\|^2 \le b^2$ *for any* $x \in \mathbb{R}^d$, *and learning rate* $\gamma$ *is set as*

$$\gamma = \min\left\{\frac{4}{\mu K}\ln\left(\frac{\Delta\mu^2 nK}{\sigma^2 L}\right), \frac{1-\beta}{4L}, \frac{\mu(1-\beta)}{24n\beta L^2}\right\}, \tag{82}$$

*it holds that*

$$\mathbb{E}[f(\bar{x}^{(K)})] - f^\star + L\mathbb{E}\|\mathbf{x}^{(K)} - \bar{\mathbf{x}}^{(K)}\|^2 \tag{83}$$
$$= \tilde{\mathcal{O}}\left(\frac{\sigma^2 L}{\mu^2 nK} + \frac{c_1\beta^2\sigma^2}{\mu^2 K^2(1-\beta)} + \frac{c_2\beta^2 b^2}{\mu^2 K^2(1-\beta)^2} + \Delta\exp\left(-\frac{\mu(1-\beta)K}{L}\right) + \Delta\exp\left(-\frac{\mu^2(1-\beta)K}{n\beta L^2}\right)\right)$$

where $\Delta = \mathbb{E}[f(x^{(0)})] - f^\star$, $c_1$ and $c_2$ are constants defined as follows

$$c_1 = \frac{6L}{\mu} + \frac{24L^3}{\mu^2} + 4nL, \quad c_2 = \frac{18L^2}{\mu} + 12nL. \tag{84}$$

*Proof.* With Lemmas 11 and 12, if $\gamma$ satisfies

$$\gamma \le \frac{1}{4L}, \quad \text{and} \quad \gamma \le \frac{1-\beta}{3\beta L}, \tag{85}$$

then we have

$$
\underbrace{\begin{bmatrix} \mathbb{E}[f(\bar{x}^{(k+1)})] - f^\star \\ L\mathbb{E}\|\mathbf{x}^{(k+1)} - \bar{\mathbf{x}}^{(k+1)}\|^2 \end{bmatrix}}_{:=z^{(k+1)}} \leq \underbrace{\begin{bmatrix} 1 - \frac{\mu\gamma}{2} & \frac{3\gamma L}{4n} \\ \frac{6n\beta^2\gamma^2 L^2}{1-\beta} & \frac{1+\beta}{2} \end{bmatrix}}_{:=A} \underbrace{\begin{bmatrix} \mathbb{E}[f(\bar{x}^{(k)})] - f^\star \\ L\mathbb{E}\|\mathbf{x}^{(k)} - \bar{\mathbf{x}}^{(k)}\|^2 \end{bmatrix}}_{:=z^{(k)}}
$$
$$
+ \underbrace{\begin{bmatrix} \frac{\gamma^2\sigma^2 L}{2n} \\ nL\gamma^2\beta^2\sigma^2 + \frac{3nL\beta^2\gamma^2 b^2}{1-\beta} \end{bmatrix}}_{:=s} \tag{86}
$$

By recursing the above inequality, we have

$$
z^{(k)} \leq A^k z^{(0)} + \sum_{\ell=0}^{k-1} A^\ell s \leq A^k z^{(0)} + (I - A)^{-1} s \tag{87}
$$

where the last inequality holds because $A$ is nonnegative and

$$
\rho(A) \leq \|A\|_1 \leq \max\{1 - \frac{\mu\gamma}{2} + \frac{6n\beta^2\gamma^2 L^2}{1-\beta}, \frac{3\gamma L}{4n} + \frac{1+\beta}{2}\} \leq 1 - \frac{\mu\gamma}{4}. \tag{88}
$$

The last inequality in (88) holds when $\gamma$ satisfies

$$
\gamma \leq \frac{(1-\beta)n}{3L} \implies \frac{3\gamma L}{4n} + \frac{1+\beta}{2} \leq \frac{3+\beta}{4} \tag{89}
$$

$$
\gamma \leq \frac{1-\beta}{\mu} \implies \frac{3+\beta}{4} \leq 1 - \frac{\mu\gamma}{4} \tag{90}
$$

$$
\gamma \leq \frac{\mu(1-\beta)}{24n\beta^2 L^2} \implies 1 - \frac{\mu\gamma}{2} + \frac{6n\beta^2\gamma^2 L^2}{1-\beta} \leq 1 - \frac{\mu\gamma}{4} \tag{91}
$$

Furthermore, we can derive from (87) that

$$
\begin{aligned}
\|z^{(k)}\|_1 &\leq \|A^k\|_1 \|z^{(0)}\|_1 + \|(I-A)^{-1}s\|_1 \\
&\leq \|A\|_1^k \|z^{(0)}\|_1 + \|(I-A)^{-1}s\|_1 \\
&\overset{(88)}{\leq} (1 - \frac{\mu\gamma}{4})^k \|z^{(0)}\|_1 + \|(I-A)^{-1}s\|_1.
\end{aligned} \tag{92}
$$

It is easy to verify that

$$
\begin{aligned}
(I-A)^{-1}s &= \frac{1}{\frac{\mu\gamma(1-\beta)}{4} - \frac{9\beta^2\gamma^3 L^3}{2(1-\beta)}} \begin{bmatrix} \frac{1-\beta}{2} & \frac{3\gamma L}{4n} \\ \frac{6n\beta^2\gamma^2 L^2}{1-\beta} & \frac{\mu\gamma}{2} \end{bmatrix} \begin{bmatrix} \frac{\gamma^2\sigma^2 L}{2n} \\ nL\gamma^2\beta^2\sigma^2 + \frac{3nL\beta^2\gamma^2 b^2}{1-\beta} \end{bmatrix} \\
&\overset{(a)}{\leq} \frac{8}{\mu\gamma(1-\beta)} \begin{bmatrix} \frac{1-\beta}{2} & \frac{3\gamma L}{4n} \\ \frac{6n\beta^2\gamma^2 L^2}{1-\beta} & \frac{\mu\gamma}{2} \end{bmatrix} \begin{bmatrix} \frac{\gamma^2\sigma^2 L}{2n} \\ nL\gamma^2\beta^2\sigma^2 + \frac{3nL\beta^2\gamma^2 b^2}{1-\beta} \end{bmatrix} \\
&= \begin{bmatrix} \frac{2\gamma\sigma^2 L}{\mu n} + \frac{6L^2\gamma^2\beta^2\sigma^2}{\mu(1-\beta)} + \frac{18L^2\beta^2\gamma^2 b^2}{\mu(1-\beta)^2} \\ \frac{24\beta^2\gamma^3\sigma^2 L^3}{\mu(1-\beta)^2} + \frac{4nL\gamma^2\beta^2\sigma^2}{1-\beta} + \frac{12nL\beta^2\gamma^2 b^2}{(1-\beta)^2} \end{bmatrix},
\end{aligned} \tag{93}
$$

where (a) holds when

$$
\gamma \leq \frac{1-\beta}{6\beta L}\sqrt{\frac{\mu}{L}}. \tag{94}
$$

Inequality (93) implies that

$$
\begin{aligned}
\|(I-A)^{-1}s\|_1 &\leq \frac{2\gamma\sigma^2 L}{\mu n} + \left(\frac{6L^2}{\mu} + \frac{24\gamma L^3}{\mu(1-\beta)} + 4nL\right)\frac{\gamma^2\beta^2\sigma^2}{1-\beta} + \left(\frac{18L^2}{\mu} + 12nL\right)\frac{\gamma^2\beta^2 b^2}{(1-\beta)^2} \\
&\overset{(a)}{\leq} \frac{2\gamma\sigma^2 L}{\mu n} + \left(\frac{6L}{\mu} + \frac{24L^3}{\mu^2} + 4nL\right)\frac{\gamma^2\beta^2\sigma^2}{1-\beta} + \left(\frac{18L^2}{\mu} + 12nL\right)\frac{\gamma^2\beta^2 b^2}{(1-\beta)^2} \\
&\overset{(b)}{=} \frac{2\gamma\sigma^2 L}{\mu n} + \frac{c_1\gamma^2\beta^2\sigma^2}{1-\beta} + \frac{c_2\gamma^2\beta^2 b^2}{(1-\beta)^2}
\end{aligned} \tag{95}
$$

where (a) holds because $\gamma$ satisfies (90), and (b) holds by introducing

$$c_1 := \frac{6L}{\mu} + \frac{24L^3}{\mu^2} + 4nL, \quad c_2 := \frac{18L^2}{\mu} + 12nL. \tag{96}$$

Substituting (95) to (92), and recalling the definition of $z^{(k)}$ in (86), we have (at iteration $K$)

$$\mathbb{E}[f(\bar{x}^{(K)})] - f^\star + L\mathbb{E}\|\mathbf{x}^{(K)} - \bar{\mathbf{x}}^{(K)}\|^2$$
$$\leq \Delta(1 - \frac{\mu\gamma}{4})^K + \frac{2\gamma\sigma^2 L}{\mu n} + \frac{c_1\gamma^2\beta^2\sigma^2}{1 - \beta} + \frac{c_2\gamma^2\beta^2 b^2}{(1 - \beta)^2}$$
$$\leq \Delta\exp(-\frac{\mu\gamma K}{4}) + \frac{2\gamma\sigma^2 L}{\mu n} + \frac{c_1\gamma^2\beta^2\sigma^2}{1 - \beta} + \frac{c_2\gamma^2\beta^2 b^2}{(1 - \beta)^2} \tag{97}$$

where $\Delta = \mathbb{E}[f(\bar{x}^{(0)})] - f^\star + L\mathbb{E}\|\mathbf{x}^{(0)} - \bar{\mathbf{x}}^{(0)}\|^2 = \mathbb{E}[f(x^{(0)})] - f^\star$ if $x_i^{(0)} = 0$ for any $i \in [n]$.
Next we let

$$\gamma_1 = \frac{4}{\mu K}\ln(\frac{\Delta\mu^2 nK}{\sigma^2 L}), \quad \gamma_2 = \frac{1 - \beta}{4L}, \quad \gamma_3 = \frac{\mu(1 - \beta)}{24n\beta L^2}, \quad \gamma = \min\{\gamma_1, \gamma_2, \gamma_3\}. \tag{98}$$

It is easy to verify that $\gamma$ satisfies all conditions in (85), (89), (90), (91) and (94). With (98), we have

$$\Delta\exp(-\frac{\mu\gamma K}{4}) \leq \Delta\left[\exp(-\frac{\mu\gamma_1 K}{4}) + \exp(-\frac{\mu\gamma_2 K}{4}) + \exp(-\frac{\mu\gamma_3 K}{4})\right]$$
$$= \mathcal{O}\left(\frac{\sigma^2 L}{\mu^2 nK} + \Delta\exp(-\frac{\mu(1 - \beta)K}{L}) + \Delta\exp(-\frac{\mu^2(1 - \beta)K}{n\beta L^2})\right) \tag{99}$$

and

$$\frac{\gamma\sigma^2 L}{\mu n} + \frac{c_1\gamma^2\beta^2\sigma^2}{1 - \beta} + \frac{c_2\gamma^2\beta^2 b^2}{(1 - \beta)^2} \leq \frac{\gamma_1\sigma^2 L}{\mu n} + \frac{c_1\gamma_1^2\beta^2\sigma^2}{1 - \beta} + \frac{c_2\gamma_1^2\beta^2 b^2}{(1 - \beta)^2}$$
$$= \tilde{\mathcal{O}}\left(\frac{\sigma^2 L}{\mu^2 nK} + \frac{c_1\beta^2\sigma^2}{\mu^2 K^2(1 - \beta)} + \frac{c_2\beta^2 b^2}{\mu^2 K^2(1 - \beta)^2}\right) \tag{100}$$

Combining (97), (99) and (100), we achieve (83).

$\square$

### E.2.2 Convergence rate of MG-DSGD

As discussed in Appendix E.1.2, it holds that $\tilde{\sigma}^2 \triangleq \sigma^2/R$ and $\tilde{\beta} \triangleq \sqrt{2}(1 - \sqrt{1 - \beta})^R$. If we let

$$R = \left\lceil \frac{\max\{\ln(2), \ln(\frac{nL}{\mu}), \frac{1}{2}\ln(\frac{n}{L}\max\{c_1, \frac{c_2 b^2}{\sigma^2\sqrt{1 - \beta}}\})\}}{\sqrt{1 - \beta}} \right\rceil = \tilde{O}\left(\frac{1}{\sqrt{1 - \beta}}\right), \tag{101}$$

we then have

$$\tilde{\beta} \triangleq \sqrt{2}(1 - \sqrt{1 - \beta})^R \leq \sqrt{2}e^{-\sqrt{1 - \beta}R} \leq \min\left\{\frac{1}{\sqrt{2}}, \sqrt{2}\frac{\mu}{nL}, \left(2\frac{L}{n}\min\left\{\frac{1}{c_1}, \frac{\sigma^2\sqrt{1 - \beta}}{b^2 c_2}\right\}\right)^{\frac{1}{2}}\right\}. \tag{102}$$

which immediately leads to $1 - \tilde{\beta} = \Omega(1)$ and

$$\tilde{\beta}^2\max\{c_1, c_2 R\frac{b^2}{\sigma^2}\} = \tilde{O}\left(\tilde{\beta}^2\max\left\{c_1, c_2\frac{b^2}{\sqrt{1 - \beta}\sigma^2}\right\}\right) = \tilde{O}(\frac{L}{n}). \tag{103}$$

**Theorem 5 (Formal version).** *Under the same assumptions as in Theorem 4 and setting $R$ as in (101), the convergence of $A$ can be bounded for any loss functions $\{f_i\}_{i=1}^n \subseteq \mathcal{F}_{L,\mu}$, and any $W \in \mathcal{W}_{n,\beta}$ by*

$$\mathbb{E}[f(\bar{x}^{(K)})] - f^\star = \tilde{O}\left(\frac{\sigma^2 L}{\mu^2 nT} + \Delta\exp(-\frac{\mu\sqrt{1 - \beta}T}{L})\right). \tag{104}$$

*where $\Delta = \mathbb{E}[f(x^{(0)})] - f^\star$.*

**Remark 11.** *Comparing with the lower bound established in* (44)*, we find the upper bound* (104) *matches it tightly in terms of* $T$*,* $n$*,* $\sigma^2$*, and* $\beta$*, but is a little bit worse in dependence on* $L$ *and* $\mu$*. We will leave it as a future work to develop new algorithms that can match* (44) *tightly in all constants.*

*Proof.* The convergence analysis of DSGD applies to MG-DSGD by replacing $\sigma^2$ with $\tilde{\sigma}^2$, and $\beta$ with $\tilde{\beta}$. As a result, we know from Lemma 13 that if the learning rate is set as

$$\gamma = \min\left\{\frac{4}{\mu K}\ln(\frac{\Delta\mu^2 nK}{\tilde{\sigma}^2 L}), \frac{1-\tilde{\beta}}{4L}, \frac{\mu(1-\tilde{\beta})}{24n\tilde{\beta}L^2}\right\} = \tilde{O}\left(\min\{\frac{1}{\mu\sqrt{1-\beta}T}, \frac{1}{L}\}\right) \qquad (105)$$

it holds that

$$\frac{1}{K+1}\sum_{k=0}^{K}\mathbb{E}\|\nabla f(\bar{x}^{(k)})\|^2$$

$$=\tilde{\mathcal{O}}\Big(\frac{\tilde{\sigma}^2 L}{\mu^2 nK} + \frac{c_1\tilde{\beta}^2\tilde{\sigma}^2}{\mu^2 K^2(1-\tilde{\beta})} + \frac{c_2\tilde{\beta}^2 b^2}{\mu^2 K^2(1-\tilde{\beta})^2} + \Delta\exp(-\frac{\mu(1-\tilde{\beta})K}{L}) + \Delta\exp(-\frac{\mu^2(1-\tilde{\beta})K}{n\tilde{\beta}L^2})\Big)$$

$$\overset{\tilde{\beta}\le\frac{1}{\sqrt{2}};\ T=KR\ge R}{=}\mathcal{O}\left(\frac{\sigma^2 L}{\mu^2 nT} + \frac{c_1\tilde{\beta}^2\sigma^2}{\mu^2 T} + \frac{c_2\tilde{\beta}^2 Rb^2}{\mu^2 T} + \Delta\exp(-\frac{\mu T}{RL}) + \Delta\exp(-\frac{\mu^2 T}{n\tilde{\beta}RL^2})\right)$$

$$\overset{(103),(102)}{=}\tilde{\mathcal{O}}\left(\frac{\sigma^2 L}{\mu^2 nT} + \Delta\exp(-\frac{\mu T}{RL})\right) \overset{(101)}{=} \tilde{\mathcal{O}}\left(\frac{\sigma^2 L}{\mu^2 nT} + \Delta\exp(-\sqrt{1-\beta}\frac{\mu T}{L})\right).$$

$\square$

## F   Experiments

### F.1   Performance in terms of epochs

Fig. 2 depicts the performance of several algorithms in terms of the communicated messages. In Fig. 3 we illustrate their performances in terms of epochs. It is observed that MG-DSGD achieves slightly better validation accuracy than other baselines in both CIFAR-10 and ImageNet dataset.

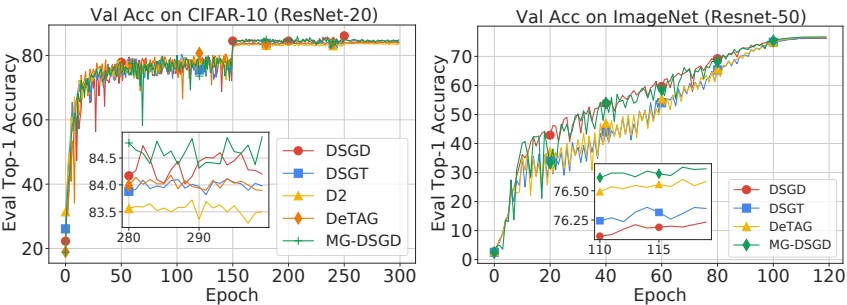

Figure 3: Convergence results in terms of validation accuracy. Left: CIFAR-10 with heterogeneous data ($\alpha = 1$); Right: ImageNet with homogeneous data.

### F.2   Effects of accumulation rounds

We further investigate the performance of the proposed MG-DSGD and DeTAG over different accumulation rounds. The "M-" prefix indicates methods with momentum acceleration. As shown in Table 9, MG-DSGD reaches almost the same performance as DeTAG with less communications. Both methods have performance degradation as accumulation round scales up. We conjecture that while gradient accumulation, which amounts to using large-batch samples in gradient evaluation, can help in the optimization and training stage, it may hurt the generalization performance because gradient with less variance can lead the algorithm to a sharp local minimum. As a result, we recommend using MG-DSGD in applications that are friendly to large-batch training.

Table 9: Effects of round numbers for CIFAR-10 dataset

| MODELS | RESNET18 | | | | RESNET20 | | | |
|---|---|---|---|---|---|---|---|---|
| ROUNDS | 2 | 3 | 4 | 5 | 2 | 3 | 4 | 5 |
| M-DETAG | 94.40±.13 | 93.50±.75 | 92.88±.50 | 92.57±.66 | 91.77±.25 | 91.50±.14 | 91.19±.26 | 90.62±.31 |
| M-OURS | 94.57±.05 | 93.95±.11 | 93.15±.25 | 92.01±.91 | 91.77±.09 | 91.17±.07 | 91.19±.13 | 90.44±.36 |
| DETAG | 93.17±.18 | 92.59±.03 | 92.26±.23 | 91.48±.27 | 88.97±.11 | 89.02±.08 | 88.77±.09 | 88.37±.24 |
| OURS | 93.75±.12 | 92.72±.20 | 92.43±.32 | 91.78±.28 | 89.18±.08 | 88.92±.10 | 88.80±.15 | 88.52±.20 |