# OpenReview forum: "Revisiting Optimal Convergence Rate for Smooth and Non-convex Stochastic Decentralized Optimization"
_NeurIPS.cc/2022/Conference — NeurIPS 2022 Accept_

### Official Review · Reviewer_Xrfd · 2022-06-30

**Rating:** 6
**Confidence:** 5
**Soundness:** 3 good
**Presentation:** 3 good
**Contribution:** 3 good

**Summary:**

The paper study optimal convergence rates in  decentralized nonconvex (smooth) optimization and develops algorithms to achieve such lower complexity bounds up to log factor. More specifically the key contributions are:

(i) Establishing optimal convergence rates for smooth, non convex stochastic optimization for  mesh networks compliant with gossip matrices whose connectivity measure is not $\cos(\pi/n)$, where $n$ is the number of agents. Example of such graphs are grids, hypercubes, toruses, etc. Previous literature considered lower complexity bound only for linear graphs.
The authors established lower complexity bounds for ring-lattice graphs, for connective values within $[0,cos(\pi/n)]$; for large $n$ the interval spans [0,1].  A key enabler of such new results is a tight relationship between the connectivity of the graph and its diameter, a fact that was established before for linear graphs (meaning connectivity values equal to $\cos(\pi/n)$).

(ii) By modifying properly the distributed SGD algorithm, the authors provide a solution method that achieve lower complexity bounds. Differently from the existing method, DeTAC, still achieving lower complexity bounds, the proposed algorithm does not require gradient tracking and hence save one communication (O(d)) per iteration.

(iii) The authors extend the results above  to the case of nonconvex functions satisfying the PL condition.

**Questions:**

Is the proof of lower complexity bounds in Th. 2 significantly different from [38], once the connection between the connectivity and the diameter of ring-lattice graphs is established?

The convergence proof of DSGD has been already established in the literature , e.g. in [12, 25]. I think it would be useful if the authors can elaborate in the paper (and in their reply) on the key difference in their analysis, which lead to slightly different results. What was the key enabler for that?

Do the author have some intuition why MG-DSGD can deal with data heterogeneity's efficient as gradient tracking methods? It would be good to provide some inside substantiated by some technical reasoning.

I would write the upper bounds making explicitly the dependence on L and b, to be consistent with the presentation of complexity bound in the literature. Is the dependence on these parameters of the same order of that in the literature?

I would change the merit function using to measure stationarity: the current one does not capture consensus errors. I would include a consensus error term rather than considering only the gradient along the average. The average is representative of the local iterates only when the consensus error is sufficiently small. It is true however that the consensus error decays faster than the optimization one, but still I would consider a more comprehensive merit function to measure performance of the algorithm.

**Limitations:**

See my comments above under weaknesses.

**Strengths And Weaknesses:**

Strengths:
 The paper advances the state of the art on SDGD-like algorithms for smooth, nonconvex problems. The authors identified a gap in the literature (the lower complexity bound were established for a restricted class of gossip matrices and associated graphs) and provide a solution to it. The overall presentation is good and the paper is easy to read. The comparison with the state of the art is satisfactory (to my knowledge).

Weaknesses:
On the technical side, it is not very clear which new technique has been developed in the proofs to derive some of the results. For instance, once the connection between the connectivity and the diameter of ring-lattice graphs is established, it seems that the proof of Th.2 follows the same path of that in [38]. Same comment apply to the convergence analysis of DSGD; except of a suitable choice of the learning rate, I see a fairly standard proof. Since the proposed MG-DSGD has the same structure of the vanilla DSGD (with a different gossip matrix and a gradient accumulation), same comment applies to its proof of convergence. I do not find the proof under PL particularly more difficult than under strong convexity, the functional relationship between the optimality gap and gradient steps is basically the same.

I would not consider MG-DSGD a new algorithm: it is the DSGD with multiple consensus steps (accelerated) and batch gradients. Both features have been extensively used in the literature.  What is unclear is why the acceleration via multiple round of consensus copes with the data heterogeneity and allows one to get ride of the gradient tracking, which is notoriously known to be beneficial under data heterogeneity.

---

> ### Author Response · Authors · 2022-07-29
> **Response to Reviewer Xrfd: Part II (For comments in Questions)**
>
>
> \
> **1. Lower bound.**
>
> \
> The proof of Thm. 2 is not much different from [Lu and De Sa, 38] except for the ring-lattice part. The existence of the complete proof is to make the paper more self-contained. We will rephrase Thm. 2 into a proposition and shorten the proof into necessary lengths in the revision.
>
> \
> **2. Insights on why MG-DSGD can handle data heterogeneity.**
>
> \
> It is an interesting and novel contribution to establish that the simple DSGD algorithm, with the help of multiple gossip and gradient accumulation, can significantly reduce the influence of data heterogeneity and achieve the optimal convergence rate. The main idea can be simplified as follows. We omit the computation details and irrelevant terms for clarity.
>
> - In Lemma 10 of the Appendix, we show that DSGD in the non-convex and smooth setting converges at rate $O(\frac{\sigma}{\sqrt{nK}} + \frac{\beta^2 b^2}{(1-\beta)^2 K \sigma^2})$ (where we omit irrelevant terms for simplicity). It is observed that **data heterogeneity $b^2$ is associated with $\beta^2$**. The influence of $\beta$ on the numerator of the second term is often ignored in previous works since $\beta$ is close to $1$ in general.
>
> - Next, we set the rounds of multiple gossip relatively large so that the resulting $\tilde{\beta}^2$ can be sufficiently small. For example, we set the gossip round $R = \ln(\frac{b^2}{\sigma^2})/\sqrt{1-\beta} = \tilde{O}(1/\sqrt{1-\beta})$ (see Eq. 72) and hence achieve $\tilde{\beta}^2 = (1-\sqrt{1-\beta})^{2R} \le \exp(-2\sqrt{1-\beta}R) \le \sigma^2/b^2$ (see Eq. 73). This will lead to $\tilde{\beta}^2 b^2/\sigma^2 = O(1)$ which significantly reduces the influence of data heterogeneity.
>
> - It is worth noting that MG-DSGD does not completely remove the influence of data heterogeneity. Since $R \propto \ln(b)$, the data heterogeneity only affects MG-DSGD as a small logarithm term and is hidden in the $\tilde{O}(\cdot)$ notation in Thms. 4 and 5.
>
> \
> The above arguments are simplified to illustrate the main insight. The precise and more complicated proofs are in Appendix D.
>
>
> \
> **3. Key difference between MG-DSGD and DSGD.**
>
> \
> The proof of MG-DSGD is adapted from DSGD. However, MG-DSGD needs additional three steps to achieve the optimal convergence rate:
> - **Step 1**. Adjust the bounds of the vanilla DSGD (if exists) to a desired formulation, in which the influence of $\beta$ on the data heterogeneity must be shown explicitly. (see Lemma 10 and 13 in Appendix).
> - **Step 2**. Choose a proper multiple-gossip round $R$ (cannot be too large or too small, very delicate) so that the influence of data heterogeneity can be significantly reduced. (see Eqs. 72-74, and Eqs. 101-103 in Appendix).
> - **Step 3**. With the help of multiple-gossip and gradient accumulation, we optimize the hyper-parameters inside the bound so that MG-DSGD achieve the optimal convergence rate. (see Thms. 4 and 5 in Appendix).
>
> \
> In fact, we **did not re-prove any exiting arguments** from vanilla DSGD. Useful lemmas from other literature are directly cited (see Lemmas 8, 9, 11 and 12. We made slight modifications in Lemma 12). Note that a proper convergence rate of vanilla DSGD under PL condition does not appear in literature to our knowledge (especially the one in which $\beta$ is associated with data heterogeneity). We spent a few lines on its convergence rate in Lemma 13.
>
> \
> **4. Explicit dependence on $L$ and $b$.**
>
> \
> The dependence on $L$ (or $\mu$) is explicitly shown in the bound Eqs. (75) and (104) in Appendix. Take the non-convex scenario as an example. MG-DSGD converges at rate $O( \frac{\sigma\sqrt{\Delta L}}{\sqrt{nT}} + \frac{L \Delta  \ln(b)}{T\sqrt{1-\beta}})$. In comparison, DeTAG converges at rate $O(\frac{\sigma\sqrt{\Delta L}}{\sqrt{nT}} + \frac{L \Delta \ln(b_0)}{T\sqrt{1-\beta}}) )$ where $b_0$ is the data heterogeneity at the initial iterate (which is smaller than $b$, see the definition in Lu and De Sa [38]). If we ignore the logarithm term, MG-DSGD converges as fast as DeTAG, and they achieve the optimal convergence rate.  We will make $L$ and $b$ explicit in the main paper during the revision.
>
> \
> Note that DeTAG does not have a bound for the PL scenario. We find MG-DSGD maintains the same dependence on L as vanilla DSGD, but is much better in dependence on network topology $\beta$ and data heterogeneity $b$.
>
> \
> **5. Merit function.**
>
> \
> For the PL condition scenario, we do utilize the merit function $\mathbb{E}[f(\bar{x}^k) - f^\star] + L \mathbb{E}\|x^k - \bar{x}^k\|^2$ to establish the bound (see Eq. (83)). However, we do not have a similar bound for the general non-convex scenario. As the reviewer said, the consensus error decays faster and it should not be too difficult to adjust the current analysis to the strong metric function. We will do the change in the revision. However, the current merit function is also standard in literature. Well known references DeTAG [38] and [25] all use the same merit function as us.

---

> ### Author Response · Authors · 2022-07-29
> **Response to Reviewer Xrfd: Part I (For Comments in Weakness)**
>
> \
> We thank the reviewer for the valuable comments. We have attempted to address them as best as we can. We are glad to clarify any further comments or questions from the reviewer.
>
> \
> **1. Main contributions**
>
> We agree with the reviewer that some contributions are important while others may be minor. We would like to clarify on the main contributions of this paper.
>
> \
> *[1.1. Ring-lattice graph]*. We thank the reviewer for the positive comments on the proposed ring-lattice graph. We believe ring-lattice graph makes an important contribution to a critical problem. Without the ring-lattice graph, the lower bound in existing literature only holds for a restrictive scenario in which $\beta = \cos(\pi/n)$.
>
> \
> *[1.2. MG-DSGD]*. We agree that the construction of MG-DSGD is not novel but a direct extension from DSGD. But we believe the insights and theoretical convergence rate brought by MG-DSGD are novel to decentralized optimization community.
>
> - As the reviewer said, MG-DSGD is very simple. It is such an obvious and straight-forward extension from DSGD and few researchers really pay attention to it. However, this paper finds that **it is the simple MG-DSGD that can achieve the optimal convergence rate without any advanced techniques** such as gradient tracking utilized in DeTAG [33].
>
> - On the theory side, MG-DSGD achieves the same convergence rate as DeTAG but with a much simpler convergence analysis (much of the analysis can be adapted from the vanilla DSGD). On the implementation side, MG-DSGD gets rid of the gradient tracking step and hence can save half of the communication overhead and memory storage. **With optimal convergence rate, simpler analysis, faster communications, and cheaper memory costs**, we hope this paper can provide evidence that the DSGD family comes back with more competitive performances than recently-proposed more advanced algorithms such as D2, Exact-Diffusion, DSGT, and DeTAG.
>
>
> - While the construction of MG-DSGD is straight-forward, we believe **it is not obvious to feel the insight why DSGD has the potential to achieve the optimal convergence rate**; otherwise previous works such as [33] may not build optimal algorithms upon the complicated gradient tracking approach. It is long believed that, being sensitive to data heterogeneity, DSGD is inferior to data-heterogeneity-corrected algorithms such as D2, Exact-Diffusion, DSGT. However, this paper finds out that, **multiple-gossip can significantly reduce the influence of data heterogeneity**, and hence endows DSGD with the potential to achieve the optimal convergence rate. We believe it is another interesting and novel contribution. The intuition why multiple gossips can reduce data heterogeneity is discussed below.
>
> \
> We hope the reviewer may find the above two contributions (1.1 and 1.2) are valuable and non-trivial to the community.
>
>
> \
> **2. Other minor contributions**
>
> \
> The other contributions claimed in the paper are on the lower bounds. These lower bounds are **new** because they are for stochastic non-convex setting and are established for a broader class of weight matrix with $\beta \in [0, \cos(\pi/n)]$. Moreover, the lower bound under PL conditions does not appear in any existing literature. However, we agree with the reviewer that these bounds are mainly established upon existing techniques in literature.  The main difference is the utilization of the results from Ring-lattice. Existence of these bounds can make the paper more complete, smooth and self-contained. If the reviewer has a severe concern on that, we will rephrase Theorems 2 and 3 as propositions and shorten their proofs to necessary length.
>
>
> \
> **3. Intuition why multiple gossips can reduce data heterogeneity**
>
> \
> The intuition is based on a simple but easy-to-ignore fact: **the data heterogeneity term in the convergence rate of DSGD is associated with the second largest eigenvalue of the weight matrix**. More specifically, DSGD in the non-convex and smooth setting converges at rate $O(\frac{\sigma}{\sqrt{nK}} + \frac{\beta^2 b^2}{(1-\beta)^2 K \sigma^2})$ (where $b^2$ is the data heterogeneity and $\beta$ is the second largest eigenvalue. We omit irrelevant terms for clarify). Previous works typically ignore the influence of $\beta$ in the numerator of the second term (see, e.g., Theorem 1 in [21]) since $\beta$ is close to $1$ in general. When proper multiple gossips are used, the resulting eigenvalue $\beta^2$ can be sufficiently small which corrects the data heterogeneity term. Please refer to Part II for more details.

---

### Official Review · Reviewer_o2Rv · 2022-07-09

**Rating:** 5
**Confidence:** 3
**Soundness:** 3 good
**Presentation:** 3 good
**Contribution:** 2 fair

**Summary:**

The paper considers decentralized stochastic non-convex optimization problems. The main innovation is the new structure of the connection graph, which can be used to revisiting the lower bounds. In a supplement, the authors apply this graph structure to obtain lower estimates in the non-convex case. Finally, the authors present an algorithm based on FastGossipAverage which achieves lower bounds.

**Questions:**

The question is about experiments. It is rather rhetorical and I am just interested in your opinion here.

I sometimes have to work with the learning problems on decentralized architectures. Despite the fact that I have a quite good theoretical experience in gossip protocol, we never use it (neither with single step, nor with multi steps). It's easier to use tricky all-reduce procedures. For me gossip protocol is about simpler tasks and simpler computing devices (not powerful GPU), for example about signal processing and transmission.

What do you think about it?

**Limitations:**

No potential negative societal impact

**Strengths And Weaknesses:**

**Strengths:**

1) The new ring-based graph structure is interesting to me. Previous methods of deriving lower bounds consist in that we fix $\beta$ (in the literature it is also $\chi$), on the basis of this $\beta$ a connection graph (typically a path) with some number of vertices $n$ is constructed. The new graph structure allows to fix not only $\beta$ but also $n$. This can be important.

2) The rest of the results, including lower bounds for non-convex problems (in the general case and in the PL case), the algorithm that achieves these lower bounds, and the experiments, complement the paper well.

3) The article is quite well written and easy to follow. The literature review is satisfactorily done.

**Weaknesses:**

1) It is a pity that the new graph construction works only for $\beta \in [0; \cos \pi/n]$.

2) Unfortunately, the rest of the results of the work, except for the construction of the graph are not particularly interesting. Most of them are obvious and represent a simple combination of existing works. Read more below.

3) Lower bounds in the general non-convex case can be obtained by simply replacing the graph in the available results in the literature.

4) Lower bounds in the PL case are actually made for the strongly convex case, and this also can be done by simply replacing the graph in the available results. I also do not like that in the main paper the estimates in the PL case does not show the dependence on the factor $L/\mu$. It replaces by $C$. Most likely this is due to the fact that in the lower bounds $C = \sqrt{L/\mu}$, and in the upper estimates $C = L/\mu$. This spoils the optimality of the algorithm, and perhaps the authors want to hide it.

5) The algorithm, which is set as optimal, is simply an analysis of SGD with inexactness + gossip consensus. Such results are obvious and do not cause much interest.

**Conclusion:**

In this paper the interesting thing is the construction of the new graph, with which one can get more accurate lower bounds. The rest of the facts (including experimental ones) complement the paper and make it more solid, but for me are minor in comparison with the new graph. I am hesitant to decide on acceptance and would like to discuss with fellow reviewers and AС their impressions of the paper. I put a borderline reject, but I'm ready to easily change it after discussion.

---

> ### Author Response · Authors · 2022-07-29
> **Response to Reviewer o2Rv: Part II (For comments in Question)**
>
> \
> We thank the reviewer for sharing his precious experience. Our thoughts on this point are as follows
>
> \
> **Gossip algorithms can save communications in large-scale GPU clusters both theoretically and practically**. Its theoretical communication advantage over Ring-Allreduce is clarified in Table 1 in [R1]. Its real single-round communication superiority to Ring-Allreduce in a 256 GPU cluster is listed in Table 17 in [R2]. A deep learning system built upon gossip algorithms can have a much higher throughput than that upon Ring-Allreduce, see Figure 12 in [R1] and the performance test in Github repos [R3] and [R4]. These evidence show that Gossip algorithms do have real values in deep learning.
>
> \
> [R1] B. Ying, et. al., "BlueFog: Make Decentralized Algorithms Practical for Optimization and Deep Learning", arXiv: 2111.04287 \
> [R2] Y. Chen, et. al. "Accelerating Gossip SGD with Periodic Global Averaging", ICML 2021 (arXiv: 2105.09080) \
> [R3] BuGua. Github repo at https://github.com/BaguaSys/bagua \
> [R4] BlueFog. Github repo at https://github.com/Bluefog-Lib/bluefog.
>
> \
> **Gossip algorithms have triggered attentions in industry**.  With the justified value mentioned above, several big companies have conducted active research on gossip algorithms. For example, [R1] is conducted by researchers from Google and Alibaba, [R2] is by Alibaba, [R4] is by Kwai, and [R5] is by Meta. Several news [R6] from IBM also show that gossip algorithms can significantly accelerate deep training in large GPU clusters.
>
> \
> [R5] M. Assran, "Stochastic gradient push for distributed deep learning", ICML 2019. \
> [R6] News: Deep learning training gets 10x performance improvement
>
> \
> **However, we do agree with the reviewer that gossip algorithms progress slowly in deep learning applications.** Few deep learning users (except for those who conduct research on large-scale GPU clusters) use gossip algorithms in their experiments and products. Several fundamental reasons, in our opinion, are as follows.
>
> - *Gossip algorithms can save communication per round, but it may degrade the final test accuracy.*  We observe that massive deep learning users value the final test accuracy more than the training speed. Users are willing to wait for longer time to get a better accuracy. This phenomenon is more evident in academia where a top test accuracy is highly preferred in paper submissions. Moreover, when the scale of GPU cluster is small, the communication efficiency brought by Gossip is very minor. For these reasons, users with just a few GPUs have no motivations to use gossip training.
>
> - *Gossip algorithms are difficult to implement. There are no mature tools to integrate decentralized training into user's deep learning tasks in a painless, high-performance and stable manner.*  Gossip algorithms require peer-to-peer communication protocols between GPUs, and they also need to organize GPUs into various topologies such as ring, grid, or exponential graphs. These have brought significant challenges to the implementation. Fortunately, several research groups from the industry are building tools to support easier implementation for decentralized algorithms, see [R3] and [R4]. But they are not as mature as those tools for centralized training until now.
>
> - *There exist no unified gossip strategies that can extend all optimizers to decentralized versions.* As the reviewer said, ring-allreduce is easy to use. Whether we are using SGD, momentum SGD, ADAM, or other optimizers, implementing them with ring-allreduce is very straightforward and the convergence is the same as the single-node case. However, gossip strategies are not that easy to use. After one develops DSGD for SGD optimizer, it has to develop another specific decentralized algorithms for momentum SGD, or ADAM. Simply extending the gossip strategy in DSGD to mSGD or ADAM optimizer may result in significant performance degradation, see an example in [R7].
>
> [R7] T. Lin, "Quasi-global Momentum: Accelerating Decentralized Deep Learning on Heterogeneous Data", ICML 2021.
>
> \
> **Fairly speaking, there is a long way to go to push decentralized algorithms into popularity in industry.** But all the difficulty we summarized above that gossip algorithms are experiencing right now also **happens to other communication-saving approaches** such as compression, federated training approaches, and asynchronous algorithms. **They are not unique to gossips**. However, we humbly think this also brings chances for us researchers to make a difference in these fields.
>
> \
> We also agree with the reviewer that gossip algorithms have valuable applications in wireless sensor networks, or mobile networks. However, due to lacking experience in these areas, we cannot provide insightful comments on these applications.
>
> \
> We hope these discussions are useful to the reviewer, and we are looking forward to any further comments or questions if any. Thanks again for bringing up such an interesting discussion.

---

> > ### Comment · Reviewer_o2Rv · 2022-08-05
> > **Reply**
> >
> > Thanks very much! Looks interesting!

---

> ### Author Response · Authors · 2022-07-29
> **Response to Reviewer o2Rv: Part I (For comments in Weakness)**
>
> \
> Many thanks for the valuable comments! We have attempted to clarify all the questions as best as we can.
>
> \
> **1. Main contributions**
>
> \
> We agree with the reviewer that some contributions are important while others may be minor. We would like to clarify on the main contributions of this paper.
>
> \
> *[1.1. Ring-lattice graph]*. We thank the reviewer for his positive comments on the proposed ring-lattice graph. We believe ring-lattice graph makes a non-trivial contribution to an important and critical problem.
>
> - In distributed stochastic optimization (including decentralized scenario), the number of computing nodes $n$ is fixed. Only when $n$ is given can a distributed stochastic algorithm show the linear speedup rate $O(\sigma/\sqrt{nT})$. However, no mature techniques are available in existing literature to derive the influence of network topology on the optimal convergence rate in decentralized stochastic optimization in which the network size $n$ is fixed. It is a critical problem needs to be addressed.
>
> - The proposed ring-lattice graph is an effective tool to help address the problem. Before us, the best known result is for weight matrix with $\beta = \cos(\pi/n)$. Our ring-lattice works for $\beta \in [0, \cos(\pi/n)]$, and it is **very close** to the ideal case that $\beta \in [0, 1]$. Note that $\cos(\pi/n) = 1 - \Theta(\frac{1}{n^2})$. When $n$ is large, the interval $[0, \cos(\pi/n)]$ approaches to $[0,1]$ quickly. Moreover, most weight matrices of interest have already been covered by the interval $[0, \cos(\pi/n)]$, please check Table 6 in Appendix.
>
> \
> *[1.2. MG-DSGD]*. While the reviewer thinks MG-DSGD does not cause interest, we would like to share some different thoughts.
>
> - As the reviewer said, MG-DSGD is very simple. It is such an obvious and straight-forward extension from DSGD and few researchers really pay attention to it. However, this paper finds that it is the simple MG-DSGD that can **achieve the optimal convergence rate without any advanced techniques** such as gradient tracking utilized in DeTAG [33]. On the theory side, MG-DSGD achieves the same convergence rate as DeTAG but with a much simpler convergence analysis (much of the analysis can be adapted from the vanilla DSGD). On the implementation side, MG-DSGD gets rid of the gradient tracking step and hence can save half of the communication overhead and memory storage. **With optimal convergence rate, simpler analysis, faster communications, and cheaper memory costs**, we hope this paper can provide evidence that the DSGD family comes back with more competitive performances than recently-proposed  advanced algorithms such as D2, Exact-Diffusion, DSGT, and DeTAG.
>
> - While the construction of MG-DSGD is straight-forward, we believe **it is not obvious to feel the insight why DSGD has the potential to achieve the optimal convergence rate**; otherwise previous works such as [33] may not build optimal algorithms upon the complicated gradient tracking approach. It is long believed that, being sensitive to data heterogeneity, DSGD is inferior to data-heterogeneity-corrected algorithms such as D2, Exact-Diffusion, DSGT. However, this paper finds out that, **multiple-gossip can significantly reduces the influence of data heterogeneity**, and hence endows DSGD with the potential to achieve the optimal convergence rate. We believe it is another interesting and novel contribution, and the other two reviewers a2U2 and Xrfd are all curious about it. It is because the data heterogeneity term in DSGD is associated with the second largest eigenvalue of the weight matrix. When proper multiple gossips are used, the resulting eigenvalue can be sufficiently small which corrects the data heterogeneity term. Please check the response (Reviewer a2U2, Part II.4) for more details.
>
> \
> We hope the reviewer may find the above two contributions are valuable and non-trivial to the community.
>
>
> \
> **2. Other minor contributions**
>
> \
> The other contributions claimed in the paper are on the lower bounds. These lower bounds are **new** because they are for stochastic non-convex setting and are established for a broader class of weight matrix with $\beta \in [0, \cos(\pi/n)]$. Moreover, the lower bound under PL conditions does not appear in any existing literature. However, we agree with the reviewer that these bounds are mainly established upon existing techniques in literature.  The main difference is the utilization of the results from Ring-lattice. Existence of these bounds can make the paper more complete, smooth and self-contained. If the reviewer has a severe concern on that, we will rephrase Theorems 2 and 3 as propositions and shorten their proofs to necessary length.
>
>  \
> **3. Constants $L$ and $\mu$**
>
> \
> As the reviewer requested, we will explicitly list the constants $L$ and $\mu$ in bounds shown in the main paper, not just in Appendix. We agree that the current bound cannot achieve $\sqrt{L/\mu}$ due to the lacking of accelerated tricks.

---

> > ### Comment · Reviewer_o2Rv · 2022-08-05
> > **Reply**
> >
> > Thanks to the authors for the response!
> >
> > 1) The new graph is certainly an interesting contribution. The authors confirmed it again.
> >
> > 2) I understand all the authors' arguments about the optimal algorithm, but I still think this contribution is minor and obvious.
> >
> > а) I don't think the authors' algorithms are fully optimal for two reasons.
> >
> > The upper bounds are obtained for a different class of problems! In the lower estimates, the homogeneity (see line 239) of the data is not assumed. **This is important to emphasize in the work.** (The authors can do it right now and upload the revision).
> >
> > There is an additional logarithmic factor in the upper bounds. At first glance this does not matter, but for several years there was a battle in top conferences over optimal decentralized algorithms in the convex case. And just the essence of this battle was an extra logarithmic factor.
> >
> > b) The technique is not new, I found a work that does the same things in the convex case. Moreover, I found SGD analysis with inaccuracy in the nonconvex case. This is very easy to combine.
> >
> > Rogozin et al. An Accelerated Method For Decentralized Distributed Stochastic Optimization Over Time-Varying Graphs
> >
> > Rogozin et al. TOWARDS ACCELERATED RATES FOR DISTRIBUTED OPTIMIZATION OVER TIME-VARYING NETWORKS
> >
> > Dvurechensky et al. Gradient Method With Inexact Oracle for Composite Non-Convex Optimization
> >
> > c) Multistep gossip protocol works like a centralized protocol with inexactness. There is no function heterogeneity in the estimates for centralized SGD. Therefore, the heterogeneity factor does not appear in the paper under review either. For me the conclusion about *reducing the influence of data heterogeneity* was clear and obvious. It also looks strange given that homogeneity is assumed (line 239).
> >
> > 3) **It is important to reflect $L$ and $\mu$ in the Table and Theorems of the main part!** (The authors can do it right now and upload the revision). Right now it looks like cheating.
> >
> > At this point, I don't want to change the score until the authors correct the places that I think are important.

---

> > > ### Author Response · Authors · 2022-08-07
> > > **We have revised our manuscript accordingly**
> > >
> > > \
> > > We thank the reviewer again for the valuable discussion. Your comments made great sense to us. **We have revised the paper accordingly and uploaded it to the system**. All important revisions are highlighted in blue.
> > >
> > > \
> > > Below are the response to your comments.
> > >
> > > \
> > > **1. Nearly-optimal**
> > >
> > > \
> > > The two arguments by the reviewer are solid. We have changed the word ''optimal'' to ''nearly-optimal'' in the paper. In addition, we add a new Remark 6 in the paper to clarify this limitation:
> > >
> > > >  The rate (10) matches with the lower bound (8) up to logarithm factors. Furthermore, we remark that the gradient similarity assumption  is not required to obtain the lower bound in Theorem 2. Due to these two restrictions, MG-DSGD is a nearly-optimal (not fully-optimal) algorithm.
> > >
> > > \
> > > However, we would like to emphasize that, in terms of the reviewer's criterion, DeTAG is not a fully-optimal algorithm either. It involves logarithm terms in the upper bound, and also uses an additional gradient similarity assumption at $x\_0=0$ (see eqn. (6) in its latest arxiv version), though weaker than ours, to establish its upper bound. To our knowledge, eqn. (6) is not justified and thus not involved (in terms of the pre-specified parameter $\zeta\_0$) in its lower bound construction.
> > >
> > > \
> > > **2. Useful literature**
> > >
> > > \
> > > We thank the reviewer for providing these useful literature. After checking these literature, we agree that our paper shares the same gist as in [R1, R2]: gradient accumulation and multiple gossip. We have acknowledged that in various places in the paper. For example:
> > >
> > > > We proved that the above two optimal complexities can be nearly-attained ... by simply integrating multiple gossip communication [38,R1] and gradient accumulation [39, 55, R1]. (Lines 83 - 85)
> > >
> > > > Inspired by the algorithm development in [R1, 39], we add two components to DSGD: gradient accumulation and multiple gossip. (Lines 215)
> > >
> > > >  This paper finds in MG-DSGD that the multiple gossip technique can significantly reduce the influence of data heterogeneity .... Similar observations also appeared in [R1] albeit in the strongly-convex scenario. (Lines 222-224)
> > >
> > > > Our analysis techniques relate to the classic SGD analysis but with inaccurate oracles in the non-convex case [R3]. (Line 233)
> > >
> > > \
> > > While our paper shares the same gist as [R1], the algorithm development as well as analysis is still different. For example, [R1] utilizes an **increasingly large mini-batch** of data per iteration in its algorithm construction. In Theorem 3.1 in [R1] (arxiv version), the batch-size $r = O(1/\epsilon)$ where $\epsilon$ is the desired accuracy. When $\epsilon$ is small, the batch-size is very large. This implementation can be impractical due to massive memory cost. In contrast, MG-DSGD can utilize constant (independent of iteration or desired accuracy) batch-size per iteration to achieve any desired accuracy $\epsilon$. On the other hand, [R1] establishes theories for strongly-convex scenario only.
> > >
> > > \
> > > **3. Multiple gossip**
> > >
> > > \
> > > We agree that multistep gossip protocol works like a centralized protocol with inexactness. Since we use a fixed round of gossips per iteration, the MG-DSGD is just an approximate of the centralized SGD, and it will not converge to the centralized SGD. For this reason, there is a difference on the heterogeneity assumption between MG-DSGD and CSGD.
> > >
> > > \
> > > The data heterogeneity can be regarded as a quantity to control the inexactness. When data heterogeneity $b^2$ is large (i.e., each local function is very different from each other), we have to use more gossip steps to make MG-DSDG perform similarly to centralized SGD. Theoretically speaking, the number of gossip steps can only be determined when $b^2$ is given, see Eq. (72) in Appendix.  We conjecture this assumption can be removed (or alleviated) by utilizing an increasing number of gossips (as well as mini-batch size) per iteration (so that it converges to centralized SGD).
> > >
> > > \
> > > **4. Constants $L$ and $\mu$**
> > >
> > > \
> > > We have clarified constants $L$ and $\mu$ in Tables 1 and  2 ([R1] is also added as a baseline), and Theorems 2-5. In the non-convex scenario, MG-DSGD can match with the lower bound even in terms of $L$. In the PL scenario, MG-DSGD cannot match with the lower bound in $L$ and $\mu$ due to lacking effective acceleration techniques. We added Remark 7 to acknowledge it. Another possible reason is the established lower bound in PL condition is not tight in terms of $L$ and $\mu$.
> > >
> > > \
> > > **5. Summary**
> > >
> > > \
> > > We thank the reviewer for the very useful comments and suggestions. We are happy to make other changes if the reviewer requests.
> > >
> > > \
> > > **6. References**
> > >
> > > \
> > > [R1] Rogozin et al. An Accelerated Method For Decentralized Distributed Stochastic Optimization Over Time-Varying Graphs
> > >
> > > [R2] Rogozin et al. TOWARDS ACCELERATED RATES FOR DISTRIBUTED OPTIMIZATION OVER TIME-VARYING NETWORKS
> > >
> > > [R3] Dvurechensky et al. Gradient Method With Inexact Oracle for Composite Non-Convex Optimization

---

> > > ### Author Response · Authors · 2022-08-08
> > > **Does our revision resolve your concerns?**
> > >
> > > \
> > > Dear Reviewer,
> > >
> > > \
> > > Thank you very much for the very helpful discussion with us. Your comments have significantly improved the quality of our paper. Does our revision and response resolve your concerns? We are happy to make other revisions if you request.
> > >
> > > \
> > > Best,\
> > > Authors

---

> > > > ### Comment · Reviewer_o2Rv · 2022-08-09
> > > > **Final comment**
> > > >
> > > > Thanks to the authors for the new version of the paper!
> > > >
> > > > I have no more questions left.
> > > >
> > > > This is a good paper with one interesting result and minor additional results. I don't consider it a bad thing, to me a conference paper is one idea around which the paper is based, the other results should complement it.
> > > >
> > > > At this point I would like to discuss with fellow reviewers how they think the new graph idea is interesting enough to be accepted.
> > > > I give 4->5 (can change it in discussion with fellow reviewers) and am ready to move on to a discussion with fellow reviewers.

---

> > > > > ### Author Response · Authors · 2022-08-09
> > > > > **Thanks for the helpful discussion during rebuttals**
> > > > >
> > > > > \
> > > > > Dear Reviewer o2Rv,
> > > > >
> > > > > \
> > > > > Thanks very much for all the valuable comments, helpful discussions, and useful suggestions during the rebuttal, with which our paper has been significantly improved. We really appreciate your time and efforts!
> > > > >
> > > > > \
> > > > > Best,\
> > > > > Authors

---

### Official Review · Reviewer_a2U2 · 2022-07-12

**Rating:** 7
**Confidence:** 4
**Soundness:** 3 good
**Presentation:** 2 fair
**Contribution:** 2 fair

**Summary:**

In this paper, the authors investigate the optimal convergence rate for smooth and nonconvex decentralized stochastic optimization. Specifically, given function class F_L, oracle class O_{\sigma^2}, and weight matrix class W_{n,\beta}, the authors establish a tight lower bound for any \beta within (0,cos(\pi/n)). An algorithm based on multiple-step gossip is also proposed.

Post-rebuttal update: I have raised the score from 5 to 7.



**Questions:**

1. For the optimal rate of decentralized stochastic optimization algorithms, readers should be interested in the following question: “Given a network topology, what are the best algorithm and its rate?” In [33], the optimal rate is in the sense of “given a line network of n nodes, finding an optimal algorithm with the best weight matrix (and hence \beta).” It is limited to the line topology, but consistent to our intuition. In this paper, the optimal rate is in the sense of “given a network of n nodes, finding the worst topology and corresponding weight matrix whose connectivity measure is \beta, and finding an optimal algorithm.” This is a little bit confusing because in most cases we do not prefer the worst weight matrix. Please clarify.
2. “Surprisingly, we find in MG-DSGD that the multiple gossip technique can significantly reduce the influence of data heterogeneity, and the gradient tracking step can be saved to achieve the convergence optimality.” Why?


**Ethics Review Area:**

["I don’t know"]

**Limitations:**

N/A.

**Strengths And Weaknesses:**

Strengths:
1. A tight lower bound for any \beta within (0,cos(\pi/n)).
2. An algorithm that matches the lower bound.

Weaknesses:
1. For the optimal rate of decentralized stochastic optimization algorithms, readers should be interested in the following question: “Given a network topology, what are the best algorithm and its rate?” In [33], the optimal rate is in the sense of “given a line network of n nodes, finding an optimal algorithm with the best weight matrix (and hence \beta).” It is limited to the line topology, but consistent to our intuition. In this paper, the optimal rate is in the sense of “given a network of n nodes, finding the worst topology and corresponding weight matrix whose connectivity measure is \beta, and finding an optimal algorithm.” This is a little bit confusing because in most cases we do not prefer the worst weight matrix. Please clarify.
2. Related to the above comment. From the very beginning the authors mention that “this paper revisits non-convex stochastic decentralized optimization and establishes an optimal convergence rate with general weight matrices.” Weight matrix design should be a part of algorithm development. I cannot fully understand the necessity of finding the worst weight matrix.
3. In the problem setup the authors talk about the zero-respecting policy, but do not mention it in the proof.
4. “Surprisingly, we find in MG-DSGD that the multiple gossip technique can significantly reduce the influence of data heterogeneity, and the gradient tracking step can be saved to achieve the convergence optimality.” Why?
5. Mistakes: The degree of a 10-node complete graph is 9, not 10. In Line 3 of Algorithm 2, z_i^{(r)} should be z_j^{(r)}.
6. For these reasons, I recommend “borderline accept” and would like to see further clarifications from the authors.

---

> ### Author Response · Authors · 2022-07-28
> **Response to Reviewer a2U2 (Part II)**
>
>
> The following discussion is to response the reviewer's comments in Weakness 3, 4, 5 and Question 2.
>
> \
> **1. DSGD.** It is known that the vanilla DSGD suffers from the data heterogeneity issue, i.e., the data heterogeneity will significantly slow down the algorithm's convergence, see the third line in Table 1.
>
> \
> **2. DSGT.** The negative influence of data heterogeneity motivates us to develop advanced algorithms that can get rid of it in convergence rate. DSGT is among these algorithm. With the gradient tracking technique, DSGT has an improved convergence rate as shown in the fifth line in Table 1. The fundamental reason that DSGT can correct the data heterogeneity is to utilize the dynamic consensus averaging to track the globally averaged gradient. DSGT pays a cost to remove data heterogeneity: it incurs twice amount of the communication overheads as much as DSGD per iteration.
>
> \
> **3. DeTAG.** DeTAG is proposed in [33] to achieve the optimal convergence rate in decentralized stochastic optimization. DeTAG can be regarded as DSGT enhanced by multiple gossips per iteration. Built upon DSGT, DeTAG is also immune to data heterogeneity and is converging at optimal rate as shown in the sixth line in Table 1. However, similar to DSGT, **DeTAG
> also incurs twice amount of the communication overheads as much as DSGD per iteration**.
>
> \
> **4. MG-DSGD.** In this paper, we find that multiple gossip is a powerful tool to improve the performance of DSGD. Not only can it help improve the dependence on network topology, but also **it can significantly reduce the influence of data heterogeneity**. With this observation, MG-DSGD does not need the gradient tracking technique utilized in DeTAG any more, and hence **it saves significant communication overheads**. Meanwhile, it achieves the same convergence rate (up to logarithm terms) as DeTAG, see the last line in Table 1. Before this paper, it seems that multiple gossip is not well recognized to be effective to relieve the data heterogeneity issue. **It is somewhat surprising that the simple DSGD algorithm, with the help of multiple gossip and gradient accumulation, can easily achieve the optimal convergence rate**. No other advanced techniques such as gradient tracking, explicit bias-correction, or dual averaging are needed.
>
> \
> The main idea to utilize multiple gossips to solve the data heterogeneity issue is simplified as follows. We omit the computation details and irrelevant terms for clarity.
>
> - An important observation is that **a small second largest eigenvalue $\beta$ can help relieve data heterogeneity in DSGD**. In Lemma 10 of the Appendix, we show that DSGD in the non-convex and smooth setting converges at rate $O(\frac{\sigma}{\sqrt{nK}} + \frac{\beta^2 b^2}{(1-\beta)^2 K \sigma^2})$ (where we omit irrelevant terms for simplicity). It is observed that **data heterogeneity $b^2$ is associated with $\beta^2$**. The influence of $\beta$ on the numerator of the second term in the convergence rate is often ignored in literature since $\beta$ is close to $1$ in general.
>
> - Next, we set the rounds of multiple gossip relatively large so that the resulting $\tilde{\beta}^2$ can be sufficiently small. For example, we set the gossip round $R = \ln(\frac{b^2}{\sigma^2})/\sqrt{1-\beta} = \tilde{O}(1/\sqrt{1-\beta})$ (see Eq. 72) and hence achieve $\tilde{\beta}^2 = (1-\sqrt{1-\beta})^{2R} \le \exp(-2\sqrt{1-\beta}R) \le \sigma^2/b^2$ (see Eq. 73). This will lead to $\tilde{\beta}^2 b^2/\sigma^2 = O(1)$ which significantly reduces the influence of data heterogeneity.
>
> - It is worth noting that MG-DSGD does not completely removes the influence of data heterogeneity. Since $R \propto \ln(b)$, the data heterogeneity only affects MG-DSGD as a logarithm term. In contrast, DeTAG and DSGT can completely remove data heterogeneity with the cost of expensive communication.
>
> The above arguments are customized to illustrate the main insight. The precise and detailed proof shall be checked in Appendix D.
>
> \
> **5. Typos.** We thank the reviewer for the careful check. We will correct them in the revision.
>
> \
> We hope the above explanation can help clarify the reviewer's confusion. We are looking forward to further discussions with the reviewer, and more than happy to clarify any further comments.

---

> > ### Author Response · Authors · 2022-07-28
> > **Response to Reviewer a2U2 (Part II, Continue)**
> >
> > **Zero-respecting Policy.**
> >
> > **1. Non-convex Part:** The lower bound is established with two constructed instances. In the argument with instance 1, zero-respecting policy is required by "Lemma 2 of [33]" (line 663. In Appendix the index appeared as [38] because we cited new papers in the later submitted appendix which caused index mismatch. It should be [33] by Lu and De Sa). For instance 2, zero-respecting policy is required by arguments in lines 679-682. We will clarify them in the revision.
> >
> > **2. Non-convex with PL Part:** The proof with instance 1 follows a hypothesis testing argument which gives an information-theoretic lower bound. The proof actually does not require zero-respecting policy. In other words, the term $\Omega(\frac{\sigma^2}{\mu nT})$ holds as a lower bound even for algorithms that do not obey zero-respecting policy. Therefore we did not mention zero-respecting policy there. For instance 2, zero-respecting policy guarantees the argument in lines 772-774. We will clarify them in the revision.
> >
> > We thank the reviewer for checking the proof detail. We are glad to discuss more if needed.

---

> > > ### Comment · Reviewer_a2U2 · 2022-08-04
> > > **Update**
> > >
> > > I would like to thank the authors for the replies.
> > >
> > > The authors claim that "[33] constructs the worst network topology (and hence the worst weight matrix) over which an algorithm can achieve its fastest convergence." However, in [33], the authors consider \inf_{A \in A_{B,W}} where W is the weight matrix and \sup_{G \in G_{n,D}} where G_{n,D} is the topology class. Then the bound is on the worst topology in the class but the best possible W. Please correct me if I misunderstand anything.
> > >
> > > The authors summarize that the scope of this paper is "if we are considering optimality over the entire class of loss functions, gradient oracles, and weight matrices, it is common to construct the worst (or hardest) loss functions, gradient oracles, and weight matrices over which an algorithm can achieve its fastest convergence." This is very good and should be emphasized from the very beginning. The current paper gives me an impression that the contribution lies in extending \beta from one value in [33] to a range. However, as mentioned above, the settings of this paper and [33] are slightly different.
> > >
> > > I am considering increasing the score to 7.

---

> > > > ### Author Response · Authors · 2022-08-05
> > > > **Many thanks for raising the evaluation score. We really appreciate it!**
> > > >
> > > > \
> > > > Thanks very much for the valuable suggestion. We will clarify the reason to use the worst weight matrix in the very beginning of the paper to avoid confusion.
> > > >
> > > > \
> > > > As to the reviewer's comments on [33], we believe the setting in [33] is the same as ours. In sec. 4.2 of [33] (the latest arxiv version), the following metric is stated:
> > > >
> > > > \
> > > > $\inf\_{A\in{\mathcal{A}\_{B\\,, \\,W}}}\sup\_{f\in \mathcal{F}_{\Delta,\\,L}}\sup\_{O\in \mathcal{O}\_{\sigma^2}}\sup\_{G\in \mathcal{G}\_{n,\\,D}}\text{complexity}(A, f, O, G)$
> > > >
> > > > \
> > > > where we use "complexity" to denote the technical measure in sec. 3.4, [33], which is equivalent to convergence rate. It is worth noting that the class $\mathcal{A}\_{B\\,, \\,W}$ contains all algorithms using a given weight matrix $W$ and mini-batch $B$. For an algorithm $A \in \mathcal{A}\_{B\\,, \\,W}$, there is no flexibility in the choice of $W$ and $B$. In other words, $\inf\_{A\in{\mathcal{A}\_{B\\,, \\,W}}}$ is not equivalent to $\inf\_{A \in \mathcal{A}\_{B\\,, \\,W}}\inf\_{W \in \mathcal{W}\_n}$ in [33].  In fact, the dependence of matrix class $\mathcal{W}$ is missing in the above metric. After carefully checking the proof details, we believe a correct interpretation of the lower bound in [33] should be
> > > >
> > > > \
> > > > $\inf\_{A\in{\mathcal{A}\_{B\\,, \\,W}}}\\;{\sup\_{W\in\mathcal{W}\_n}}\sup\_{f\in \mathcal{F}_{\Delta,\\,L}}\sup\_{O\in \mathcal{O}\_{\sigma^2}}\sup\_{G\in \mathcal{G}\_{n,\\,D}}\text{complexity}(A, f, O, G)$
> > > >
> > > > \
> > > > There are two evidences. First, Corollary 1 in [33] is stated as follows:
> > > >
> > > >
> > > > > "For every $\Delta>0$, $L>0$, $n\in\\{2,\,3,\,4,\dots\\}$, $\sigma>0$, and $B\in\mathbb{N}$, there exists ${\text{a loss function
> > > > }f\in\mathcal{F}\_{\Delta,L}}$, ${\text{a set of underlying oracles }O\in\mathcal{O}\_{\sigma^2}}$, ${\text{a gossip matrix }W\in\mathcal{W}\_n\text{ with second largest eigenvalue being }\lambda = \cos(\pi/n)}$, and ${\text{a graph }G\in\mathcal{G}\_{n,\,D}}$ such that ${\text{no matter what}A\in\mathcal{A}\_{B\\,, \\,W}}$  is used, $T(A, f, O, G)$ will always be lower bounded by $\dots$.
> > > >
> > > > In this statement, the four classes $\mathcal{F}\_{\Delta,L}$, $\mathcal{O}\_{\sigma^2}$, $\mathcal{W}\_n$, $\mathcal{G}\_{n,\,D}$ are stated similarly, which indicates the same role (supremum) they play in the result. Otherwise, the weight matrix $W$ will be stated oppositely to indicate the infimum role it plays.
> > > >
> > > >
> > > > \
> > > > Second, if the lower bound in [33] is established for the optimal weight matrix $W$, the argument in [33] that "In Section 6, we propose DeTAG, a practical algorithm that achieves the lower bound with only a logarithm gap ..." (see the contribution summary in Sec. 1) is not valid. It is because the DeTAG algorithm can use an arbitrary weight matrix (with certain second largest eigenvalue) to achieve that lower bound, not necessarily the optimal one.
> > > > Only when the lower bound in [33] is established for the worst weight matrix, can the claim that "DeTAG achieves the lower bound with arbitrary $W$" makes sense.
> > > >
> > > > \
> > > > In other words, if the lower bound is interpreted as the reviewer previously suggested, there should be an explicit procedure of using/finding the optimal matrix in a so-called "optimal" algorithm. However, there is no such procedure in DeTAG.
> > > >
> > > > \
> > > > We hope these explanation can clarify the reviewer's confusion on the setting in [33]. We are happy to address any further comments or questions.

---

> > > > > ### Comment · Reviewer_a2U2 · 2022-08-05
> > > > > **Update**
> > > > >
> > > > > Thank the authors for the clarification! I recommend to explicitly claim you are using the same setting, either in your final conference paper or in your arxiv version.
> > > > >
> > > > > I have changed the score to 7.

---

> ### Author Response · Authors · 2022-07-28
> **Response to Reviewer a2U2 (Part I)**
>
> Many thanks for the valuable comments! We have attempted to clarify all the questions as best as we can.
>
> \
> The following discussion is to response your comments in Weakness 1, 2 and Question 1.
>
> \
> **1. Optimal algorithms for a given class.** The reviewer might have some misunderstandings on the definition of the optimal algorithm for a given class. In this paper, we are trying to answer the following question.
> \
> \
> *Given a class of non-convex loss functions satisfying the L-smooth assumption (Eq. 2), a class of stochastic gradient oracles satisfying Eq. (4), and a class of network weight matrices with size $n\times n$ and second largest eigenvalue $\beta$ satisfying Eq.(5), what is the optimal algorithm and its convergence rate?*
>
> \
> How to characterize the optimality over a class of loss functions, gradient oracles, or weight matrices? This paper formulate it into a minimax problem listed in Eq. (6) or (7). Take the weight matrix as an example. To find a fastest algorithm over **the whole class** $W_{n,\beta}$, i.e.,  all weight matrices with size $n\times n$ and second largest eigenvalue $\beta$, a standard way is to find an algorithm that achieves fastest convergence when the worst weight matrix belonging to $W_{n,\beta}$ is given. This can be formally expressed as $\inf_{A\in \mathcal{A}} \sup_{W \in \mathcal{W}_{n,\beta}}\|\nabla f(A, W)\|$.
>
> \
> This minimax formulation follows the convention in existing literature. For example, to find the optimal convergence rate for a class of non-convex loss functions, the well-known reference [5] utilizes $\inf_{A\in \mathcal{A}} \sup_{f \in \mathcal{F}} T(A, f)$ (see Sec. 2.3 therein) to formulate the problem where $\mathcal{F}$ is the class of the non-convex and smooth functions.
>
> \
> We agree that the worst weight matrix is not preferred in real specific implementation. However, the worst weight matrix is very useful to gauge how an algorithm performs over the examined class of weight matrices. In contrast, a good weight matrix cannot be used to validate the performance of the algorithm over the class of weight matrix. Even if some algorithm performs well with a specially good weight matrix, it may get degraded using another weight matrix belonging to the same class.
>
> \
> In summary, if we are considering optimality over the entire class of loss functions, gradient oracles, and weight matrices, it is common to construct the worst (or hardest) loss functions, gradient oracles, and weight matrices over which an algorithm can achieve its fastest convergence.
>
> \
> **2. Optimal weight matrix for a given topology.** The reviewer asks an excellent question: *Given a network of $n$ nodes, what is the optimal weight matrix (and hence $\beta$)?* This question is important but is not the focus of this paper. Reference [R1] is to solve this problem. Typically speaking, finding the optimal weight matrix is **decoupled from** the algorithm design. It can be formulated into an
>  optimization problem in which weight matrix eigenvalue $\beta$ is minimized given the network topology constraints.
>
> \
> [R1] S. Boyd, P. Diaconis, L, Xiao, Fastest mixing Markov chain on a graph. 2004.
>
> \
> **3. Results in [33].** To our knowledge, [33] is not ''finding an optimal algorithm with the best weight matrix given a network topology''. Similar to our setting, [33] is finding an optimal convergence complexity given a class of non-convex loss functions, gradient oracles, and networks with size $n$. To achieve it, [33] constructs the worst network topology (and hence the worst weight matrix) over which an algorithm can achieve its fastest convergence. This can be seen from their problem formulation $\inf_{A \in \mathcal{A}} \sup_{g \in G_{n, \Delta}}\|\nabla f(A,g)\|$ in which $G_{n,\Delta}$ is the class of networks with size $n$ and diameter $\Delta$. However, the results in [33] only holds for weight matrices with $\beta = \cos(\pi/n)$ which are mainly resulted from line graphs.
>
> \
> In fact, we are not aware of existing works that can find an *optimal* algorithm with the *best* weight matrix given a network topology. It is a co-design problem in which both the algorithm and the weight matrix are optimized. This co-design problem shall be formulated as $\inf\_{A\in \mathcal{A}\_{W}} \inf\_{W \in \mathcal{W}\_{n,\beta}}\|\nabla f(A, W)\|$ which does not appear in literature to our knowledge. We are only aware of work [R2] in which the optimal $W$ is found for a given **fixed** algorithm EXTRA and a class of networks.
>
> \
> [R2] Y. Chow, W. Shi, T. Wu, and W. Yin, Expander Graph and Communication-Efficient Decentralized Optimization. 2016
>
> \
> We are looking forward to further discussions with the reviewer, and more than happy to clarify any further comments.

---

### Official Review · Reviewer_oRXZ · 2022-07-15

**Rating:** 7
**Confidence:** 2
**Soundness:** 3 good
**Presentation:** 3 good
**Contribution:** 3 good

**Summary:**

This paper revisits non-convex stochastic decentralized optimization and extends existing convergence analysis with special weight matrices to establish an optimal convergence rate with general weight matrices. The algorithms proposed has a convergence rate that matches DeTAG's rate but avoids gradient tracking and saves half of the communication overheads per step.

**Questions:**

Is it possible for a time-varying communication pattern to improve the convergence? It will be nice to include the discussion if some conclusions can be drawn.

**Strengths And Weaknesses:**

The paper studies decentralized optimization algorithm over general communication weight matrix, but the weight matrix has to be fixed over time, so it does not seem to apply to decentralized algorithms where the communication pattern is different in different steps.

When the matrix is fixed, the paper proves the optimal convergence rate for nonconvex objectives (where the lower bound and upper bound match). Also an algorithm MG-DSGD is proposed to achieve that optimal convergence rate while being simpler and less communication costy than existing algorithms such as DeTAG.

---

> ### Author Response · Authors · 2022-07-31
> **Response to Reviewer oRXZ**
>
> \
> We thank the reviewer for bringing up this interesting discussion on time-varying topologies. Our thoughts can be briefly summarized as follows.
>
> - Optimal convergence rate for the family of time-varying weight matrices is worse than that for the family of static weight matrices.
>
> - Some special time-varying weight matrices (not the entire time-varying family) can endow decentralized algorithms with better convergence rate than static weight matrices.
>
> \
> Below are the detailed discussions.
>
> \
> **1. Optimal rate for the entire family of time-varying weight matrices**
>
> \
> Time-varying weight matrices are typically defined as matrices that can vary with time. Static weight matrices are generally regarded as a **subset** of time-varying weight matrices. Recall the optimal convergence rate for static weight matrices is defined as (see Eq. 6) $\inf\_{A \in \mathcal{A}} \sup\_{W \in \mathcal{W}\_{\mathrm{static}}} \mathbb{E}\Vert \nabla f(A, W)\Vert^2$ (we omit the function and gradient oracle classes for simplicity). Similarly, the optimal convergence rate for time-varying weight matrices is $\inf\_{A \in \mathcal{A}} \sup\_{W \in \mathcal{W}\_{\mathrm{dynamic}}} \mathbb{E}\Vert \nabla f(A, W)\Vert^2$. Since $\mathcal{W}\_{\mathrm{static}} \subseteq  \mathcal{W}\_{\mathrm{dynamic}}$,  we can easily derive that
>
> \
> $\inf\_{A \in \mathcal{A}} \sup\_{W \in \mathcal{W}\_{\mathrm{dynamic}}} \mathbb{E}\Vert \nabla f(A, W)\Vert^2 \ge \inf\_{A \in \mathcal{A}} \sup\_{W \in \mathcal{W}\_{\mathrm{static}}} \mathbb{E}\Vert \nabla f(A, W)\Vert^2$
>
> \
> Since a smaller rate is faster, we thus conclude that the optimal convergence rate for time-varying weight matrices is **worse** than that for static time-varying weight matrices.
>
> \
> The above conclusion can be illustrated by results in existing literature. We take the deterministic and strongly-convex scenario as an example. It is proved in [44] that the optimal convergence rate for static weight matrix is $\Omega(\exp(-\sqrt{\mu/L}\sqrt{1-\beta} T)$ while the optimal rate for time-varying weight matrix is given by [22] as $\Omega(\exp(-\sqrt{\mu/L}(1-\beta) T)$ where $\beta \in (0,1)$ is the second largest eigenvalue of the weight matrix. Apparently, the optimal rate for time-varying weight matrix is worse than that for static weight matrix.
>
> \
> **2. Some special time-varying weight matrix can admit better convergence rate**
>
> \
> If we are not considering the entire family of the time-varying weight matrix, some single special weight matrices are identified in literature to endow decentralized algorithms with shorter transient iterations and hence faster convergence rate. In reference [59], the one-peer exponential graph is proposed to boost the performance of DSGD. The one-peer exponential graph is a time-varying weight matrix in which each node cycles through all its neighbors, communicating, only, to a single neighbor per iteration, see Figure 2 therein. This time-varying weight matrix is communication efficient (because each node only communicates to one neighbor per iteration), and it endows DSGD with rate ${O}(1/\sqrt{nT} + n\ln(n)/T)$. In contrast, the corresponding static weight matrix, i.e., the ring (in which each node also  communicates to one neighbor per iteration), has a much slower rate at ${O}(1/\sqrt{nT} + n^3/T)$.
>
> \
> **3. Future work**
>
> \
> The results on optimal rates over time-varying matrices are still very limited in literature. For example, existing results mainly focus on deterministic scenario. Inspired by the reviewer's questions, we will study the optimal convergence rate in the stochastic setting with time-varying networks in the future.
>
> \
> **4. References**
>
> \
> [22] Dmitry Kovalev, et.al.,  Lower bounds and optimal algorithms for smooth and strongly convex decentralized optimization over time-varying networks, NeurIPS 2021.
>
> [44] Kevin Scaman, et. al. Optimal algorithms for smooth and strongly convex distributed optimization in networks. ICML 2017
>
> [59] Bicheng Ying et.al. Exponential graph is provably efficient for decentralized deep training, NeurIPS 2021.
>
> \
> We are looking forward to further discussions from the reviewer, and more than happy to clarify any further comments or questions.

---

### Meta-Review · Area_Chair_hTEJ · 2022-08-25

**Recommendation:** Accept
**Confidence:** Less certain

**Metareview:**

This paper presents some new results on near-optimum algorithms for distributed optimization, nearly matching lower bounds. Most of the reviewers are positive about the contributions of this work. However, one issue that came up is the assumption of bounded gradient dissimilarity, which is essentially a gap between upper and lower bounds. While I am recommending to accept this paper, I believe this gap should be more prominently discussed in the abstract and introduction.



**Award:**

No

---

### Decision · Program_Chairs · 2022-09-14

Accept